# When Data Geometry Meets Deep Function: Generalizing Offline Reinforcement Learning

**Jianxiong Li[1], Xianyuan Zhan[1,2]\*, Haoran Xu[1], Xiangyu Zhu[1], Jingjing Liu[1] & Ya-Qin Zhang[1]\***

[1] Institute for AI Industry Research (AIR), Tsinghua University, Beijing, China
[2] Shanghai Artificial Intelligence Laboratory, Shanghai, China
`li-jx21@mails.tsinghua.edu.cn, zhanxianyuan@air.tsinghua.edu.cn`

## Abstract

In offline reinforcement learning (RL), one detrimental issue to policy learning is the error accumulation of deep $Q$ function in out-of-distribution (OOD) areas. Unfortunately, existing offline RL methods are often over-conservative, inevitably hurting generalization performance outside data distribution. In our study, one interesting observation is that deep $Q$ functions approximate well inside the convex hull of training data. Inspired by this, we propose a new method, *DOGE (Distance-sensitive Offline RL with better GEneralization)*. DOGE marries dataset geometry with deep function approximators in offline RL, and enables exploitation in generalizable OOD areas rather than strictly constraining policy within data distribution. Specifically, DOGE trains a state-conditioned distance function that can be readily plugged into standard actor-critic methods as a policy constraint. Simple yet elegant, our algorithm enjoys better generalization compared to state-of-the-art methods on D4RL benchmarks. Theoretical analysis demonstrates the superiority of our approach to existing methods that are solely based on data distribution or support constraints. Code is available at https://github.com/Facebear-ljx/DOGE.

## 1 Introduction

Offline reinforcement learning (RL) provides a new possibility to learn optimized policies from large, pre-collected datasets without any environment interaction (Levine et al., 2020). This holds great promise to solve many real-world problems when online interaction is costly or dangerous yet historical data is easily accessible (Zhan et al., 2022). However, the optimization nature of RL, as well as the need for counterfactual reasoning on unseen data under offline setting, have caused great technical challenges for designing effective offline RL algorithms. Evaluating value function outside data coverage areas can produce falsely optimistic values; without corrective information from online interaction, such estimation errors can accumulate quickly and misguide policy learning process (Van Hasselt et al., 2018; Fujimoto et al., 2018; Kumar et al., 2019).

Recent model-free offline RL methods investigate this error accumulation challenge in several ways: 1) *Policy Constraint*: directly constraining learned policy to stay inside distribution, or with the support of dataset (Kumar et al., 2019); 2) *Value Regularization*: regularizing value function to assign low values at out-of-distribution (OOD) actions (Kumar et al., 2020b); 3) *In-sample Learning*: learning value function within data samples (Kostrikov et al., 2021b) or simply treating it as the value function of behavioral policy (Brandfonbrener et al., 2021). All three schools of methods share similar traits of being conservative and omitting evaluation on OOD data, which brings benefits of minimizing model exploitation error, but at the expense of poor generalization of learned policy in OOD regions. Thus, a gaping gap still exists when such methods are applied to real-world tasks, where most datasets only partially cover state-action space with suboptimal policies.

Meanwhile, online deep reinforcement learning (DRL) that leverages powerful deep neural network (DNN) with optimistic exploration on unseen samples can yield high-performing policies with promising generalization performance (Mnih et al., 2015; Silver et al., 2017; Degrave et al., 2022;

---

*Corresponding authors

Figure 1: Left: Visualization of *AntMaze* dataset. Data transitions of two small areas on the critical pathways to the destination have been removed (red box). Right: Performance of three SOTA offline RL methods.

Packer et al., 2018). This staring contrast propels us to re-think the question: *Are we being too conservative*? It is well known that DNN has unparalleled approximation and generalization abilities, compared with other function approximators. These attractive abilities have not only led to huge success in computer vision and natural language processing (He et al., 2016; Vaswani et al., 2017), but also amplified the power of RL. Ideally, in order to obtain the best policy, an algorithm should enable offline policy learning on unseen state-action pairs that function approximators (*e.g.*, $Q$ function, policy network) can generalize well, and add penalization only on non-generalizable areas.

However, existing offline RL methods heed too much conservatism on data-related regularizations, while largely overlooking the generalization ability of deep function approximators. Intuitively, let us consider the well-known AntMaze task in the D4RL benchmark (Fu et al., 2020), where an ant navigates from the start to the destination in a large maze. We observe that existing offline RL methods fail miserably when we remove only small areas of data on the critical pathways to the destination. As shown in Figure 1, the two missing areas reside in close proximity to the trajectory data. Simply "stitching" up existing trajectories as approximation is not sufficient to form a near-optimal policy at missing regions. *Exploiting the generalizability of deep function appoximators*, however, can potentially compensate for the missing information.

In our study, we observe that the value function approximated by DNN can interpolate well but struggles to extrapolate (see Section 2.2). Such an "interpolate well" phenomenon is also observed in previous studies on the generalization of DNN (Haley & Soloway, 1992; Barnard & Wessels, 1992; Arora et al., 2019a; Xu et al., 2020; Florence et al., 2022). This finding motivates us to reconsider the generalization of function approximators in offline RL in the context of dataset geometry. Along this line, we discover that a closer distance between a training sample to the offline dataset often leads to a smaller value variation range of the learned neural network, which effectively yields more accurate inference of the value function inside the convex hull (formed by the dataset). By contrast, outside the convex hull, especially in those areas far from the training data, the value variation range usually renders too large to guarantee a small approximation error.

Inspired by this, we design a new algorithm **DOGE** (*Distance-sensitive Offline RL with better GEneralization*) from the perspective of generalization performance of deep $Q$ function. We first propose a state-conditioned distance function to characterize the geometry of offline datasets, whose output serves as a proxy to the network generalization ability. The resulting algorithm learns a state-conditioned distance function as a policy constraint on standard actor-critic RL framework. Theoretical analysis demonstrates the superior performance bound of our method compared to previous policy constraint methods that are based on data distribution or support constraints. Evaluations on D4RL benchmarks validate that our algorithm enjoys better performance and generalization abilities than state-of-the-art offline RL methods.

## 2 DATA GEOMETRY VS. DEEP $Q$ FUNCTIONS

### 2.1 NOTATIONS

We consider the standard continuous action space Markov decision process (MDP) setting, which can be represented by a tuple $(\mathcal{S}, \mathcal{A}, \mathcal{P}, r, \gamma)$, where $\mathcal{S}$ and $\mathcal{A}$ are the state and action space, $\mathcal{P}(s'|s, a)$ is the transition dynamics, $r(s, a)$ is a reward function, and $\gamma \in [0, 1)$ is a discount factor. The objective of the RL problem is to find a policy $\pi(a|s)$ that maximizes the expected cumulative discounted return, which can be represented by a $Q$ function $Q_\theta^\pi(s, a) = \mathbb{E}[\sum_{t=0}^\infty \gamma^t r(s_t, a_t)|s_0 = s, a_0 = a, a_t \sim \pi(\cdot|s_t), s_{t+1} \sim \mathcal{P}(\cdot|s_t, a_t)]$. The $Q$ function is typically approximated by function

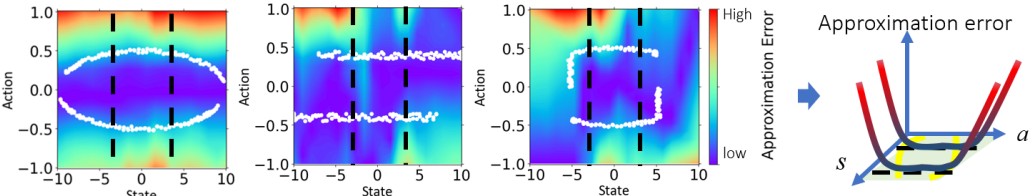

Figure 2: Approximation error of deep $Q$ functions with different dataset geometry. Offline data are marked as white dots (Please refer to Appendix E.5 for detailed experimental setup) .

approximators with learnable parameters $\theta$, such as deep neural networks. Under offline RL setting, we are only given a fixed dataset $\mathcal{D}$ and cannot interact further with the environment. Therefore, the parameters $\theta$ are optimized by minimizing the following temporal difference (TD) error:

$$\min_{\theta} \mathbb{E}_{(s,a,s') \in \mathcal{D}} \left[ \left( r(s,a) + \gamma \mathbb{E}_{a' \sim \pi(\cdot|s')} \left[ Q_{\theta'}^{\pi}(s', a') \right] \right) - Q_{\theta}^{\pi}(s, a) \right]^2 \tag{1}$$

where $Q_{\theta'}^{\pi}$ is the target $Q$ function, which is a delayed copy of the current $Q$ network.

## 2.2 INTERPOLATE VS. EXTRAPOLATE

**Motivating examples.** Let's first consider a set of simple one-dimensional random walk tasks with different offline datasets, where agents at each step can take an action to move in the range of $[-1, 1]$, and the state space is a straight line ranging from $[-10, 10]$. The destination is located at $s = 10$. The closer to the destination, the larger reward the agent gets (*i.e.*, $r = 1$ at $s = 10$, $r = 0$ at $s = -10$). The approximation errors of the learned $Q$ functions are visualized in Figure 2. Note that the approximation errors of the learned $Q$ functions tend to be low at state-action pairs that lie inside or near the boundaries of the convex hull formed by the dataset. Under continuous state-action space, state-action pairs within the convex hull of the dataset can be represented in an interpolated manner (referred as *interpolated data*), *i.e.*, $x_{in} = \sum_{i=1}^{n} \alpha_i x_i$, $\sum_{i=1}^{n} \alpha_i = 1, \alpha_i \geq 0, x_i = (s_i, a_i) \in \mathcal{D}$; similarly, we can define the *extrapolated data* that lie outside the convex hull of the dataset as $x_{out} = \sum_{i=1}^{n} \beta_i x_i$, where $\sum_{i=1}^{n} \beta_i = 1$ and $\beta_i \geq 0$ do not hold simultaneously.

We observe that the geometry of the datasets play a special role on the approximation error of deep $Q$ functions, or in other words, **deep $Q$ functions interpolate well but struggle to extrapolate**. This phenomenon is also reflected in studies on the generalization performance of deep neural networks under a supervised learning setting (Haley & Soloway, 1992; Barnard & Wessels, 1992; Arora et al., 2019a; Xu et al., 2020; Florence et al., 2022), but is largely overlooked in modern offline RL.

**Theoretical explanations.** Based on advanced theoretical machinery from the generalization analysis of DNN, such as neural tangent kernel (NTK) (Jacot et al., 2018), we can theoretically demonstrate that this phenomenon is also carried over to the offline RL setting for deep $Q$ functions. Define $\text{Proj}_{\mathcal{D}}(x) := \arg\min_{x_i \in \mathcal{D}} \|x - x_i\|$ (we denote $\|x\|$ as Euclidean norm) as the projection operator that projects unseen data $x$ to the nearest data point in dataset $\mathcal{D}$. Theorem 1 gives a theoretical explanation of the "interpolate well" phenomenon for deep $Q$ functions under the NTK assumptions (see Appendix B.2 for detailed proofs):

**Theorem 1.** *(Value difference of deep $Q$ function for interpolated and extrapolated data). Under the NTK regime, given an unseen interpolated data $x_{in}$ and an extrapolated data $x_{out}$, then the value difference of deep $Q$ function for interpolated and extrapolated input data can be bounded as:*

$$\|Q_{\theta}(x_{in}) - Q_{\theta}(\text{Proj}_{\mathcal{D}}(x_{in}))\| \leq C_1 (\sqrt{\min(\|x_{in}\|, \|\text{Proj}_{\mathcal{D}}(x_{in})\|)} \sqrt{d_{x_{in}}} + 2d_{x_{in}})$$

$$\leq C_1 (\sqrt{\min(\|x_{in}\|, \|\text{Proj}_{\mathcal{D}}(x_{in})\|)} \sqrt{B} + 2B) \tag{2}$$

$$\|Q_{\theta}(x_{out}) - Q_{\theta}(\text{Proj}_{\mathcal{D}}(x_{out}))\| \leq C_1 (\sqrt{\min(\|x_{out}\|, \|\text{Proj}_{\mathcal{D}}(x_{out})\|)} \sqrt{d_{x_{out}}} + 2d_{x_{out}}) \tag{3}$$

*where $d_{x_{in}} = \|x_{in} - \text{Proj}_{\mathcal{D}}(x_{in})\| \leq \max_{x_i \in \mathcal{D}} \|x_{in} - x_i\| \leq B$ and $d_{x_{out}} = \|x_{out} - \text{Proj}_{\mathcal{D}}(x_{out})\|$ are distances of $x_{in}$ and $x_{out}$ to the nearest data points in dataset $\mathcal{D}$. $B$ and $C_1$ are finite constants.*

Theorem 1 shows that given an unseen input $x$, $Q_{\theta}(x)$ can be controlled by in-sample $Q$ value $Q_{\theta}(\text{Proj}_{\mathcal{D}}(x))$ and the distance $\|x - \text{Proj}_{\mathcal{D}}(x)\|$. The smaller the distance, the more controllable the output of deep $Q$ functions. Therefore, because the distance to dataset is strictly bounded (at

most $B$ for interpolated data), the approximated $Q$ values at interpolated data as well as extrapolated data near the boundaries of the convex hull formed by the dataset cannot be too far off. Moreover, as $d_{x_{out}}$ can take substantially larger values than $d_{x_{in}}$, interpolated data generally enjoys a tighter bound compared with extrapolated data, if the dataset only narrowly covers a large state-action space.

Empirical observations in Figure 2 and Theorem 1 both demonstrate that data geometry can induce different approximation error accumulation patterns for deep $Q$ functions. While approximation error accumulation is generally detrimental to offline RL, a fine-grained analysis is missing in previous studies about where value function can approximate well. We argue that it is necessary to take data geometry into consideration when designing less conservative offline RL algorithms.

## 3    GENERALIZABLE OFFLINE RL FRAMEWORK

In this section, we present our algorithm DOGE (Distance-sensitive Offline RL with better GEneralization). By introducing a specially designed state-conditioned distance function to characterize the geometry of offline datasets, we can construct a very simple, less conservative and also more generalizable offline RL algorithm upon standard actor-critic framework.

### 3.1    STATE-CONDITIONED DISTANCE FUNCTION

As revealed in Theorem 1, the sample-to-dataset distance plays an important role in measuring the controllability of $Q$ values. However, given an arbitrary state-action sample $(s, a)$, naively computing its distance to the closest data point in a large dataset can be costly and impractical. Ideally, we prefer to have a learnable distance function which also has the ability to reflect the overall dataset geometry. Based on this intuition, we design a state-conditioned distance function that can be learned in an elegantly simple supervised manner with desirable properties.

Specifically, we learn the state-conditioned distance function $g(s, a)$ by solving the following regression problem, with state-action pairs $(s, a) \sim \mathcal{D}$ and synthetic noise actions sampled from the uniform distribution over the full action space $\mathcal{A}$:

$$\min_g \mathbb{E}_{(s,a) \sim \mathcal{D}} \left[ \mathbb{E}_{\hat{a} \sim Unif(\mathcal{A})}[\|a - \hat{a}\| - g(s, \hat{a})]^2 \right] \tag{4}$$

In practical implementation, for each $(s, a) \sim \mathcal{D}$, we sample $N$ noise actions uniformly in the action space $\mathcal{A}$ to train $g(\cdot)$. More implementation details can be found in Appendix E. Moreover, with the optimization objective defined in Eq. (4), we can show that the optimal state-conditioned distance function has two desirable properties (proofs can be found in Appendix C):

**Property 1.** *The optimal state-conditioned distance function of Eq. (4) is convex w.r.t. actions and is an upper bound of the distance to the state-conditioned centroid $a_o(s)$ of training dataset $\mathcal{D}$:*

$$g^*(s, \hat{a}) = \mathbb{E}_{a \sim Unif(\mathcal{A})} \left[ C(s, a) \|\hat{a} - a\| \right]$$
$$\geq \|\hat{a} - \mathbb{E}_{a \sim Unif(\mathcal{A})}[C(s, a) \cdot a]\| = \|\hat{a} - a_o(s)\|, \quad \forall \hat{a} \in \mathcal{A}, s \in \mathcal{D} \tag{5}$$

where $C(s, a) = \frac{\mu(s,a)}{\mathbb{E}_{a \sim Unif(\mathcal{A})}\mu(s,a)} \geq 0$, $\mu(s, a)$ is state-action distribution of dataset $\mathcal{D}$. Given a state $s \in \mathcal{D}$, the state-conditioned centroid is defined as $a_o(s) = \mathbb{E}_{a \sim Unif(\mathcal{A})}[C(s, a) \cdot a]$. Since $L_2$-norm is convex and the non-negative combination of convex functions is still convex, $g^*(s, \hat{a})$ is also a convex function *w.r.t.* $\hat{a}$.

**Property 2.** *The negative gradient of the optimal state-conditioned distance function at an extrapolated action $\hat{a}$, $-\nabla_{\hat{a}} g^*(s, \hat{a})$, points inside the convex hull of the dataset.*

From Property 1, we can see that the optimal state-conditioned distance function characterizes data geometry and outputs an upper bound of the distance to the state-conditioned centroid of the training dataset. Property 2 indicates that if we use the learned distance function as a policy constraint, it can drive the learned policy to move inside the convex hull of training data. We visualize the value of the trained state-conditioned distance function in Figure 3. It is clear that the learned distance function can accurately predict the sample-to-dataset centroid distance. By utilizing such distance function, we can constrain the policy based on the global geometric information of training datasets. This

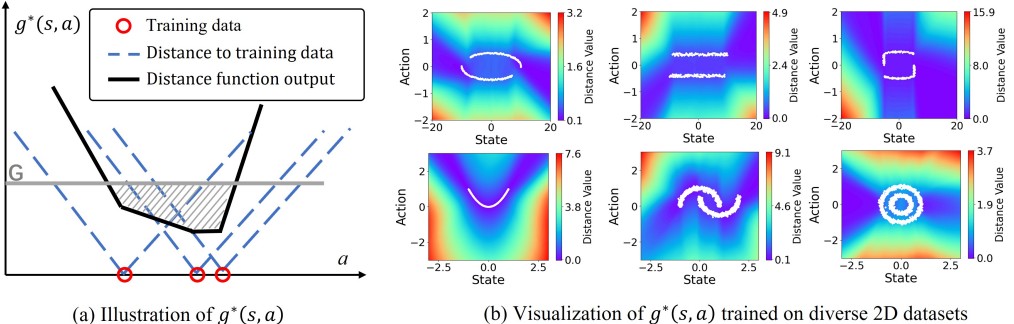

(a) Illustration of $g^*(s, a)$

(b) Visualization of $g^*(s, a)$ trained on diverse 2D datasets

Figure 3: Illustration of the state-conditioned distance function. The output of the optimal distance function is the non-negative combination of the distances to all training data. $G$ is the threshold in Eq. (6) In (b), Offline data are marked as white dots.

desirable property is non-obtainable by simply constraining the policy based on sample-to-sample distance such as the MSE loss between policy generated and dataset actions, which can only bring local geometric information. Moreover, the learned distance function can not only predict well at in-distribution states but also generalize well at OOD states.

## 3.2 DISTANCE-SENSITIVE OFFLINE REINFORCEMENT LEARNING

Capturing the geometry of offline datasets, we now construct a minimalist distance-sensitive offline RL framework, by simply plugging the state-conditioned distance function as a policy constraint into standard online actor-critic methods (such as TD3 (Fujimoto et al., 2018) and SAC (Haarnoja et al., 2018)). This results in the following policy maximization objective:

$$\pi = \arg\max_{\pi} \mathbb{E}_{s\sim\mathcal{D}, a\sim\pi(\cdot|s)} [Q(s, a)] \quad s.t. \ \mathbb{E}_{s\sim\mathcal{D}, a\sim\pi(\cdot|s)}[g(s, a)] \leq G \tag{6}$$

where $G$ is a task-dependent threshold varying across tasks. In our method, we adopt a non-parametric treatment by setting $G$ as the mean output (50% quantile) of the learned distance function on the training dataset, *i.e.*, $\mathbb{E}_{(s,a)\sim\mathcal{D}}[g(s, a)]$, which is approximated over mini-batch samples to reduce computational complexity (see Appendix G for ablation on $G$). The constrained optimization problem in Eq. (6) can be reformulated as:

$$\pi = \arg\max_{\pi}\min_{\lambda} \mathbb{E}_{s\sim\mathcal{D}, a\sim\pi(\cdot|s)} [\beta Q(s, a) - \lambda(g(s, a) - G)] \quad s.t. \ \lambda \geq 0 \tag{7}$$

where $\lambda$ is the Lagrangian multiplier, which is auto-adjusted using dual gradient descent. Following TD3+BC (Fujimoto & Gu, 2021), $Q$ values are rescaled by $\beta = \frac{\alpha}{\frac{1}{n}\sum_{i=1}^{n}|Q(s_i, a_i)|}$ to balance $Q$ function maximization and policy constraint satisfaction, controlled by a hyperparameter $\alpha$. To reduce computations, the denominator of $\beta$ is approximated over mini-batch of samples. The resulting algorithm is easy to implement. In our experiments, we use TD3. Please refer to Appendix E for implementation details.

## 3.3 RELAXATION WITH BELLMAN-CONSISTENT COEFFICIENT

### 3.3.1 BELLMAN-CONSISTENT COEFFICIENT AND CONSTRAINED POLICY SET

The key difference between DOGE and other policy constraint methods lies in that DOGE relaxes the strong full coverage assumption[1] on offline datasets and allows exploitation on generalizable OOD areas. To relax the unrealistic full-coverage assumption, we resort to a weaker condition proposed by (Xie et al., 2021a), the Bellman-consistent coefficient (Definition 1), to measure how well Bellman errors can transfer to different distributions (Theorem 2).

Denote $\|f\|_{2,\mu}^2 := \mathbb{E}_{\mu}[\|f\|^2]$; $\mathcal{T}^{\pi}Q$ is the Bellman operator of policy $\pi$, defined as $\mathcal{T}^{\pi}Q(s, a) := r(s, a) + \gamma\mathbb{E}_{a'\sim\pi(\cdot|s'), s'\sim\mathcal{P}(\cdot|s,a)}[Q(s', a')] := r(s, a) + \gamma\mathbb{P}^{\pi}[Q(s', a')]$. $\mathbb{P}^{\pi}[\cdot]$ is the brief notation for $\mathbb{E}_{a'\sim\pi(\cdot|s'), s'\sim\mathcal{P}(\cdot|s,a)}[\cdot]$. $\mathcal{F}$ is the function class of $Q$ networks. The Bellman-consistent coefficient is defined as:

---

[1]$\sup_{s,a} \frac{v(s,a)}{\mu(s,a)} < \infty$, $v$ and $\mu$ are marginal distributions of the learned policy and the dataset (Le et al., 2019).

**Definition 1.** *(Bellman-consistent coefficient). We define $\mathcal{B}(v, \mu, \mathcal{F}, \pi)$ to measure the distributional shift from an arbitrary distribution $v$ to data distribution $\mu$, w.r.t. $\mathcal{F}$ and $\pi$,*

$$\mathcal{B}(v, \mu, \mathcal{F}, \pi) := \sup_{Q \in \mathcal{F}} \frac{\|Q - \mathcal{T}^\pi Q\|_{2,v}^2}{\|Q - \mathcal{T}^\pi Q\|_{2,\mu}^2} \tag{8}$$

This definition captures the generalization performance of function approximation across different distributions. Intuitively, a small value of $\mathcal{B}(v, \mu, \mathcal{F}, \pi)$ means Bellman errors for policy $\pi$ can accurately transfer from distribution $\mu$ to $v$. This suggests that Bellman errors can transfer well between two distributions even if a large discrepancy exists, as long as the Bellman-consistent coefficient is small.

Based on Definition 1, we introduce the definition of Bellman-consistent constrained policy set.

**Definition 2.** *(Bellman-consistent constrained policy set). We define the Bellman-consistent constrained policy set as $\Pi_\mathcal{B}$. The Bellman-consistent coefficient under the transition induced by $\Pi_\mathcal{B}$ can be bounded by some finite constants $l(k)$:*

$$\mathcal{B}(\rho_k, \mu, \mathcal{F}, \pi) \le l(k) \tag{9}$$

*where $\rho_k = \rho_0 P^{\pi_1} ... P^{\pi_k}, \forall \pi_1, ..., \pi_k \in \Pi_\mathcal{B}$, $\rho_0$ is the initial state-action distribution and $P^{\pi_i}$ is the transition operator induced by $\pi_i$, i.e., $P^{\pi_i}(s', a'|s, a) = \mathcal{P}(s'|s, a)\pi_i(a'|s')$.*

We denote the constrained Bellman operator induced by $\Pi_\mathcal{B}$ as $\mathcal{T}^{\Pi_\mathcal{B}}$, $\mathcal{T}^{\Pi_\mathcal{B}} Q(s, a) := r(s, a) + \max_{\pi \in \Pi_\mathcal{B}} \gamma \mathbb{P}^\pi[Q(s', a')]$. $\mathcal{T}^{\Pi_\mathcal{B}}$ can be seen as a Bellman operator on a redefined MDP, thus theoretical results of MDP also carry over, such as contraction mapping and existence of a fixed point.

### 3.3.2 BELLMAN CONSISTENT COEFFICIENT AND PERFORMANCE BOUND OF DOGE

We show that the policy set induced by DOGE is essentially a Bellman-consistent policy set defined in Definition 2. Meanwhile, the distance constraint in DOGE can produce a small value of $\mathcal{B}$ and hence guarantee the learned policy deviates only to those generalizable areas.

**Theorem 2.** *(Upper bound of Bellman-consistent coefficient). Under the NTK assumption, the Bellman-consistent coefficient $\mathcal{B}(v, \mu, \mathcal{F}, \pi)$ is upper bounded as:*

$$\mathcal{B}(v, \mu, \mathcal{F}, \pi) \le \frac{1}{\epsilon_\mu} \left\| \underbrace{(1-\gamma)Q(s_o, a_o) + R_{\max}}_{\mathcal{B}_1} + \underbrace{C_1\left(C_2\sqrt{d_1} + d_1\right)}_{\mathcal{B}_2} + \underbrace{(2-\gamma)C_1\mathbb{P}^\pi\left(C_2\sqrt{d_2} + d_2\right)}_{\mathcal{B}_3} \right\|_{2,v}^2 \tag{10}$$

*where we denote $x = (s, a)$ and $x' = (s', a')$. $x_o = \mathbb{E}_{x \sim \mathcal{D}}[x]$ is the centroid of offline dataset. $d_1 = \|x - x_o\|$ and $d_2 = \|x' - x_o\|$ are the sample-to-centroid distances. $C_2 = \sqrt{\sup_{x \in \mathcal{S} \times \mathcal{A}} \|x\|}$ is related to the upper bound of the input scale. $\epsilon_\mu$ is the lower bound of Bellman error (square) for $\pi$ under distribution $\mu$, i.e., $\epsilon_\mu \le \|Q - \mathcal{T}^\pi Q\|_{2,\mu}^2$.*

The RHS of Eq. (10) contains four parts: $\frac{1}{\epsilon_\mu}$, $\mathcal{B}_1$, $\mathcal{B}_2$ and $\mathcal{B}_3$. It is reasonable to assume $\epsilon_\mu > 0$, because of the approximation error of $Q$ networks and the distribution mismatch between $\mu$ and $\pi$. $\mathcal{B}_1$ is only dependent on the $Q$ value $Q(s_o, a_o)$ at the centroid of the dataset and the max reward $R_{\max}$. $\mathcal{B}_2$ is related to distance $d_1$ and distribution $v$. $\mathcal{B}_3$ is related to $d_2$, $v$ and $\mathbb{P}^\pi$. To be mentioned, the distance regularization in DOGE compels the learned policy to output the action that is near the state-conditioned centroid of dataset, thus $\mathcal{B}_2$ and $\mathcal{B}_3$ can be driven to small values. Therefore, the RHS of Eq. (10) can be bounded by finite constants under DOGE, which shows that the constrained policy set induced by DOGE is essentially a Bellman-consistent constrained policy set.

Then, the performance gap between the policy learned by DOGE and the optimal policy can be bounded as given in Theorem 3. See Appendix D.1 and D.2 for the proof of Theorem 2 and 3.

**Theorem 3.** *(Performance bound of the learned policy by DOGE). Let $Q^{\Pi_\mathcal{B}}$ be the fixed point of $\mathcal{T}^{\Pi_\mathcal{B}}$, i.e., $Q^{\Pi_\mathcal{B}} = \mathcal{T}^{\Pi_\mathcal{B}} Q^{\Pi_\mathcal{B}}$, and $\epsilon_k = Q^k - \mathcal{T}^{\Pi_\mathcal{B}} Q^{k-1}$ is the Bellman error at the $k$-th iteration. $\|f\|_\mu := \mathbb{E}_\mu[\|f\|]$. The performance of the learned policy $\pi_n$ is bounded by:*

$$\lim_{n \to \infty} \|Q^* - Q^{\pi_n}\|_{\rho_0} \le \frac{2\gamma}{(1-\gamma)^2} \left[ L(\Pi_\mathcal{B}) \sup_{k \ge 0} \|\epsilon_k\|_\mu + \frac{1-\gamma}{2\gamma} \alpha(\Pi_\mathcal{B}) \right] \tag{11}$$

where $L(\Pi_{\mathcal{B}}) = \sqrt{(1-\gamma)^2 \sum_{k=1}^{\infty} k\gamma^{k-1}l(k)}$, which is similar to the concentrability coefficient in BEAR (Kumar et al., 2019) but in a different form. Note that $l(k)$ is related to the RHS of Eq. (10) and can be driven to a small value by DOGE according to Theorem 2. $\alpha(\Pi_{\mathcal{B}}) = \|\mathcal{T}^{\Pi_{\mathcal{B}}}Q^{\Pi_{\mathcal{B}}} - \mathcal{T}Q^*\|_{\infty}$ is the suboptimality constant, which is similar to $\alpha(\Pi) = \|\mathcal{T}^{\Pi}Q^{\Pi} - \mathcal{T}Q^*\|_{\infty}$ in BEAR.

Compared with BEAR, DOGE allows a policy shift to some generalizable OOD areas and relaxes the strong full-coverage assumption. In addition, we have $L(\Pi_{\mathcal{B}}) \leq L(\Pi) \propto \frac{\rho_0 P^{\pi_1}...P^{\pi_k}}{\mu(s,a)}$, where $L(\Pi)$ is the concentrability coefficient in BEAR. This is evident when $\mu(s,a) = 0$ and $\rho_0 P^{\pi_1}...P^{\pi_k}(s,a) > 0$, $L(\Pi_{\mathcal{B}})$ can be bounded by finite constants but $L(\Pi) \to \infty$. Moreover, as $\Pi_{\mathcal{B}}$ extends the policy set to cover more generalizable OOD areas ($\Pi \subseteq \Pi_{\mathcal{B}}$) and produces a larger feasible region for optimization, lower degree of suboptimality can be achieved (*i.e.*, $\alpha(\Pi_{\mathcal{B}}) \leq \alpha(\Pi)$) compared to only performing optimization on $\Pi$. Therefore, we can see that DOGE enjoys a tighter performance bound than previous more conservative methods when allowed to exploit generalizable OOD areas.

## 4 EXPERIMENTS

For evaluation, We compare DOGE and prior offline RL methods over D4RL Mujoco and AntMaze tasks (Fu et al., 2020). Mujoco is a standard benchmark commonly used in previous work. AntMaze tasks are far more challenging due to the non-markovian and mixed-quality offline dataset, the stochastic property of environments, and the high dimensional state-action space. Implementation details, experimental setup and additional experimental results can be found in Appendix E and F.

### 4.1 COMPARISON WITH SOTA

We compare DOGE with model-free SOTA methods, such as TD3+BC (Fujimoto & Gu, 2021), CQL (Kumar et al., 2020b) and IQL (Kostrikov et al., 2021b). For fairness, we use the "-v2" datasets for all methods. For most Mujoco tasks, we report the scores from the IQL paper. We obtain the other results using the authors' or our implementations. For AntMaze tasks, we obtain the results of CQL, TD3+BC, and IQL using the authors' implementations. For BC (Pomerleau, 1988), BCQ (Fujimoto et al., 2019) and BEAR (Kumar et al., 2019), we report the scores from (Fu et al., 2020). All methods are evaluated over the final 10 evaluations for Mujoco tasks and 100 for AntMaze tasks.

Table 1 shows that DOGE achieves comparable or better performance than SOTA methods on most Mujoco and AntMaze tasks. Compared to other policy constraint approaches such as BCQ, BEAR and TD3+BC, DOGE is the first policy constraint method to successfully solve AntMaze-medium and AntMaze-large tasks. Note that IQL is an algorithm designed for multi-step dynamics programming and attains strong advantage on AntMaze tasks. Nevertheless, DOGE can compete with or even surpass IQL on most AntMaze tasks, by only employing a generalization-oriented policy constraint. These results illustrate the benefits of allowing policy learning on generalizable OOD areas.

### 4.2 EVALUATION ON GENERALIZATION

To evaluate the generalization ability of DOGE, we remove small areas of data from the critical pathways to the destination in AntMaze medium and large tasks, to construct an OOD dataset. The two removed areas reside in close proximity to the trajectory data (see Figure 1). We evaluate representative methods (such as TD3+BC, CQL, IQL) and DOGE on these modified datasets. Figure 4 shows the comparison before and after data removal.

For such a dataset with partial state-action space coverage, existing policy constraint methods tend to over-constrain the policy to stay inside the support of a dataset, where the optimal policy is not well-covered. Value regularization methods suffer from deteriorated generalization performance, as the value function is distorted to assign low value at all OOD areas. In-sample learning methods are only guaranteed to retain the best policy within the partially covered dataset (Kostrikov et al., 2021b). As shown in Figure 4, all these methods struggle to generalize well on the missing areas and suffer severe performance drop, while DOGE maintains competitive performance. This further demonstrates the benefits of relaxing over-conservatism in existing methods.

Table 1: Average normalized scores and standard deviations over 5 seeds on benchmark tasks

| Dataset | BC | BCQ | BEAR | TD3+BC | CQL | IQL | DOGE(ours) |
|---|---|---|---|---|---|---|---|
| hopper-r | 4.9 | 7.1 | 14.2 | 8.5±0.6 | 8.3±0.2 | 7.9±0.4 | **21.1±12.6** |
| halfcheetah-r | 0.2 | 8.8 | 15.1 | 11.0±1.1 | **20.0±0.4** | 11.2±2.9 | 17.8±1.2 |
| walker2d-r | 1.7 | 6.5 | **10.7** | 1.6±1.7 | 8.3±0.1 | 5.9±0.5 | 0.9 ±2.4 |
| hopper-m | 52.9 | 56.7 | 51.9 | 59.3±4.2 | 58.5±2.1 | 66.2±5.7 | **98.6±2.1** |
| halfcheetah-m | 42.6 | 47.0 | 41.0 | **48.3±0.3** | 44.0±5.4 | 47.4±0.2 | 45.3±0.6 |
| walker2d-m | 75.3 | 72.6 | 80.9 | 83.7±2.1 | 72.5±0.8 | 78.3±8.7 | **86.8±0.8** |
| hopper-m-r | 18.1 | 53.3 | 37.3 | 60.9±18.8 | **95.0±6.4** | 94.7±8.6 | 76.2±17.7 |
| halfcheetah-m-r | 36.6 | 40.4 | 29.7 | 44.6±0.5 | **45.5±0.5** | 44.2±1.2 | 42.8±0.6 |
| walker2d-m-r | 26.0 | 52.1 | 18.5 | 81.8±5.5 | 77.2±5.5 | 73.8±7.1 | **87.3±2.3** |
| hopper-m-e | 52.5 | 81.8 | 17.7 | 98.0±9.4 | **105.4±6.8** | 91.5±14.3 | 102.7±5.2 |
| halfcheetah-m-e | 55.2 | 89.1 | 38.9 | 90.7±4.3 | **91.6±2.8** | 86.7±5.3 | 78.7±8.4 |
| walker2d-m-e | 107.5 | 109.5 | 95.4 | 110.1±0.5 | 108.8±0.7 | 109.6±1.0 | **110.4±1.5** |
| locomation total | 473.5 | 624.9 | 451.3 | 698.5±49.0 | 726.1±31.7 | 717.4±55.9 | **768.6±55.4** |
| antmaze-u | 65.0 | 78.9 | 73.0 | 91.3±5.7 | 84.8±2.3 | 88.2±1.9 | **97.0±1.8** |
| antmaze-u-d | 55.0 | 55.0 | 61.0 | 54.6±16.2 | 43.3±5.4 | **66.7±4.0** | 63.5±9.3 |
| antmaze-m-p | 0.0 | 0.0 | 0.0 | 0.0 | 65.2±4.8 | 70.4±5.3 | **80.6±6.5** |
| antmaze-m-d | 0.0 | 0.0 | 8.0 | 0.0 | 54.0±11.7 | 74.6±3.2 | **77.6±6.1** |
| antmaze-l-p | 0.0 | 6.7 | 0.0 | 0.0 | 18.8±15.3 | 43.5±4.5 | **48.2±8.1** |
| antmaze-l-d | 0.0 | 2.2 | 0.0 | 0.0 | 31.6±9.5 | **45.6±7.6** | 36.4±9.1 |
| antmaze-total | 120.0 | 142.8 | 142.0 | 145.9±21.9 | 297.7±49.0 | 389.0±26.5 | **403.3±40.9** |

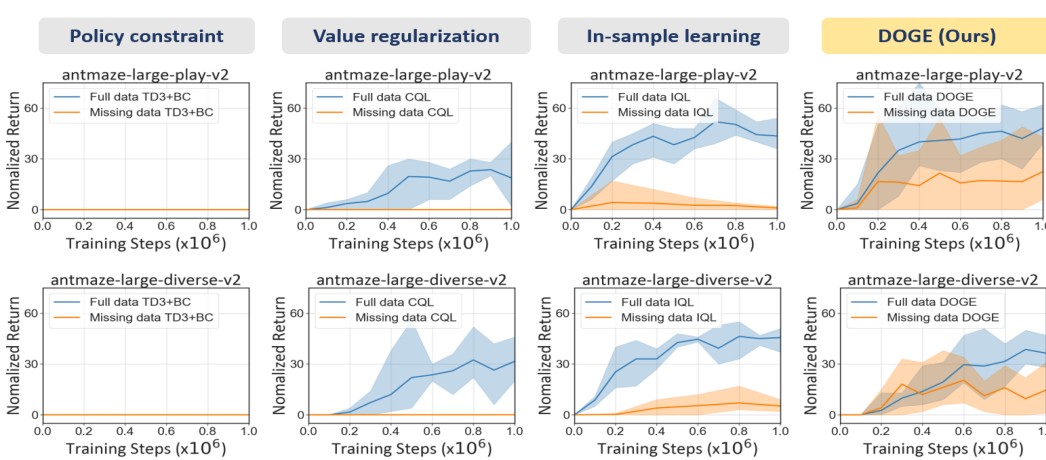

Figure 4: Generalization performance after removing data from AntMaze large tasks (see Appendix F.1 for detailed setup and additional results on AntMaze medium tasks).

## 4.3 ABLATION STUDY

We conduct ablation studies to evaluate the impact of the hyperparameter $\alpha$, the non-parametric distance threshold $G$ in Eq. (6), and the number of noise actions $N$ used to train the distance function. For $\alpha$, we add or subtract 2.5 to the original value; for $G$, we choose 30%, 50%, 70% and 90% upper quantile of the distance values in mini-batch samples; for $N$, we choose $N = 10, 20, 30$.

Compared to $N$ and $\alpha$, we find that $G$ has a more significant impact on the performance. Figure 5b shows that an overly restrictive $G$ (30% quantile) results in a policy set too small to cover near-optimal policies. A more tolerant $G$, on the other hand, is unlikely to cause excessive error accumulation and achieves relatively good performance. In addition, Figure 5a and Figure 5c show that performance is stable across variations of hyperparameters, indicating that our method is hyperparameter-robust.

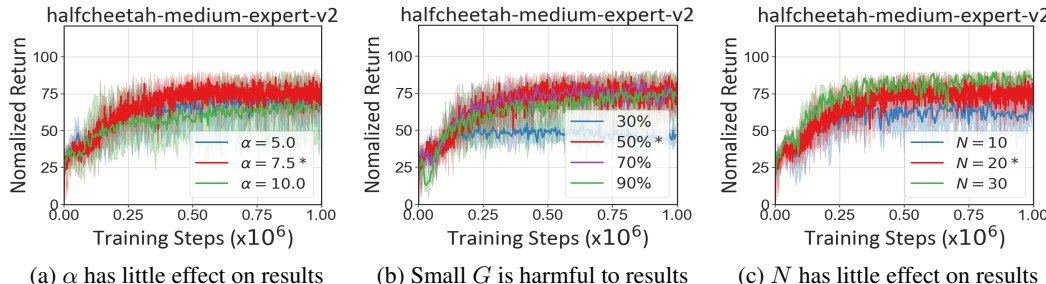

(a) $\alpha$ has little effect on results    (b) Small $G$ is harmful to results    (c) $N$ has little effect on results

Figure 5: Ablation results. The default parameters in our implementation are marked by *. The error bars indicate min and max over 5 seeds. See Appendix G for more detailed ablation studies.

## 5    RELATED WORK

To prevent distributional shift and exploitation error accumulation when inferring the value function at unseen samples, a direct approach is to restrict policy learning from deviating to OOD areas. To make sure the leaned policy stays inside the distribution or support of training data, These policy constraint methods either carefully parameterize the learned policy (Fujimoto et al., 2019; Matsushima et al., 2020), or use explicit divergence penalties (Kumar et al., 2019; Wu et al., 2019; Fujimoto & Gu, 2021; Xu et al., 2021; Dadashi et al., 2021) or implicit divergence constraints (Peng et al., 2019; Nair et al., 2020; Xu et al., 2022a). The theories behind these methods typically assume full state-action space coverage of the offline datasets(Le et al., 2019; Kumar et al., 2019). However, policy constraint under full-coverage assumption is unrealistic in most real-world settings, especially on datasets with partial coverage and only sub-optimal behavior policies. Some recent works try to relax the full-coverage assumption to partial coverage by introducing different distribution divergence metrics, but only in theoretical analysis (Liu et al., 2020; Zanette et al., 2021; Xie et al., 2021b; Uehara & Sun, 2021; Xie et al., 2021a). Our method is an enhanced policy constraint method, where we relax the full-coverage assumption and allow the policy to learn on OOD areas where networks can generalize well.

Another type of offline RL method, value regularization (Kumar et al., 2020b; Kostrikov et al., 2021a; Yu et al., 2021; Xu et al., 2022b; 2023), directly penalizes the value function to produce low values at OOD actions. In-sample learning methods (Brandfonbrener et al., 2021; Kostrikov et al., 2021b), on the other hand, only learn the value function within data or treat it as the value function of the behavior policy. Compared with our approach, these methods exercise too much conservatism, which limits the generalization performance of deep neural networks on OOD regions, largely weakening the ability of dynamic programming. There are also uncertainty-based and model-based methods that regularize the value function or policy with epistemic uncertainty estimated from model or value function (Janner et al., 2019; Yu et al., 2020; Uehara & Sun, 2021; Wu et al., 2021; Zhan et al., 2022; Bai et al., 2021). However, the estimation of the epistemic uncertainty of DNN is still an under-explored area, with results highly dependent on evaluation methods and the structure of DNN.

## 6    CONCLUSION

In this study, we provide new insights on the relationship between approximation error of deep $Q$ functions and geometry of offline datasets. Through empirical and theoretical analysis, we find that deep $Q$ functions attain relatively low approximation error when interpolating rather than extrapolating the dataset. This phenomenon motivates us to design a new algorithm, DOGE, to empower policy learning on OOD samples within the convex hull of training data. DOGE is simple yet elegant, by plugging a dataset geometry-derived distance constraint into TD3. With such a minimal surgery, DOGE outperforms existing model-free offline RL methods on most D4RL tasks. We theoretically prove that DOGE enjoys a tighter performance bound compared with existing policy constraint methods under the more realistic partial-coverage assumption. Empirical results and theoretical analysis suggest the necessity of re-thinking the conservatism principle in offline RL algorithm design, and points to sufficient exploitation of the generalization ability of deep $Q$ functions.

ACKNOWLEDGMENTS

This work is supported by National Key Research and Development Program of China under Grant (2022YFB2502904). This work is also supported by Baidu Inc. through Apollo-AIR Joint Research Center. The authors would also like to thank the anonymous reviewers for their feedback on the manuscripts. Jianxiong Li would like to thank Zhixu Du, Yimu Wang, Li Jiang, Haoyi Niu, Hao Zhao and all colleagues in AIR-Dream group for valuable discussions.

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

## A  SKETCH OF THEORETICAL ANALYSIS

In this section, we present in Figure 6 a sketch of the overall logical flow in our theoretical analyses and the proposed algorithm, DOGE. We start by analyzing the effects of data geometry on the generalization patterns of deep Q-functions. We find that a small sample-to-dataset distance leads to a tightened Q-function approximation error and thus interpolation enjoys better generalization properties than extrapolation (Theorem 1). Motivated by this, we propose DOGE, which tries to control the upper bound of the sample-to-centroid distance to be small (Property 1) and enforces a convex hull based policy constraint (Property 2). Then, we dive deeper and find that the upper bound of the Bellman-consistent coefficient is well controlled by sample-to-centroid distance and thus DOGE enjoys a bounded bellman-consistent coefficient (Theorem 2). Based on these findings, we can derive a tighter performance bound of DOGE as compared to support constraint methods like BEAR (Theorem 3).

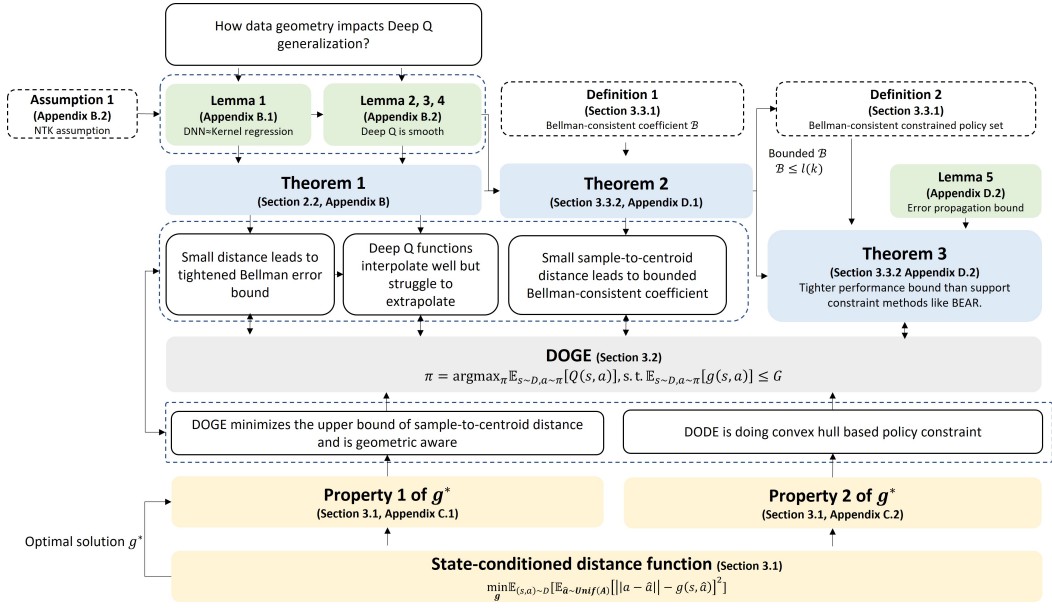

Figure 6: Sketch of theoretical analysis

## B  THEORETICAL ANALYSIS OF THE IMPACT OF DATA GEOMETRY ON DEEP $Q$ FUNCTIONS

To analyze the generalization of a function approximator, one can refer to some classical methods such as Rademacher complexity (Bartlett & Mendelson, 2002) and VC-dimension (Vapnik & Chervonenkis, 2015). However, the generalization bounds that obtained by these methods are usually trivial and cannot explain the generalization behavior in the overparameterized regime (Zhang et al., 2021). Recent breakthroughs in neural tangent kernel (NTK) shed light on the generalization of DNN. NTK builds the connection between the training dynamics of DNN and the solution of the kernel regression *w.r.t.* NTK, and is widely used in recent analysis of DNN generalization (Jacot et al., 2018; Arora et al., 2019b; Bietti & Mairal, 2019). What's more, NTK is also a popular analyzing tool in the convergence and optimality of deep RL (Cai et al., 2019; Fan et al., 2020; Kumar et al., 2020a; Xiao et al., 2021) and thus is used in our study.

### B.1  NEURAL TANGENT KERNEL

We denote a general neural network by $f(\theta, x) : \mathbb{R}^d \to \mathbb{R}$, where $\theta$ is all the parameters in the network and $x \in \mathbb{R}^d$ is the input. Given, a training dataset $\{(x_i, y_i)\}_{i=1}^n$, the parameters $\theta$ are optimized by minimizing the squared loss function, *i.e.*, $\mathcal{L}(\theta) = \frac{1}{2} \sum_{i=1}^n (f_\theta(x_i) - y_i)^2$ by gradient

descent. The dynamics of the networks output can be formulated by Lemma 1 (Lemma 3.1. of (Arora et al., 2019b)); see (Arora et al., 2019b) for the proof of Lemma 1.

**Lemma 1.** *Consider minimizing the squared loss $\mathcal{L}(\theta)$ by gradient descent with infinitesimally small learning rate, i,e., $\frac{d\theta(t)}{dt} = -\nabla\mathcal{L}(\theta(t))$. Let $\mathbf{u}(t) = (f(\theta(t), x_i))_{i \in [n]} \in \mathbb{R}^n$ be the network outputs on all $x_i$'s at time $t$, and $\mathbf{Y} = (y_i)_{i \in [n]}$ be the desired outputs. Then $\mathbf{u}(t)$ follows the following evolution, where $\mathbf{H}(t)$ is an $n \times n$ positive semidefinite matrix whose $(i, j)$-th entry is $\left\langle \frac{\partial f(\theta(t), x_i)}{\partial \theta}, \frac{\partial f(\theta(t), x_j)}{\partial \theta} \right\rangle$:*

$$\frac{d\mathbf{u}(t)}{dt} = -\mathbf{H}(t) \cdot (\mathbf{u}(t) - \mathbf{Y}). \tag{12}$$

Plenty of works (Jacot et al., 2018; Arora et al., 2019b; Allen-Zhu et al., 2019; Xu et al., 2020) study the dynamics of the neural networks' training process and find that if the width of networks is sufficiently large, $\mathbf{H}(t)$ stays almost constant during training, *i.e.*, $\mathbf{H}(t) = \mathbf{H}(0)$. What's more, if the neural networks' parameters are randomly initialized with certain scales and the networks width goes to infinity, $\mathbf{H}(0)$ converges to a fixed matrix $\mathbf{K}$, called neural tangent kernel (NTK) (Jacot et al., 2018).

$$\mathbf{K}(x, x') = \mathbb{E}_{\theta \sim W} \left\langle \frac{\partial f(\theta(t), x)}{\partial \theta}, \frac{\partial f(\theta(t), x')}{\partial \theta} \right\rangle \tag{13}$$

where, $W$ is Gaussian distribution. The training dynamics in Lemma 1 is identical to the dynamics of kernel regression under gradient flow, because $\mathbf{K}$ stays constant during training when the width of neural networks goes to infinity. Then, the final prediction function ($t \to \infty$, assuming $\mathbf{u}(0) = 0$) is equal to the kernel regression solution:

$$f_{ntk}(x) = (\mathbf{K}(x, x_1), ..., \mathbf{K}(x, x_n)) \cdot \mathbf{K}_{train}^{-1} \mathbf{Y} \tag{14}$$

where $\mathbf{K}_{train}^{-1}$ is the $n \times n$ NTK for the training data (the state-action pair $x = (s, a)$ in the policy evaluation in offline RL) and stays constant during training once the training data is fixed. $\mathbf{Y}$ is the training labels ($r(s, a) + \gamma \mathbb{E}_{a' \sim \pi(\cdot|s')}[Q_{\theta'}(s', a')]$ in offline RL). $\mathbf{K}(x, x_i)$ is the kernel value between test data $x$ and training data $x_i$. We denote the feature map of $\mathbf{K}(\cdot, \cdot)$ as $\Phi(\cdot)$, and $\mathbf{K}(x, x') = \langle \Phi(x), \Phi(x') \rangle$. Then, Eq. (14) is equivalent to:

$$f_{ntk}(x) = (\langle \Phi(x), \Phi(x_1) \rangle, ..., \langle \Phi(x), \Phi(x_n) \rangle) \cdot \mathbf{K}_{train}^{-1} \mathbf{Y} \tag{15}$$

### B.2 IMPACT OF DATA GEOMETRY ON DEEP $Q$ FUNCTIONS

In this section, we analyze the impact of data geometry on deep $Q$ functions under the NTK regime. We first introduce the smoothness property of the feature map $\Phi(x)$ induced by NTK (Lemma 2). Then, we introduce the equivalence between the kernel regression solution in Eq. (15) and a min-norm solution (Lemma 3). Builds on Lemma 2 and Lemma 3, Lemma 4 analyzes the smoothness of the deep $Q$ functions. At last, we study how data geometry affects deep $Q$ functions (Theorem 1).

**Assumption 1.** *(NTK assumption). We assume the function approximators discussed in our paper are two-layer fully-connected ReLU neural networks with infinity width and are trained with infinitesimally small learning rate unless otherwise specified.*

Although there exist some gaps between the NTK assumption and the real setting, NTK is one of the most advanced theoretical machinery from the generalization analysis of DNN. In addition, Assumption 1 is common in previous analysis on the generalization of DNN (Jacot et al., 2018; Arora et al., 2019a; Bietti & Mairal, 2019) and the convergence of DRL (Cai et al., 2019; Liu et al., 2019; Xu & Gu, 2020; Fan et al., 2020). For more accurate analysis, we should adopt more advanced analysis tools than NTK and hence leave it for future work.

We first introduce Lemma 2 (Proposition 4 of (Bietti & Mairal, 2019)), which shows the feature map $\Phi(x)$ induced by NTK is not Lipschitz continuous but holds a weaker Hölder smoothness property.

**Lemma 2.** *(Smoothness of the kernel map of two-layer ReLU networks). Let $\Phi$ be the kernel map of the neural tangent kernel induced by a two-layer ReLU neural network, $x$ and $y$ be two inputs, then $\Phi$ satisfies the following smoothness property.*

$$\|\Phi(x) - \Phi(y)\| \leq \sqrt{\min(\|x\|, \|y\|)\|x - y\|} + 2\|x - y\|. \tag{16}$$

Lemma 3 (Lemma 2 of (Xu et al., 2020)) builds the connection between the kernel regression solution in Eq. (14) and the a min-norm solution. For the proof of Lemma 3, we refer the reader to(Xu et al., 2020).

**Lemma 3.** *(Equivalence to a min-norm optimization problem). Let $\Phi(x)$ be the feature map induced by a neural tangent kernel, for any $x \in \mathbb{R}^d$. The solution to the kernel regression in Eq. (14) and Eq. (15) is equivalent to $f_{ntk}(x) = \Phi(x)^T \beta_{ntk}$, where $\beta_{ntk}$ is the optimal solution of a min-norm optimization problem defined as*

$$\min_{\beta} \|\beta\|$$
$$\text{s.t. } \Phi(x_i)^T \beta = y_i, \text{ for } i = 1, ..., n. \tag{17}$$

Then, deep $Q$ functions satisfy the following smoothness property.

**Lemma 4.** *(Smoothness for deep Q functions). Given two inputs $x$ and $x'$, the distance between these two data points is $d = \|x - x'\|$. $C_1 := \sup \|\beta_{ntk}\|_{\infty}$ is a finite constant. Then the difference between the output at $x$ and the output at $x'$ can be bounded by:*

$$\|Q_\theta(x) - Q_\theta(x')\| \leq C_1(\sqrt{\min(\|x\|, \|x'\|)}\sqrt{d} + 2d) \tag{18}$$

*Proof.* In offline RL, we denote a general $Q$ network by $Q_\theta(x) : \mathbb{R}^{|\mathcal{S}|+|\mathcal{A}|} \to \mathbb{R}$, where $\theta$ is all the parameters in the network and $x = (s, a) \in \mathbb{R}^{|\mathcal{S}|+|\mathcal{A}|}$ is the brief notation for state-action pair $(s, a)$. The $Q$ function is trained via minimizing the temporal difference error defined as $\frac{1}{2}\sum_{i=1}^{n}(Q_\theta(x_i) - y_i)^2$ by gradient descent, where $y_i = r(x_i) + \gamma \mathbb{E}_{a'_i \sim \pi(\cdot|s'_i)}[Q^\pi_{\theta'}(x'_i)] \in \mathbb{R}$ is the target value.

Using kernel method from NTK, $Q$ function can be formulated as $Q_\theta(x) = \Phi(x)^T\beta$, where $\Phi(x)$ is independent of the changes on training labels when NTK assumption holds. This is because as the width of a neural net goes to infinity, the NTK kernel $\mathbf{K}(x, x') =< \Phi(x), \Phi(x') >$ produced by this network stays constant during training, and so is the property of the feature map $\Phi(x)$ (Jacot et al., 2018). So, the learning process under NTK framework is actually adjusting $\beta$ to fit the label rather than $\Phi(x)$. As a result, Lemma 2 holds when deep $Q$ function satisfies NTK assumptions. Given two inputs $x$ and $x'$, the distance between these two inputs is $d = \|x - x'\|$. Based on Lemma 2, it is easy to see that

$$\begin{aligned}
\|Q_\theta(x) - Q_\theta(x')\| &= \|\Phi(x)^T\beta - \Phi(x')^T\beta\| \\
&\leq \|\Phi(x) - \Phi(x')\|\|\beta\|_\infty \quad \text{(Infinity norm)} \\
&\leq \|\beta\|_\infty(\sqrt{\min(\|x\|, \|x'\|) \cdot \|x - x'\|} + 2\|x - x'\|) \text{ (Lemma2)} \\
&= \|\beta\|_\infty(\sqrt{\min(\|x\|, \|x'\|) \cdot d} + 2d) \\
&\leq C_\beta(\sqrt{\min(\|x\|, \|x'\|) \cdot d} + 2d) \quad (C_\beta := \sup \|\beta\|_\infty)
\end{aligned} \tag{19}$$

Additionally, if we consider the delayed $Q$ target and delayed actor updates during policy learning, we can assume the target value used for $Q$ evaluation stays relatively stable during each policy evaluation step and the problem can be seen as solving a series of regression problems. Under this mild assumption, we can learn the actual $\beta_{ntk}$ at each step ($\beta \to \beta_{ntk}$ and so $C_\beta \to C_1$, where $C_1 := \sup \|\beta_{ntk}\|_\infty$) and thus complete the proof. Similar assumptions and treatments are also used

in Section 4 of (Kumar et al., 2020a) that Q function at each iteration can fit its label well, Appendix A.8 of (Xiao et al., 2021), as well as Appendix F of (Ghasemipour et al.).

□

Lemma 4 states the value difference of a deep $Q$ function for two inputs is related to the distance between these two inputs. The closer the distance, the smaller the value difference.

### B.2.1   PROOF OF THEOREM 1

Builds on Lemma 4, we can combine the data geometry and analyze the impact of data geometry on deep $Q$ functions.

*Proof.* We first review the definition of interpolated data and extrapolated data. Under continuous state-action space, state-action pairs within the convex hull of the dataset can be represented in an interpolated manner (referred as interpolated data $x_{in}$):

$$x_{in} = \sum_{i=1}^{n} \alpha_i x_i, \quad \sum_{i=1}^{n} \alpha_i = 1, \alpha_i \geq 0 \tag{20}$$

Similarly, we can define extrapolated data that lie outside the convex hull of the dataset as $x_{out}$:

$$x_{out} = \sum_{i=1}^{n} \beta_i x_i, \tag{21}$$

where $\sum_{i=1}^{n} \beta_i = 1$ and $\beta_i \geq 0$ does not hold simultaneously.

We define $\mathrm{Proj}_{\mathcal{D}}(x) := \arg\min_{x_i \in \mathcal{D}} \|x - x_i\|$ as a projector that projects unseen data $x$ to its nearest data in dataset $\mathcal{D}$. Given an interpolated data $x_{in}$ and an extrapolated data $x_{out}$, the distances to their nearest data in dataset are $d_{x_{in}} = \|x_{in} - \mathrm{Proj}_{\mathcal{D}}(x_{in})\|$ and $d_{x_{out}} = \|x_{out} - \mathrm{Proj}_{\mathcal{D}}(x_{out})\|$. Because interpolated data lie inside the convex hull of training data, $d_{x_{in}} \leq \max_{x_i \in \mathcal{D}} \|x_{in} - x_i\| \leq B$ is bounded, where $B := \max_{x_i, x_j \in \mathcal{D}} \|x_i - x_j\|$ is a finite constant. Then, by applying Lemma 4, the value difference of deep $Q$ function for interpolated and extrapolated data can be formulated as the following shows.

$$\begin{aligned} \|Q_\theta(x_{in}) - Q_\theta(\mathrm{Proj}_{\mathcal{D}}(x_{in}))\| &\leq C_1(\sqrt{\min(\|x_{in}\|, \|\mathrm{Proj}_{\mathcal{D}}(x_{in})\|)}\sqrt{d_{x_{in}}} + 2d_{x_{in}}) \\ &\leq C_1(\sqrt{\min(\|x_{in}\|, \|\mathrm{Proj}_{\mathcal{D}}(x_{in})\|)}\sqrt{B} + 2B) \end{aligned} \tag{22}$$

$$\|Q_\theta(x_{out}) - Q_\theta(\mathrm{Proj}_{\mathcal{D}}(x_{out}))\| \leq C_1(\sqrt{\min(\|x_{out}\|, \|\mathrm{Proj}_{\mathcal{D}}(x_{out})\|)}\sqrt{d_{x_{out}}} + 2d_{x_{out}}) \tag{23}$$

□

### B.3   QUANTITATIVE EXPERIMENTS ON THEOREM 1

In addition to the one-dimensional random walk experiments presented in Section 2.2, we conduct additional experiments on the more complex and high-dimensional MuJoCo tasks (including D4RL Hopper-medium-v2, Halfcheetah-medium-v2, and Walker2d-medium-v2) to provide quantitative support to Theorem 1, in particular, the pertinence of interpolation and extrapolation. We first synthesize lots of interpolated data $x_{in}$ and extrapolated data $x_{out}$ ($x = (s, a) \in \mathcal{S} \times \mathcal{A}$) and then search for their nearest data points in offline dataset $\mathcal{D}$ accordingly, i.e., $\mathrm{Proj}_{\mathcal{D}}(x_{in})$ and $\mathrm{Proj}_{\mathcal{D}}(x_{out})$. Then, we can evaluate the Q-value differences $\|Q_\theta(x) - Q_\theta(\mathrm{Proj}_{\mathcal{D}}(x))\|$ (LHS of Theorem 1) at these generated data and see whether the Q-value differences align well with Theorem 1.

For the detailed experiment setup, recall that an interpolated data point $x_{in}$ is a convex combination of the offline dataset, i.e., $x_{in} = \sum_{i=1}^{n} \alpha_i x_i, x_i \sim \mathcal{D}$ with weights $\alpha_i$ that satisfy $\sum_{i=1}^{n} \alpha_i = 1, \alpha_i \geq 0$. Therefore, we can interpolate the offline dataset based on $\alpha_i$ sampled from the Dirichlet distribution to generate the interpolated data. Also, an extrapolated data point $x_{out}$ is expressed as a weighted sum of the offline dataset, i.e., $x_{out} = \sum_{i=1}^{n} \beta_i x_i, x_i \sim \mathcal{D}$, but its weights $\beta_i$ do not satisfy the non-negativity and the summing to 1 constraint. Therefore, we can generate extrapolated data by setting the sign of some weights to negative values and varying the weights not summing to

1. After obtaining the interpolated and extrapolated data, we search for their closest data points in the offline dataset $\mathcal{D}$ and calculate their corresponding distance $\|x - \text{Proj}_{\mathcal{D}}(x)\|$ and Q-value difference$\|Q_\theta(x) - Q_\theta(\text{Proj}_{\mathcal{D}}(x))\|$. Figure 7a shows the relationship between the distance to dataset $\|x - \text{Proj}_{\mathcal{D}}(x)\|$ and the Q value difference $\|Q_\theta(x) - Q_\theta(\text{Proj}_{\mathcal{D}}(x))\|$ (LHS of Theorem 1). We also report the learned state-conditioned distance value $g(s, a)$ on these generated data in Figure 7b.

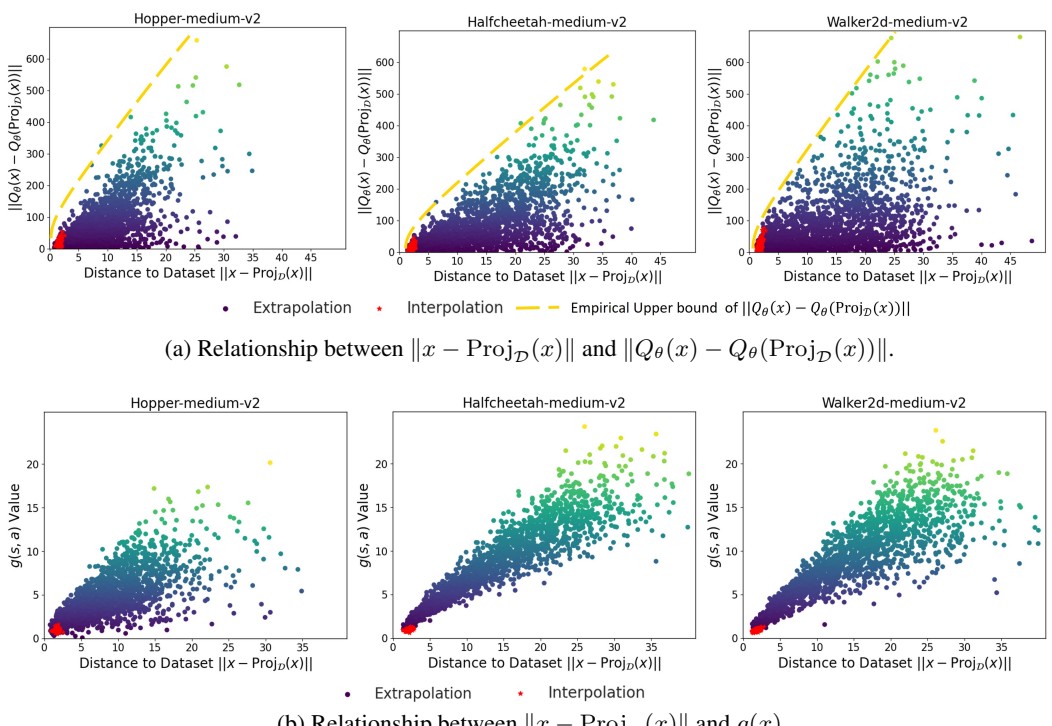

(a) Relationship between $\|x - \text{Proj}_{\mathcal{D}}(x)\|$ and $\|Q_\theta(x) - Q_\theta(\text{Proj}_{\mathcal{D}}(x))\|$.

(b) Relationship between $\|x - \text{Proj}_{\mathcal{D}}(x)\|$ and $g(x)$

Figure 7: Quantitative experiments of Theorem 1 on the D4RL MuJoCo-medium datasets. The red star-shaped dots are the interpolated data and the circle dots are the extrapolated data. The color of the dots represents $\|Q_\theta(x) - Q_\theta(\text{Proj}_{\mathcal{D}}(x))\|$ values in (a) and $g(x)$ values in (b), respectively. The darker the color, the smaller the corresponding value. In (a), the yellow dash line is the empirical upper bound of $\|Q_\theta(x) - Q_\theta(\text{Proj}_{\mathcal{D}}(x))\|$.

Figure 7a demonstrates that the interpolated data enjoy a tighter empirical upper bound of $\|Q_\theta(x) - Q_\theta(\text{Proj}_{\mathcal{D}}(x))\|$ (LHS of Theorem 1) than most of the extrapolated data. Moreover, the empirical upper bound of the Q-value difference grows with the increase of the sample-to-dataset distance $\|x - \text{Proj}_{\mathcal{D}(x)}\|$, which is consistent with Theorem 1 (the upper bound of value difference of deep Q function is well controlled by distance to the dataset). Figure 7b shows that the state-conditioned distance function $g(s, a)$ can output low values for interpolated data and some near-dataset extrapolated data, and thus can be used as a relaxed policy constraint in these OOD regions.

## C  STATE-CONDITIONED DISTANCE FUNCTION

### C.1  PROOF OF PROPERTY 1

*Proof.* Given a state-action pair from the training data $(s, a) \sim \mathcal{D}$, we synthetic random noise actions from a uniform distribution over the action space, *i.e.* $\hat{a} \sim Unif(\mathcal{A})$. Then the distance function $g(\cdot)$ is trained by Eq. (24).

$$\min_g \mathbb{E}_{(s,a)\sim\mathcal{D}} \left[ \mathbb{E}_{\hat{a}\sim Unif(\mathcal{A})} \left[ [\|\hat{a} - a\| - g(s, \hat{a})]^2 \right] \right] \tag{24}$$

$[\|\hat{a} - a\| - g(s, \hat{a})]^2$ can be upper bounded by some finite constants because $\mathcal{S} \times \mathcal{A}$ is compact in our analysis. The optimization problem in Eq. (24) can be reformulated as the following form according to the Fubini's Theorem.

$$\min_{g} \mathbb{E}_{\hat{a} \sim Unif(\mathcal{A})} \left[ \mathbb{E}_{(s,a) \sim \mathcal{D}} \left[ \|\hat{a} - a\| - g(s, \hat{a}) \right]^2 \right] \tag{25}$$

Note that the objective of Eq. (25) can be also written as a functional $J[g(s, \hat{a})]$ with respect to function $g$ in following form:

$$J[g(s, \hat{a})] = \int_{\mathcal{A}} \frac{1}{|\mathcal{A}|} \left[ \mathbb{E}_{(s,a) \sim \mathcal{D}} [\|\hat{a} - a\| - g(s, \hat{a})]^2 \right] d\hat{a} = \int_{\mathcal{A}} F(s, \hat{a}, g(s, \hat{a})) d\hat{a} \tag{26}$$

Based on calculus of variation, the extrema (maxima or minima) of functional $J[g(s, \hat{a})]$ can be obtained by solving the associated Euler-Langrane equation ($\partial F / \partial g = 0$). In our case, it requires the optimal state-conditioned distance function $g^*$ satisfies the following conditions:

$$\frac{\partial}{\partial g^*} \mathbb{E}_{(s,a) \sim \mathcal{D}} [\|\hat{a} - a\| - g^*(s, \hat{a})]^2 = 0$$

$$\Rightarrow \quad \mathbb{E}_{(s,a) \sim \mathcal{D}} \left[ \frac{\partial}{\partial g^*} [\|\hat{a} - a\| - g^*(s, \hat{a})]^2 \right] = 0 \quad \text{(DNN is continuous)} \tag{27}$$

$$\Rightarrow \quad \mathbb{E}_{(s,a) \sim \mathcal{D}} [\|\hat{a} - a\| - g^*(s, \hat{a})]] = 0$$

Conditioned on a state $s \in \mathcal{D}$, the optimal state-conditioned distance function in Eq. (27) satisfies the following conditions:

$$\int_{\mathcal{A}} \|\hat{a} - a\| \mu(s, a) da - \int_{\mathcal{A}} \mu(s, a) da g^*(s, \hat{a}) = 0, s \in \mathcal{D}$$

$$\Rightarrow g^*(s, \hat{a}) = \frac{\int_{\mathcal{A}} \|\hat{a} - a\| \mu(s, a) da}{\int_{\mathcal{A}} \mu(s, a) da}, s \in \mathcal{D} \tag{28}$$

$$\Rightarrow g^*(s, \hat{a}) = \int_{\mathcal{A}} C(s, a) \|\hat{a} - a\| da, s \in \mathcal{D}$$

where, $\mu(s, a)$ is the empirical distribution on a finite offline dataset $\mathcal{D} = \{(x_i)\}_{i=1}^n$, *i.e.*, the sum of the Dirac measures $\frac{1}{n} \sum_{i=1}^n \delta_{x_i}$. $\forall (s, a) \notin \mathcal{D}, \mu(s, a) = 0. \forall (s, a) \in \mathcal{D}, \mu(s, a) > 0$. $C(s, a) = \frac{\mu(s,a)}{\int_{\mathcal{A}} \mu(s,a) da} \geq 0$ and $\int_{\mathcal{A}} C(s, a) da = 1$. Because $L_2$-norm is convex and the non-negative combination of convex functions is still convex, $g^*(s, \hat{a})$ is a convex function *w.r.t.* $\hat{a}$. In addition, $\forall \hat{a} \in \mathcal{A}$, by the Jensen inequality, we have:

$$g^*(s, \hat{a}) \geq \left\| \hat{a} - \mathbb{E}_{a \sim Unif(\mathcal{A})} [C(s, a)a] \right\| = \|\hat{a} - a_o(s)\|, s \in \mathcal{D} \tag{29}$$

where $a_o(s) := \mathbb{E}_{a \sim Unif(\mathcal{A})} [C(s, a)a], s \in \mathcal{D}$ is the state-conditioned centroid of training dataset.

$\square$

## C.2 PROOF OF PROPERTY 2

*Proof.* The negative gradient of the optimal state-conditioned distance function can be formulated as:

$$-\nabla_{\hat{a}} g^*(s, \hat{a}) = -\int_{\mathcal{A}} C(s, a) \frac{\hat{a} - a}{\|\hat{a} - a\|} da, \forall \hat{a} \in \mathcal{A}, s \in \mathcal{D}$$

$$= \frac{1}{\int_{\mathcal{A}} \mu(s, a) da} \int_{\mathcal{A}} \mu(s, a) \frac{-(\hat{a} - a)}{\|\hat{a} - a\|} da, \forall \hat{a} \in \mathcal{A}, s \in \mathcal{D} \tag{30}$$

Observe that the direction of the negative gradient of $g^*(s, \hat{a})$ is related to the integral of vector $-(\hat{a} - a)$ (points towards $a$). When $(s, a) \notin \mathcal{D}$, $-(\hat{a} - a)$ doesn't influence the final gradient because

$\mu(s, a) = 0$. Therefore, $-(\hat{a} - a)$ only contribute to the final gradient of $g^*(s, \hat{a})$ for $(s, a) \in \mathcal{D}$ as $\mu(s, a) > 0$. For a given $s \in \mathcal{D}$ and any extrapolated action $\hat{a}$ that lies outside the convex hull of training data, the integral of vector $-(\hat{a} - a)$ is basically a non-negative combination of vectors $-(\hat{a} - a)$ that point toward actions $a \in \mathcal{D}$ inside the convex hull. As a result, it's easy to see that $-\nabla_{\hat{a}} g^*(s, \hat{a})$ also points inside the convex hull formed by the data.

$\square$

# D   THEORETICAL ANALYSIS OF DOGE

In this section, we analyze the performance of the policy learned by DOGE. We first adopt the Bellman-consistent coefficient from (Xie et al., 2021a) to quantify the distributional shift from the perspective of deep $Q$ functions generalization. Then, we gives the upper bound of the Bellman-consistent coefficient under the NTK regime (Appendix D.1). At last, we give the performance bound of DOGE (Appendix D.2).

## D.1   UPPER BOUND OF BELLMAN-CONSISTENT COEFFICIENT

Let us first review the definition of Bellman-consistent coefficient $\mathcal{B}(v, \mu, \mathcal{F}, \pi)$ in (Xie et al., 2021a). We define $\mathcal{B}(v, \mu, \mathcal{F}, \pi)$ to measure the distributional shift from an arbitrary distribution $v$ to data distribution $\mu$, *w.r.t.* $\mathcal{F}$ and $\pi$. $\mathcal{F}$ is the function class of $Q$ networks.

$$\mathcal{B}(v, \mu, \mathcal{F}, \pi) := \sup_{Q \in \mathcal{F}} \frac{\|Q - \mathcal{T}^\pi Q\|_{2,v}^2}{\|Q - \mathcal{T}^\pi Q\|_{2,\mu}^2} \tag{31}$$

where the $\mu$-weighted norm (square) is defined as $\|f\|_{2,\mu}^2 := \mathbb{E}_\mu[\|f\|^2]$, which is also applicable for any distribution $v$. $\mathcal{T}^\pi Q$ is the Bellman operator of policy $\pi$, defined as $\mathcal{T}^\pi Q(s, a) := r(s, a) + \gamma \mathbb{E}_{a' \sim \pi(\cdot|s'), s' \sim \mathcal{P}(\cdot|s,a)}[Q(s', a')] := r(s, a) + \gamma \mathbb{P}^\pi[Q(s', a')]$. $\mathbb{P}^\pi[\cdot]$ is the brief notation for $\mathbb{E}_{a' \sim \pi(\cdot|s'), s' \sim \mathcal{P}(\cdot|s,a)}[\cdot]$. The smaller the ratio of the Bellman error under $v$ and $\mu$, the more transferable the $Q$ function from $\mu$ to $v$, even when $\sup_{(s,a)} \frac{v(s,a)}{\mu(s,a)} = \infty$. Then we give the proof of Theorem 2 (Upper bound of Bellman-consistent coefficient).

*Proof.* We denote $x = (s, a)$ and $x' = (s', a')$. $x_o = \mathbb{E}_{x \sim \mathcal{D}}[x]$ is the centroid of offline dataset. $d_1 = \|x - x_o\|$ and $d_2 = \|x' - x_o\|$ are the sample-to-centroid distances. Let $\mu(x)$ be the distribution under the offline dataset and $v(x)$ be any distribution. Then, for the numerator in Eq. (8) and Eq. (31), we have the following inequalities.

$$\|Q - \mathcal{T}^\pi Q\|_{2,v}^2$$
$$= \int_{\mathcal{S} \times \mathcal{A}} v(x) \|Q(x) - r(x) - \gamma \mathbb{P}^\pi[Q(x')]\|^2$$
$$= \int_{\mathcal{S} \times \mathcal{A}} v(x) \|Q(x) - \mathbb{P}^\pi[Q(x')] - r(x) + (1 - \gamma) \mathbb{P}^\pi[Q(x')]\|^2$$
$$\leq \int_{\mathcal{S} \times \mathcal{A}} v(x) \left[ \|Q(x) - \mathbb{P}^\pi[Q(x')]\| + \|r(x)\| + \|(1 - \gamma) \mathbb{P}^\pi[Q(x')]\| \right]^2 \text{ (Triangle)}$$
$$= \int_{\mathcal{S} \times \mathcal{A}} v(x) \left[ \|Q(x) - Q(x_o) + Q(x_o) - \mathbb{P}^\pi[Q(x')]\| + \|r(x)\| + (1 - \gamma)\|\mathbb{P}^\pi[Q(x')] - Q(x_o) + Q(x_o)\| \right]^2$$
$$\leq \int_{\mathcal{S} \times \mathcal{A}} v(x) \left[ (1 - \gamma)\|Q(x_o)\| + \|r(x)\| + \|Q(x) - Q(x_o)\| + (2 - \gamma)\|\mathbb{P}^\pi[Q(x')] - Q(x_o)\| \right]^2 \text{ (Triangle)}$$
$$\leq \int_{\mathcal{S} \times \mathcal{A}} v(x) \left[ \underbrace{(1 - \gamma)\|Q(x_o)\| + \|r(x)\|}_{\mathcal{I}_1} + \underbrace{\|Q(x) - Q(x_o)\|}_{\mathcal{I}_2} + \underbrace{(2 - \gamma)\mathbb{P}^\pi[\|Q(x') - Q(x_o)\|]}_{\mathcal{I}_3} \right]^2 \text{ (Jensen)}$$

$$\tag{32}$$

The RHS contains three parts: $\mathcal{I}_1 = (1 - \gamma)\|Q(x_o)\| + \|r(x)\|$, $\mathcal{I}_2 = \|Q(x) - Q(x_o)\|$ and $\mathcal{I}_3 = (2 - \gamma)\mathbb{P}^\pi[\|Q(x') - Q(x_o)\|]$. Because $\|r(x)\| \in [0, R_{\max}], \forall x \in \mathcal{S} \times \mathcal{A}$, $\mathcal{I}_1$ can be upper

bounded as:

$$\mathcal{I}_1 \leq (1-\gamma)Q(x_o) + R_{\max} \tag{33}$$

By applying Lemma 4, $\mathcal{I}_2$ is upper bounded as

$$\mathcal{I}_2 \leq C_1 \left[ \sqrt{\min(\|x\|, \|x_o\|)d_1} + 2d_1 \right] \tag{34}$$

$\mathcal{I}_3$ is upper bounded as

$$\mathcal{I}_3 \leq C_1(2-\gamma)\mathbb{P}^\pi \left[ \sqrt{\min(\|x'\|, \|x_o\|)d_2} + 2d_2 \right] \tag{35}$$

In addition, we denote $C_2 := \sqrt{\sup_{x \in \mathcal{S} \times \mathcal{A}} \|x\|}$. Then, $\mathcal{I}_2$ and $\mathcal{I}_3$ can be further upper bounded by

$$\mathcal{I}_2 \leq C_1 \left( C_2\sqrt{d_1} + 2d_1 \right) \tag{36}$$

$$\mathcal{I}_3 \leq (2-\gamma)C_1\mathbb{P}^\pi(C_2\sqrt{d_2} + 2d_2) \tag{37}$$

The above relaxation of the upper bound in Eq. (36) and Eq. (37) is not necessary, but for notation brevity, we choose to relax the upper bound by treating $C_2 := \sqrt{\sup_{x \in \mathcal{S} \times \mathcal{A}} \|x\|}$.

Plug Eq. (33), Eq. (36) and Eq. (37) into the RHS of Eq. (32), we can get

$$\begin{aligned}
&\|Q - \mathcal{T}^\pi Q\|_{2,v}^2 \\
&\leq \int_{\mathcal{S} \times \mathcal{A}} v(x) \left[ (1-\gamma)Q(x_o) + R_{\max} + C_1(C_2\sqrt{d_1} + 2d_1) + (2-\gamma)C_1\mathbb{P}^\pi(C_2\sqrt{d_2} + 2d_2) \right]^2 \\
&= \left\| (1-\gamma)Q(s_o, a_o) + R_{\max} + C_1 \left( C_2\sqrt{d_1} + 2d_1 \right) + (2-\gamma)C_1\mathbb{P}^\pi \left( C_2\sqrt{d_2} + 2d_2 \right) \right\|_{2,v}^2
\end{aligned} \tag{38}$$

For the denominator $\|Q - \mathcal{T}^\pi Q\|_{2,\mu}^2$ in Eq. (8) and Eq. (31), because the $Q$ function is approximated, there exists approximation error between $Q$ and $\mathcal{T}^\pi Q$, i.e., $Q - \mathcal{T}^\pi Q \geq \epsilon$. In addition, the distribution $\mu$ contains some mismatch w.r.t. the equilibrium distribution induced by policy $\pi$. Therefore, it is reasonable to assume $\|Q - \mathcal{T}^\pi Q\|_{2,\mu}^2 \geq \epsilon_\mu > 0$.

Then, we can complete the proof by plugging the upper bound in Eq. (38) and $\|Q - \mathcal{T}^\pi Q\|_{2,\mu}^2 \geq \epsilon_\mu > 0$ into Eq. (8) or Eq. (31).

$$\mathcal{B}(v, \mu, \mathcal{F}, \pi) \leq \frac{1}{\epsilon_\mu} \left\| \underbrace{(1-\gamma)Q(s_o, a_o) + R_{\max}}_{\mathcal{B}_1} + \underbrace{C_1 \left( C_2\sqrt{d_1} + 2d_1 \right)}_{\mathcal{B}_2} + \underbrace{(2-\gamma)C_1\mathbb{P}^\pi \left( C_2\sqrt{d_2} + 2d_2 \right)}_{\mathcal{B}_3} \right\|_{2,v}^2 \tag{39}$$

$\square$

To be mentioned, the distance regularization in DOGE compels the leaned policy to output the action that near the state-conditioned centroid of dataset and thus $\mathcal{B}_2$ and $\mathcal{B}_3$ can be driven to some small values. $\mathcal{B}_1$ is independent on the distributional shift. Therefore, $\mathcal{B}(v, \mu, \mathcal{F}, \pi)$ can be bounded by some finite constants under DOGE. Therefore, the constrained policy set induced by DOGE is essentially a Bellman-consistent constrained policy set $\Pi_\mathcal{B}$ defined in Definition 2. In addition, other policy constraint methods such as BEAR (Kumar et al., 2019) can also have bounded $\mathcal{B}$. However, these policy constraint methods do not allow the learned policy shifts to those generalizable distributions where $\mathcal{B}(v, \mu, \mathcal{F}, \pi)$ is small but $\sup_{(s,a)} \frac{v(s,a)}{\mu(s,a)} \to \infty$, which is essentially different with DOGE.

## D.2 Performance of the Policy learned by DOGE

Here, we briefly review the definition of the Bellman-consistent constrained policy set $\Pi_{\mathcal{B}}$ defined in Definition 2. The Bellman-consistent coefficient under the transition induced by $\Pi_{\mathcal{B}}$ can be bounded by some finite constants $l(k)$:

$$\mathcal{B}(\rho_k, \mu, \mathcal{F}, \pi) \leq l(k) \tag{40}$$

where, $\rho_0$ is the initial state-action distribution and $\mu$ is the distribution of training data. $\rho_k = \rho_0 P^{\pi_1} P^{\pi_2} ... P^{\pi_k}, \forall \pi_1, \pi_2, ..., \pi_k \in \Pi_{\mathcal{B}}$ and $P^{\pi_i}$ is the transition operator on states induced by $\pi_i$, i.e., $P^{\pi_i}(s', a'|s, a) = \mathcal{P}(s'|s, a)\pi_i(a'|s')$.

We denote the constrained Bellman operator induced by $\Pi_{\mathcal{B}}$ as $\mathcal{T}^{\Pi_{\mathcal{B}}}$, and $\mathcal{T}^{\Pi_{\mathcal{B}}}Q(s, a) := r(s, a) + \max_{\pi \in \Pi_{\mathcal{B}}} \gamma \mathbb{P}^\pi[Q(s', a')]$. $\mathcal{T}^{\Pi_{\mathcal{B}}}$ can be seen as a operator in a redefined MDP and hence is a contraction mapping and exists a fixed point. We denote $Q^{\Pi_{\mathcal{B}}}$ as the fixed point of $\mathcal{T}^{\Pi_{\mathcal{B}}}$, i.e., $Q^{\Pi_{\mathcal{B}}} = \mathcal{T}^{\Pi_{\mathcal{B}}}Q^{\Pi_{\mathcal{B}}}$.

The Bellman optimal operator $\mathcal{T}$ is

$$\mathcal{T}Q(s, a) := r(s, a) + \max_\pi \gamma \mathbb{P}^\pi[Q(s', a')] \tag{41}$$

$\mathcal{T}$ is also a contraction mapping. Its fixed point is the optimal value function $Q^*$ and $Q^* = \mathcal{T}Q^*$.

Then, by the triangle inequality, we have:

$$\begin{aligned}
\|Q^* - Q^{\pi_n}\|_{\rho_0} &= \|Q^* - Q^{\Pi_{\mathcal{B}}} + Q^{\Pi_{\mathcal{B}}} - Q^{\pi_n}\|_{\rho_0} \\
&\leq \underbrace{\|Q^* - Q^{\Pi_{\mathcal{B}}}\|_{\rho_0}}_{L_1} + \underbrace{\|Q^{\Pi_{\mathcal{B}}} - Q^{\pi_n}\|_{\rho_0}}_{L_2}
\end{aligned} \tag{42}$$

where $Q^{\pi_n}$ is the true $Q$ value of policy $\pi_n$. $\pi_n$ is the greedy policy w.r.t. to $Q_n$ in the Bellman-consistent constrained policy set $\Pi_{\mathcal{B}}$, i.e., $\pi_n = \sup_{\pi \in \Pi_{\mathcal{B}}} \mathbb{E}_{a \sim \pi(\cdot|s)}[Q_n(s, a)]$. $Q_n$ is the $Q$ function after $n$-th value iteration under the constrained Bellman operator $\mathcal{T}^{\Pi_{\mathcal{B}}}$.

For $L_1$ part in Eq. (42), we first focus on the infinity norm.

$$\begin{aligned}
\|Q^* - Q^{\Pi_{\mathcal{B}}}\|_\infty &= \|\mathcal{T}Q^* - \mathcal{T}^{\Pi_{\mathcal{B}}}Q^{\Pi_{\mathcal{B}}}\|_\infty \\
&\leq \|\mathcal{T}Q^* - \mathcal{T}^{\Pi_{\mathcal{B}}}Q^{\Pi_{\mathcal{B}}}\|_\infty + \|\mathcal{T}^{\Pi_{\mathcal{B}}}Q^{\Pi_{\mathcal{B}}} - \mathcal{T}^{\Pi_{\mathcal{B}}}Q^*\|_\infty \\
&\leq \|\mathcal{T}Q^* - \mathcal{T}^{\Pi_{\mathcal{B}}}Q^{\Pi_{\mathcal{B}}}\|_\infty + \gamma\|Q^* - Q^{\Pi_{\mathcal{B}}}\|_\infty \quad (\mathcal{T}^{\Pi_{\mathcal{B}}} \text{ is } \gamma-\text{contraction}) \\
&= \alpha(\Pi_{\mathcal{B}}) + \gamma\|Q^* - Q^{\Pi_{\mathcal{B}}}\|_\infty
\end{aligned} \tag{43}$$

where $\alpha(\Pi_{\mathcal{B}}) := \|\mathcal{T}Q^* - \mathcal{T}^{\Pi_{\mathcal{B}}}Q^{\Pi_{\mathcal{B}}}\|_\infty$ is the suboptimality constant. Then, we get $\|Q^* - Q^{\Pi_{\mathcal{B}}}\|_\infty \leq \frac{\alpha(\Pi_{\mathcal{B}})}{1-\gamma}$ and $L_1 \leq \|Q^* - Q^{\Pi_{\mathcal{B}}}\|_\infty \leq \frac{\alpha(\Pi_{\mathcal{B}})}{1-\gamma}$.

For $L_2$, we introduce Lemma 5, which upper bounds $\|Q^{\Pi_{\mathcal{B}}} - Q^{\pi_n}\|_{2,\rho_0}^2$. The proof of Lemma 5 can be get by directly replacing $Q^*$ with $Q^{\Pi_{\mathcal{B}}}$ in the Appendix F.3. in (Le et al., 2019), because $Q^{\Pi_{\mathcal{B}}}$ is the optimal value function under the modified MDP induced by $\mathcal{T}^{\Pi_{\mathcal{B}}}$.

**Lemma 5.** *(Upper bound of error propagation).* $\|Q^{\Pi_{\mathcal{B}}} - Q^{\pi_n}\|_{2,\rho_0}^2$ *can be upper bounded as*

$$\|Q^{\Pi_{\mathcal{B}}} - Q^{\pi_n}\|_{2,\rho_0}^2 \leq \left[\frac{2\gamma(1-\gamma^{n+1})}{(1-\gamma)^2}\right]^2 \int_{\mathcal{S}\times\mathcal{A}} \rho_0(ds, da) \left[\sum_{k=0}^{n-1} \alpha_k A_k \epsilon_k^2 + \alpha_n A_n (Q^{\Pi_{\mathcal{B}}} - Q_0)^2\right](s, a) \tag{44}$$

*where*

$$\epsilon_k = Q_{k+1} - \mathcal{T}^{\Pi_{\mathcal{B}}}Q_k \tag{45}$$

$$\begin{aligned}
\alpha_k &= \frac{(1-\gamma)\gamma^{n-k-1}}{1-\gamma^{n+1}} \quad \text{for } k < n \\
\alpha_n &= \frac{(1-\gamma)\gamma^n}{1-\gamma^{n+1}}
\end{aligned} \tag{46}$$

$$A_k = \frac{1-\gamma}{2} \sum_{m \geq 0} \gamma^m (P^{\pi_n})^m \left[ (P^{\pi^{\Pi_{\mathcal{B}}}})^{n-k} + P^{\pi_n} P^{\pi_{n-1}} ... P^{\pi_{k+1}} \right] \quad \text{for } k < n$$

$$A_n = \frac{1-\gamma}{2} \sum_{m \geq 0} \gamma^m (P^{\pi_n})^m \left[ (P^{\pi^{\Pi_{\mathcal{B}}}})^{n+1} + P^{\pi_n} P^{\pi_{n-1}} ... P^{\pi_0} \right]$$

(47)

$Q_0$ is the $Q$ function after initialization. Note that $\lim_{n \to \infty} \left[ \alpha_n A_n (Q^{\Pi_{\mathcal{B}}} - Q_0)^2 \right] = 0$, we leave out this term for analysis simplicity. In addition, each $A_k$ is a probability kernel that combine $P^{\pi_i}$ and $P^{\pi^{\Pi_{\mathcal{B}}}}$ (the transition operator on states induced by the constrained optimal policy $\pi^{\Pi_{\mathcal{B}}} \in \Pi_{\mathcal{B}}$) and $\sum_k a_k = 1$.

The key part in Eq. (44) is $\int_{\mathcal{S} \times \mathcal{A}} \rho_0 A_k \epsilon_k^2$ and we expand this term as the following shows.

$$\int_{\mathcal{S} \times \mathcal{A}} \rho_0 A_k \epsilon_k^2 = \int_{\mathcal{S} \times \mathcal{A}} \frac{1-\gamma}{2} \rho_0 \sum_{m \geq 0} \gamma^m (P^{\pi_n})^m \left[ (P^{\pi^{\Pi_{\mathcal{B}}}})^{n-k} + P^{\pi_n} P^{\pi_{n-1}} ... P^{\pi_{k+1}} \right] \epsilon_k^2$$

$$= \frac{1-\gamma}{2} \sum_{m \geq 0} \gamma^m \int_{\mathcal{S} \times \mathcal{A}} \left[ (P^{\pi_n})^m (P^{\pi^{\Pi_{\mathcal{B}}}})^{n-k} + (P^{\pi_n})^m P^{\pi_n} P^{\pi_{n-1}} ... P^{\pi_{k+1}} \right] \rho_0 \epsilon_k^2$$

(48)

As Eq. (40) shows, the policy set induced by DOGE is a Bellman-consistent constrained policy set $\Pi_{\mathcal{B}}$ defined in Definition 2. Therefore, let $\rho_0$ be the initial state-action distribution and $\mu$ denote the distribution of training data. For any policy $\pi_1, \pi_2, ..., \pi_k \in \Pi_{\mathcal{B}}$, the distribution after $k$-th Bellman-consistent iteration is $\rho_k = \rho_0 P^{\pi_1} P^{\pi_2} ... P^{\pi_k}$, there exits some finite constants $l(k)$, that $\mathcal{B}(\rho_k, \mu, \mathcal{F}, \pi) \leq l(k)$ holds. Then we can get the following inequalities.

$$\|Q - \mathcal{T}^\pi Q\|_{2,\rho_k}^2 \leq \|Q - \mathcal{T}^\pi Q\|_{2,\mu}^2 l(k)$$

$$\int_{\mathcal{S} \times \mathcal{A}} \rho_k \epsilon^2 \leq \int_{\mathcal{S} \times \mathcal{A}} \mu \epsilon^2 l(k) \quad (\epsilon = Q - \mathcal{T}^\pi Q)$$

(49)

As a result, by applying the result of Eq. (49) to Eq. (48), we can get

$$\int_{\mathcal{S} \times \mathcal{A}} \rho_0 A_k \epsilon_k^2 \leq \int_{\mathcal{S} \times \mathcal{A}} (1-\gamma) \sum_{m \geq 0} \gamma^m \epsilon_k^2 \mu l(m+n-k)$$

(50)

Plugs Eq. (50) into Eq. (44) and leaves out $\left[ \alpha_n A_n (Q^{\Pi_{\mathcal{B}}} - Q_0)^2 \right]$ in Eq. (44), we get

$$\lim_{n \to \infty} L_2^2 \leq \lim_{n \to \infty} \left[ \frac{2\gamma(1-\gamma^{n+1})}{(1-\gamma)^2} \right]^2 \left[ \sum_{k=0}^{n-1} (1-\gamma) \sum_{m \geq 0} \gamma^m l(m+n-k) \alpha_k \|\epsilon_k\|_{2,\mu}^2 \right]$$

$$= \lim_{n \to \infty} \left[ \frac{2\gamma(1-\gamma^{n+1})}{(1-\gamma)^2} \right]^2 \left[ \frac{1}{1-\gamma^{n+1}} \sum_{k=0}^{n-1} (1-\gamma)^2 \sum_{m \geq 0} \gamma^{m+n-k-1} l(m+n-k) \|\epsilon_k\|_{2,\mu}^2 \right]$$

$$\leq \lim_{n \to \infty} \left[ \frac{2\gamma(1-\gamma^{n+1})}{(1-\gamma)^2} \right]^2 \left[ \frac{1}{1-\gamma^{n+1}} L(\Pi_{\mathcal{B}})^2 \sup_{k \geq 0} \|\epsilon_k\|_{2,\mu}^2 \right]$$

$$= \left[ \frac{2\gamma}{(1-\gamma)^2} \right]^2 L(\Pi_{\mathcal{B}})^2 \sup_{k \geq 0} \|\epsilon_k\|_{2,\mu}^2$$

(51)

where, $L(\Pi_{\mathcal{B}}) = \sqrt{(1-\gamma)^2 \sum_{k=1}^{\infty} k \gamma^{k-1} l(k)}$. Then, we can bound $L_2$ by

$$\lim_{n \to \infty} L_2 \leq \frac{2\gamma}{(1-\gamma)^2} L(\Pi_{\mathcal{B}}) \sup_{k \geq 0} \|\epsilon_k\|_\mu$$

(52)

With the upper bound of $L_1$ and $\lim_{n \to \infty} L_2$, we can complete the proof by adding these two term together.

$$\lim_{n \to \infty} \|Q^* - Q^{\pi_n}\|_{\rho_0} \leq \frac{2\gamma}{(1-\gamma)^2} \left[ L(\Pi_{\mathcal{B}}) \sup_{k \geq 0} \|\epsilon_k\|_\mu + \frac{1-\gamma}{2\gamma} \alpha(\Pi_{\mathcal{B}}) \right] \tag{53}$$

# E    IMPLEMENTATION DETAILS

DOGE can build on top of standard online actor-critic algorithms such as TD3(Fujimoto et al., 2018) and SAC(Haarnoja et al., 2018). We choose TD3 as our base because of its simplicity compared to other methods. We build DOGE on top of TD3 by simply plugging the state-conditioned distance function as a policy regularization term during policy training process. Then, the learning objective of policy $\pi$ in Eq. (7) can be formulated as:

$$\pi = \arg \max_\pi \min_\lambda \mathbb{E}_{s \sim \mathcal{D}} \left[ \beta Q(s, \pi(s)) - \lambda(g(s, \pi(s)) - G) \right] \quad \text{s.t.} \ \lambda \geq 0 \tag{54}$$

The $Q$ function, policy and state-conditioned distance function networks are represented by 3 layers ReLU activated MLPs with 256 units for each hidden layer and are optimized by Adam optimizer. In addition, we normalize each dimension of state to a standard normal distribution for Mujoco tasks. The hyperparameters of DOGE are listed in Table 2.

Table 2: Hyperparameters of DOGE

| | Hyperparameters | Value |
|---|---|---|
| Shared parameters | Optimizer | Adam |
| | Standard Normalize state | True for Mujoco |
| | | False for AntMaze |
| | Batch size | 256 |
| | Layers | 3 |
| | Hidden dim | 256 |
| TD3 | Actor learning rate | $3 \times 10^{-4}$ |
| | Critic learning rate | $3 \times 10^{-4}$ for Mujoco |
| | | $1 \times 10^{-3}$ for AntMaze |
| | Discount factor $\gamma$ | 0.99 for Mujoco |
| | | 0.995 for AntMaze |
| | Number of iterations | $10^6$ |
| | Target update rate $\tau$ | 0.005 |
| | Policy noise | 0.2 |
| | Policy noise clipping | 0.5 |
| | Policy update frequency | 2 |
| State-Conditioned Distance Function | Learning rate | $1 \times 10^{-3}$ for Mujoco |
| | | $1 \times 10^{-4}$ for AntMaze |
| | Number of noise actions $N$ | 20 |
| | Number of iterations $N_g$ | $10^5$ for Mujoco |
| | | $10^6$ for AntMaze |
| DOGE | $\alpha$ | {7.5, 17.5} Mujoco |
| | | {5, 10, 70} AntMaze |
| | Lagrangian multiplier $\lambda$ | clipped to [1, 100] |
| | $\lambda$ learning rate | $3e - 4$ |

## E.1    TD3'S IMPLEMENTATION DETAILS

For the choice of the Critic learning rate and discount factor $\gamma$, we find that for AntMaze tasks, a high Critic learning rate can improve the stability of value function during training process. This may be because the AntMaze tasks require the value function to dynamic programs more times to "stitch" suboptimal trajectories than Mujoco tasks. Therefore, we choose $1 \times 10^{-3}$ and 0.995 as the Critic learning rate and discount factor $\gamma$ for AntMaze tasks, respectively. The other implementations such as policy noise scale and policy noise clipping are the same with author's implementation (Fujimoto et al., 2018).

### E.2 STATE-CONDITIONED DISTANCE FUNCTION'S IMPLEMENTATION DETAILS

We sample $N = 20$ noise actions from a uniform distribution that covers the full action space to approximate the estimation value in Eq. (4). We find $N = 20$ can balance the computation complexity and estimation accuracy and is the same sample numbers with CQL (Kumar et al., 2020b). The ablation of $N$ can be found in Fig. 15. The practical training objective of the state-conditioned distance function is as follows:

$$\min_g \mathbb{E}_{(s,a)\in\mathcal{D}, \hat{a}_i \sim Unif(\mathcal{A})} \left[ \frac{1}{N} \sum_{i=1}^{N} [\|a - \hat{a}_i\| - g(s, \hat{a}_i)]^2 \right] \tag{55}$$

We find that a wider sample range than the max action space $[-a_{\max}, a_{\max}]$ is helpful to characterize the geometry of the full offline dataset. This is because some actions in the offline dataset lie at the boundary of the action space, which can only be sampled with little probability when sampling from a narrow distribution. At this time, the noise actions may not cover the geometry information near the boundary. Therefore, we sample noise actions from a uniform distribution that is 3 times wider than the max action space, *i.e.*, $\hat{a} \sim Unif[-3a_{\max}, 3a_{\max}]$. For the learning rate, we find that a high learning rate enables a stable training process in Mujoco tasks. Therefore, we choose $1 \times 10^{-3}$ and $1 \times 10^{-4}$ as the distance function learning rate for Mujoco and AntMaze, respectively. We also observe that for Mujoco tasks, $10^5$ iterations can already produce a relatively good state-conditioned distance function, and training more times won't hurt the final results. To reduce computation, we only train the state-conditioned distance function for $10^5$ steps for Mujoco tasks.

### E.3 HYPERPARAMETERS TUNING OF DOGE

The scale of $\alpha$ determines the strength of policy constraint. We tune $\alpha$ to balance the trade-off between policy constraint and policy improvement. To be mentioned, $\alpha$ is tuned within only 5 candidates for 20 tasks (17.5 for hopper-m, hopper-m-r and all Mujoco random datasets; 7.5 for other Mujoco datasets; 5 for antmaze-u; 10 for antmaze-u-d; 70 for other AntMaze tasks). This is acceptable in offline policy tuning following (Kumar et al., 2019; Brandfonbrener et al., 2021). To ensure numerical stability, we clip the Lagrangian multiplier $\lambda$ to $[1, 100]$. We also find a large initial $\lambda$ enables stable training for Mujoco tasks but slows down AntMaze training. Therefore, the initial value of Lagrangian multiplier $\lambda$ is 5 for Mujoco and 1 for AntMaze tasks, respectively.

### E.4 PSEUDOCODE OF DOGE

The pseudocode of DOGE is listed in Algorithm 1. Changes we make based on TD3 (Fujimoto et al., 2018) are marked in red. The only modification is the training process of the additional state-conditioned distance function and the constrained actor update. We can perform 1M training steps on one GTX 3080Ti GPU in less than 50min for Mujoco tasks and 1h 40min for AntMaze tasks.

---

**Algorithm 1** Our implementation for DOGE

---

**Require:** Dataset $\mathcal{D}$. State-conditioned distance network $g_\psi$. Policy network $\pi_\phi$ and target policy network $\pi_{\phi'}$ with $\phi' \leftarrow \phi$. Value network $Q_{\theta_i}, i = 1, 2$ and target value network $Q_{\theta'_i}, i = 1, 2$ with $\theta'_i \leftarrow \theta_i$. State-conditioned distance network training steps $N_g$. Policy update frequency $m$.
1: **for** $t = 0, 1, ..., M$ **do**
2:      Sample mini-batch transitions $\{(s_i, a_i, r_i, s'_i)\} \sim \mathcal{D}$
3:      **if** $t < N_g$ **then**
4:          **State-Conditioned Distance Function Update:** Update $\psi$ as Eq. (55) shows.
5:      **end if**
6:      **Critic Update:** Update $\theta_i$ using policy evaluation method in TD3.
7:      **if** $t \bmod m = 0$ **then**
8:          **Constrained Actor Update:** Update $\phi, \lambda$ via Eq. (54).
9:          Update target networks: $\theta'_i \leftarrow \tau\theta_i + (1 - \tau)\theta'_i, \phi' \leftarrow \tau\phi + (1 - \tau)\phi$
10:      **end if**
11: **end for**

---

## E.5 EXPERIMENT SETUP FOR THE IMPACT OF DATA GEOMETRY ON DEEP $Q$ FUNCTIONS

We consider an one-dimensional random walk task with a fixed-horizon (50 steps for each episode), where agents at each step can move in the range of $[-1, +1]$ and the state space is a straight ranges from $[-10, 10]$. The destination is located at $s = 10$. The closer the distance to the destination, the larger the reward that the agent can get. The discount factor $\gamma = 0.9$. The reward function is defined as follows:

$$r = \frac{400 - (s' - 10)^2}{400} \tag{56}$$

We generate offline datasets with different geometry and train the agent based on these datasets. Each synthetic dataset consists of 200 transition steps. We get the approximated $Q$ value $\hat{Q}$ by training TD3 for $1e+4$ steps each dataset. The learning rate of Actor and Critic networks are both $10^{-3}$. The other implementation details are the same as the implementation of original TD3 (Fujimoto et al., 2018). The true $Q$ function can be get by Monte-Carlo estimation. We find that the near-destination states hold higher approximation error than that far away from the destination due to the scale of true $Q$ value near the destination is large. To alleviate the impact of $Q$ value scale on the approximation error, we define the relative approximation error as follows:

$$\hat{\epsilon}(s, a) = \epsilon(s, a) - \min_a \epsilon(s, a) \tag{57}$$

where, $\epsilon(s, a) = \hat{Q}(s, a) - Q(s, a)$. The relative error in the above definition eliminates the effect of different states on the approximation error and can capture the over-estimation error that we care about. We plot the relative approximation error of deep $Q$ functions with different random seeds and data geometry in Fig. 13.

## F ADDITIONAL EXPERIMENT RESULTS

### F.1 COMPARISON OF GENERALIZATION ABILITY

In the well known AntMaze task in D4RL benchmark (Fu et al., 2020), where an ant needs to navigate from the start to the destination in a large maze. The trajectories with coordinates at $x \times y \in [4, 13] \times [7, 9] \cup [11.5, 20.5] \times [11, 13]$ in AntMaze medium tasks and $x \times y \in [10.5, 21] \times [7, 9] \cup [19, 29.5] \times [15, 17]$ in AntMaze large tasks are clipped, as Fig. 8 shows.

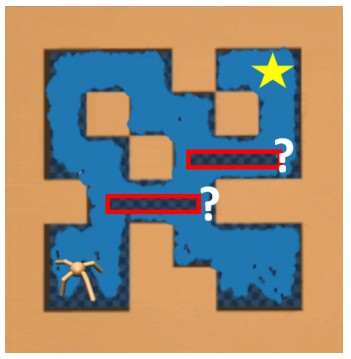 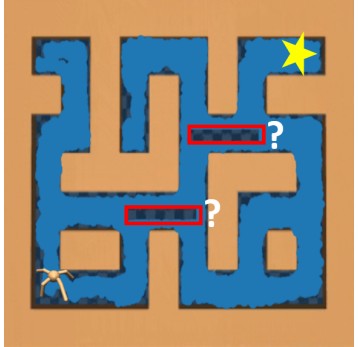

(a) Modified Medium AntMaze          (b) Modified Large AntMaze

Figure 8: The trajectories in the offline dataset are visualized as blue. Data transitions of two small areas on the crtical pathways to the destination have been removed (red box).

These clipped data counts only about one-tenth of the original dataset and lies in the close proximity of the original trajectories. Under these modified datasets, simply relaying on "stitching" data transitions is not enough to solve the navigation problems. We evaluate representative policy constraint method (TD3+BC (Fujimoto & Gu, 2021)), value regularization method (CQL (Kumar et al., 2020b)), in-sample learning method (IQL (Kostrikov et al., 2021b)) and DOGE (our method) on these modified datasets. The evaluation results before and after clipping the trajectories are listed in Table 3. The

learning curves for the modified AntMaze medium and AntMaze large tasks are listed in Fig. 9 and Fig. 4.

Observe in Table 3 that existing offline RL methods fail miserably and suffer from severe performance drops. By contrast, DOGE maintains competitive performance after the modification of the dataset and shows good generalization ability on unknown areas.

Apart from above experiments, we also evaluate DOGE when removing only one area: $[10.5, 21] \times [7, 9], [10.5, 21] \times [7, 9]$ for AntMaze-large datasets and $[4, 13] \times [7, 9], [4, 13] \times [7, 9]$ for AntMaze-medium datasets. The final results can be seen in Table 4.

Table 3: The performance drop after removing the data at the only way to destination.

| Datatset type | | TD3+BC | CQL | IQL | DOGE(ours) |
|---|---|---|---|---|---|
| antmaze-m-p-v2 | full data | 0 | 65.2±4.8 | 70.4±5.3 | **80.6±6.5** |
| | miss data | 0 | 10.7±18.4 | 10.2±2.2 | **33.2±27.3** |
| Performance drop ↓ | | - | 84% | 86% | **59%** |
| antmaze-m-d-v2 | full data | 0 | 54.0±11.7 | 74.6±3.2 | **77.6±6.1** |
| | miss data | 0 | 8.5±5.3 | 7.6±5.7 | **40.2±32.9** |
| Performance drop ↓ | | - | 84% | 90% | **48%** |
| antmaze-l-p-v2 | full data | 0 | 18.8±15.3 | 43.5±4.5 | **48.2±8.1** |
| | miss data | 0 | 0 | 1.0±0.7 | **22.4±15.9** |
| Performance drop ↓ | | - | 100% | 98% | **54%** |
| antmaze-l-d-v2 | full data | 0 | 31.6±9.5 | **45.6±7.6** | 36.4±9.1 |
| | miss data | 0 | 0 | 5.2±3.1 | **14.6±11.1** |
| Performance drop ↓ | | - | 100% | 89% | **60%** |

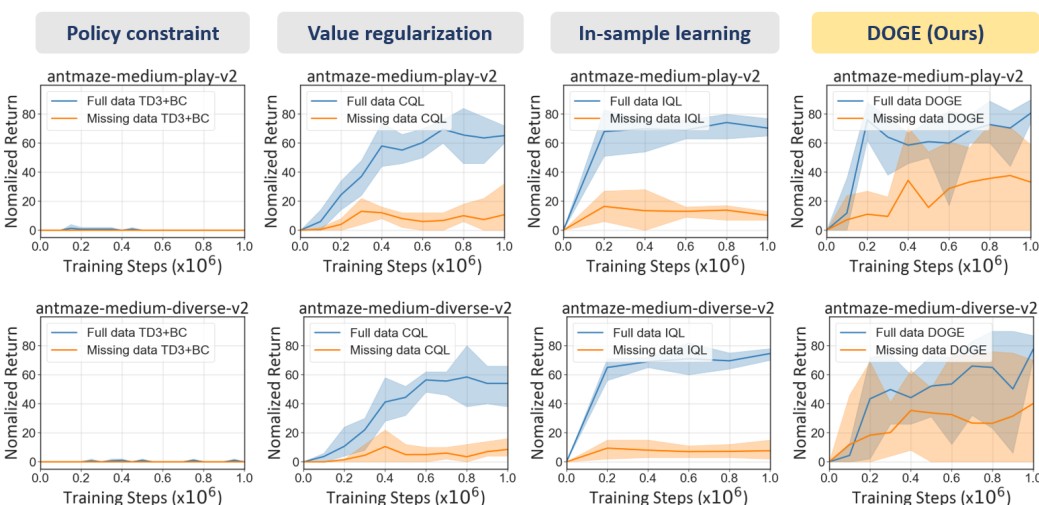

Figure 9: Evaluation on TD3+BC(Fujimoto & Gu, 2021), CQL(Kumar et al., 2020b), IQL(Kostrikov et al., 2021b), and DOGE (ours) before and after removing the data shown in Fig.8a for AntMaze medium tasks.

Table 4: Ablation for DOGE generalization with different removal areas.

| Dataset | Full dataset | One removal | Two removal |
|---|---|---|---|
| antmaze-m-p-v2 | 80.6±6.5 | 62.3±7.5 | 33.2±27.3 |
| antmaze-m-d-v2 | 77.6±6.1 | 41.3±42.8 | 40.2±32.9 |
| antmaze-l-p-v2 | 48.2±8.1 | 26.4±19.4 | 22.4±15.9 |
| antmaze-l-d-v2 | 36.4±9.1 | 12.3±4.2 | 14.6±11.1 |
| Total score | 242.8±29.8 | 142.3±73.9 | 110.4±87.2 |

## F.2 ADDITIONAL COMPARISON WITH TD3+BC

In this section, we further demonstrate the superiority of DOGE over our most related practical work TD3+BC (Fujimoto & Gu, 2021). One can find that the biggest difference between DOGE and TD3+BC lies in the policy constraint used for policy optimization:

- **TD3+BC**: constrains the policy to minimize the MSE BC loss.

- **DOGE**: constrains the policy to minimize the learned state-conditioned distance function $g(s, a)$.

As discussed in Section 3.1, the learned distance function $g(s, a)$ can capture the global geometric information of the offline dataset, while the MSE BC loss can only provide local sample-to-sample regularization, which may be noisy, especially in datasets that contain low-quality samples. Taking Figure 10 as an illustration, under strict BC constraint, policy learning on noisy low-quality samples may provide contradicting learning signals to near-optimal samples, which can cause inferior policy performance and unstable training process. By contrast, the state-conditioned distance function $g(s, a)$ in DOGE is trained on the whole dataset and hence brings global geometric information, which is far more informative and stable as compared with the MSE BC loss.

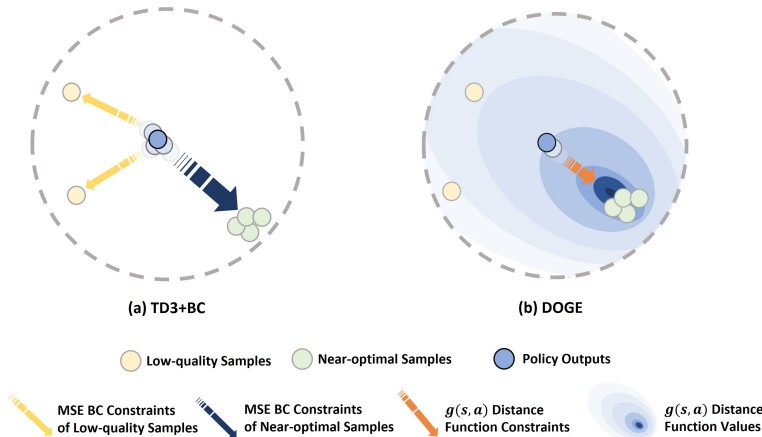

Figure 10: Illustrations of the differences between (a) the MSE BC constraint of TD3+BC and (b) the state-conditioned distance function constraint of DOGE. In (a), the MSE BC constraint in TD3+BC blindly enforces the imitation behavior on any data samples, which may lead to an inferior policy in the presence of noisy low-quality samples. In (b), the state-conditioned distance function $g(s, a)$ can provide more informative global dataset geometry information to guide the stable learning of the policy.

To better illustrate the superiority of DOGE over TD3+BC, we add extra comparative experiments with TD3+BC on a new set of mixed-quality datasets. In halfcheetah-random dataset, we add different proportions (1% to 20%) of the near-optimal halfcheetah-medium-expert dataset to form new mixed datasets and evaluate how TD3+BC and DOGE perform. See Figure 11 for detailed results.

Figure 11 shows that DOGE enjoys more performance gains when the random dataset involves near-optimal data, while TD3+BC is heavily influenced by the local information from the larger proportion of the low-quality random data. Moreover, TD3+BC suffers from severe oscillation and training instability, while DOGE enjoys a stable training process due to the use of the more informative state-conditioned distance constraint that captures the overall dataset geometry.

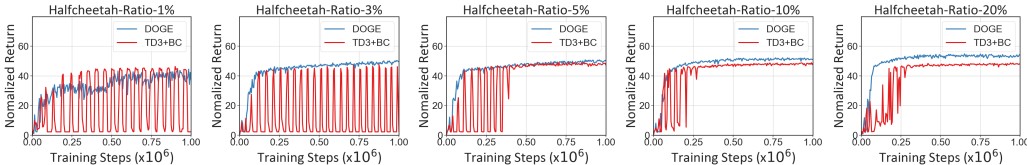

Figure 11: Comparisons between DOGE and TD3+BC on mixed datasets with different proportions of halfcheetah-medium-expert dataset added into halfcheetah-random dataset. Ratio-1% means 1% medium-expert dataset is added into the original halfcheetah-random dataset. TD3+BC suffers severe oscillation and training instability, while DOGE enjoys stable training processes and substantial performance gains.

### F.3 COMPARISON WITH UNCERTAINTY-BASED METHODS

We also compare DOGE with SOTA uncertainty-based offline RL approaches, including EDAC (An et al., 2021) and PBRL (Bai et al., 2021) are more complex D4RL AntMaze tasks. The final results are presented in Table 5. Table 5 shows that the SOTA uncertainty-based methods are unable to provide reasonable performance on the difficult Antmaze tasks, despite that they can achieve good performance on simpler MuJoCo tasks. A similar finding is also reported in a recent offline RL study (Anonymous, 2023).

In practical implementation of EDAC and PBRL, to obtain relatively accurate uncertainty measures and achieve reasonable performance, these methods typically need dozens of ensemble Q-networks, which can be quite costly and inefficient. Moreover, heavy hyperparameter tuning is also required for them to obtain the best performance. In contrast, our method quantifies the generalization ability of the Q-function from the perspective of dataset geometry and is trained using a simple regression loss in Eq. (4), which enjoys better training stability and simplicity.

Table 5: Average normalized scores over 5 seeds on Antmaze tasks

| Dataset | EDAC | PBRL | DOGE(Ours) |
|---|---|---|---|
| antmaze-u-v2 | 0 | 0 | **97.0±1.8** |
| antmaze-u-p-v2 | 0 | 0 | **63.5±9.3** |
| antmaze-m-p-v2 | 0 | 0 | **80.6±6.5** |
| antmaze-m-d-v2 | 0 | 0 | **77.6±6.1** |
| antmaze-l-p-v2 | 0 | 0 | **48.2±8.1** |
| antmaze-l-d-v2 | 0 | 0 | **36.4±9.1** |

### F.4 ADDITIONAL ANALYSIS ON DISTANCE FUNCTION

We report the learning curves of the state-conditioned distance function $g(s, a)$ trained on different datasets (including hopper-m-v2, halfcheetah-m-v2, and walker2d-m-v2 in Figure 12. Our proposed state-conditioned distance function is learned through a simple regression task (Eq. (4)), which is very easy to train. Figure 12 shows that it reaches convergence within only 1K training steps on D4RL MuJoCo medium datasets.

We also change the network configurations (i.e., number of hidden layers and hidden units) of the state-conditioned distance function $g(s, a)$ to investigate how the expressivity of $g$ influences the performance of the policy. Table 6 shows that DOGE achieves similar performance across different $g$ network configurations, indicating that DOGE is robust to model complexity and expressivity of the state-conditioned distance function.

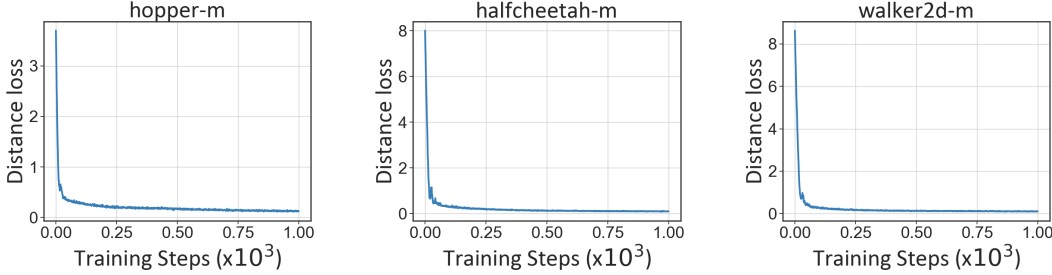

Figure 12: Learning curves of the state-conditioned distance function $g(s, a)$

Table 6: Normalized scores of DOGE trained on distance functions with different network configurations. [128, 128] means $g$ network has 2 hidden layers with 128 units. [256, 256, 256] means 3 hidden layers with 256 units.

| Dataset | [128, 128] | [256, 256] | [256, 256, 256] |
|---|---|---|---|
| hopper-m | 99.4 | 101.4 | 98.6 |
| halfcheetah-m | 47.4 | 46.9 | 45.3 |
| walker2d-m | 85.3 | 86.4 | 86.8 |

## F.5 ADDITIONAL EXPERIMENTS OF THE IMPACT OF DATA GEOMETRY ON DEEP $Q$ FUNCTIONS

We run several experiments with different random seeds (see Figure 13). Although the approximation error pattern of different random seeds is not the same, they all perform in the same manner that deep $Q$ functions produce relatively low approximation error inside the convex hull of training data. We refer to this phenomenon as **deep $Q$ functions interpolate well but struggle to extrapolate**.

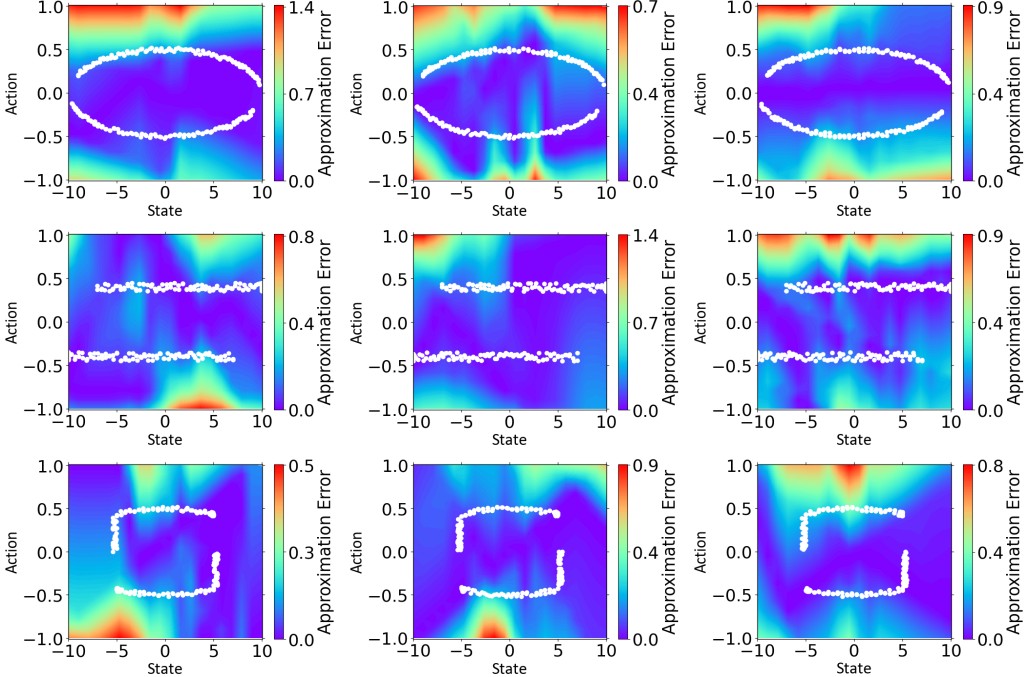

Figure 13: The figures above depict the effect of different data geometries on the final deep $Q$ functions approximation error. The training data are marked as white dots.

## G  ABLATIONS

We conduct ablation studies on the effect of $\alpha$ in $\beta = \frac{\alpha}{\frac{1}{n}\sum_{i=1}^{n}|Q(s_i,a_i)|}$ (see Figure 14), the non-parametric threshold $G$ in Eq. (6) (see Figure 16) and the non-parametric number of noise actions $N$ to train state-conditioned distance function (see Figure 15) on the performance of the final algorithm. We also conduct ablation studies on the effect of $G$ on the Lagrangian multiplier $\lambda$ (see Figure 17).

For $\alpha$, we add or subtract 2.5 to the original value. For $N$, we choose $N = 10, 20, 30$ to conduct experiments respectively. For $G$, we choose 30%, 50%, 70%, 90% and 100% upper quantile of the distance value in mini-batch samples and the results can be found in Table 7.

Table 7: Ablations on G with different quantile.

| Dataset | $G = 30\%$ | $G = 50\%$ | $G = 70\%$ | $G = 90\%$ | $G = 100\%$ |
|---|---|---|---|---|---|
| hopper-r-v2 | 19.8±0.3 | **21.1±12.6** | 15.5±13.5 | 17.6±12.2 | 16.4±12.4 |
| halfcheetah-r-v2 | **19.4±0.6** | 17.8±1.2 | 17.8±0.7 | 17.7±1.0 | 17.7±0.8 |
| walker2d-r-v2 | **2.6±3.9** | 0.9±2.4 | 2.2±2.6 | 1.8±3.3 | 2.2±3.2 |
| hopper-m-v2 | 44.6±5.7 | 98.6±2.1 | **99.4±0.4** | 91.5±9.9 | 32.9±54.3 |
| halfcheetah-m-v2 | 41.3±1.2 | 45.3±0.6 | 46.0±0.1 | 46.0±0.8 | **46.1±0.5** |
| walker2d-m-v2 | 83.7±7.5 | 86.8±0.8 | **87.3±1.6** | 69.9±28.9 | 84.2±1.0 |
| hopper-m-r-v2 | 51.5±11.2 | 76.2±17.7 | **79.6±36.9** | 78.4±27.6 | 65.7±37.2 |
| halfcheetah-m-r-v2 | 5.9±5.7 | 42.8±0.6 | **43.2±0.1** | 42.2±0.8 | 42.0±0.6 |
| walker2d-m-r-v2 | 28.3±14.3 | 87.3±2.3 | **87.9±2.4** | 77.8±21.6 | 78.6±24.1 |
| hopper-m-e-v2 | 61.7±10.4 | **102.7±5.2** | 82.8±5.8 | 88.9±17.7 | 70.0±48.4 |
| halfcheetah-m-e-v2 | 46.9±5.2 | **78.7±8.4** | 75.1±15.4 | 73.5±13.6 | 69.9±8.7 |
| walker2d-m-e-v2 | 110.5±0.7 | 110.4±1.5 | **111.1±0.5** | 110.2±22.5 | 80.0±54.3 |

Seen from Table 7 that using different $G$ for different tasks may achieve even better performance. Particularly, for some datasets with diverse data distributions that need to find good data from suboptimal data, a more tolerant quantile (e.g., $G = 70\%$) can reasonably extend feasible region and increase the opportunity to find the optimal policy, such as hopper-m-r, halfcheetah-m-r, walker2d-m-r, hopper-m-e, halfcheetah-m-e. However, an overly relaxed quantile (e.g., $G = 90\%$ and 100%) increases the risk of including problematic OOD actions in policy learning, causing performance drop due to value overestimation and high variance.

By contrast, an overly restrictive quantile such as $G = 30\%$ can be over-conservative and cause significant constraints violations that impede policy learning, as constraints satisfaction is favored over the max-Q operation in most updates. This can be reflected in the additional results for the Lagrangian multiplier $\lambda$ (see Appendix E.2 for learning curves and Figure 11 for additional ablations), where $\lambda \to \infty$ for some tasks under $G = 30\%$. This will cause the suboptimality gap ($\frac{1-\gamma}{2\gamma}\alpha(\Pi_{\mathcal{D}})$) in Theorem 3 to dominate the performance bound, leading to inferior policy.

As hyperparameter tuning in practical offline RL applications without online interaction is very difficult, to reduce the computational load, we set $G = 50\%$ as default in a non-parametric manner, since it consistently achieves good performance, and is neither too conservative nor too aggressive for most tasks.

Observe in Figure 14 that DOGE maintains the similar performance with the changes of $\alpha$ on most of Mujoco tasks. At the same time, we also observe that the effect of $N$ on the experiment is not obvious. Compared with $N$ and $\alpha$, we find that $G$ has a more significant effect on the experimental results. Observe in Figure 16 that a small $G$ usually causes the policy set induced by DOGE to be too small to obtain near-optimal policy. By contrast, a large $G$ is not likely to cause excessive error accumulation and hence maintains relatively good performance.

In addition, the ablation studies show that our method is hyperparameter-robust and maintains good performance with changes in hyperparameters.

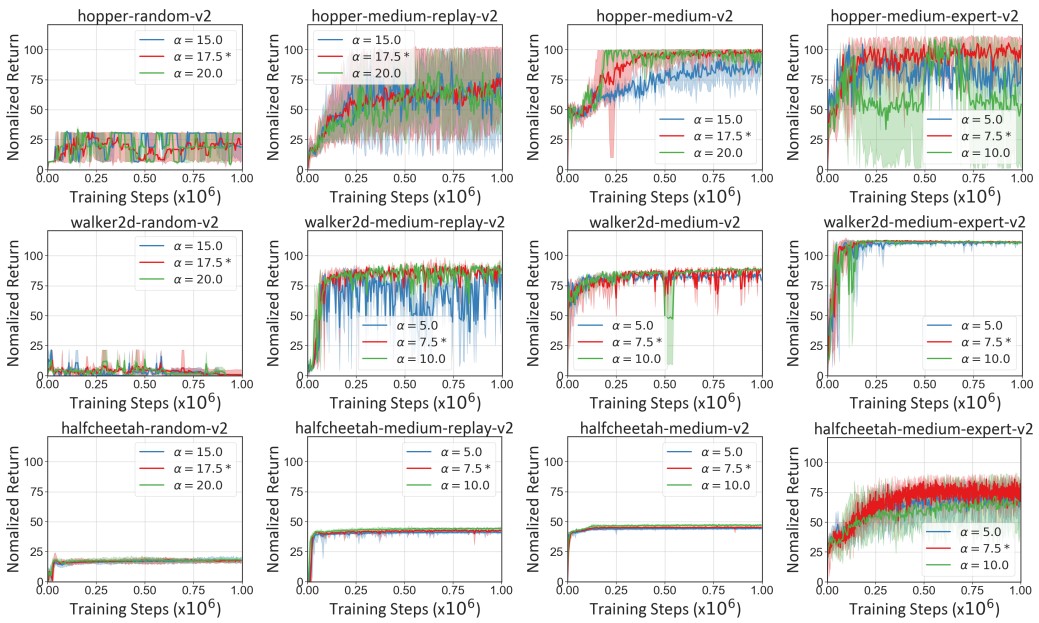

Figure 14: Ablation for $\alpha$. Error bars indicate min and max over 5 seeds.

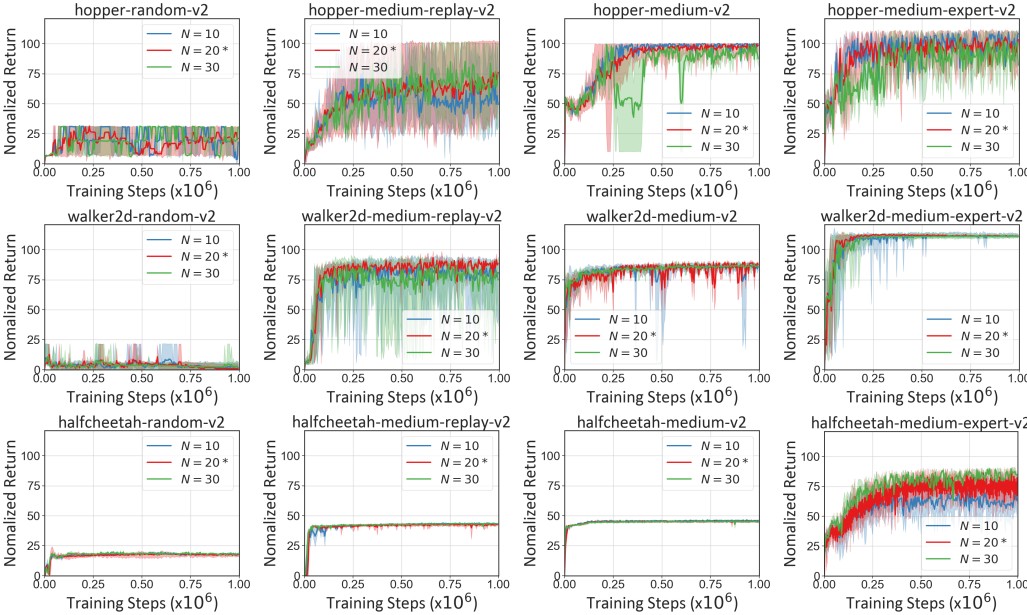

Figure 15: Ablation for $N$. Error bars indicate min and max over 5 seeds.

## H  LEARNING CURVES

The learning curves for Mujoco and AntMaze tasks are listed in Fig. 18 and Fig.19. The learned policies are evaluated for 10 episodes and 100 episodes each seed for Mujoco and AntMaze tasks, respectively. For AntMaze tasks, we subtract 1 from rewards for the AntMaze datasets following (Kumar et al., 2020b; Kostrikov et al., 2021b).

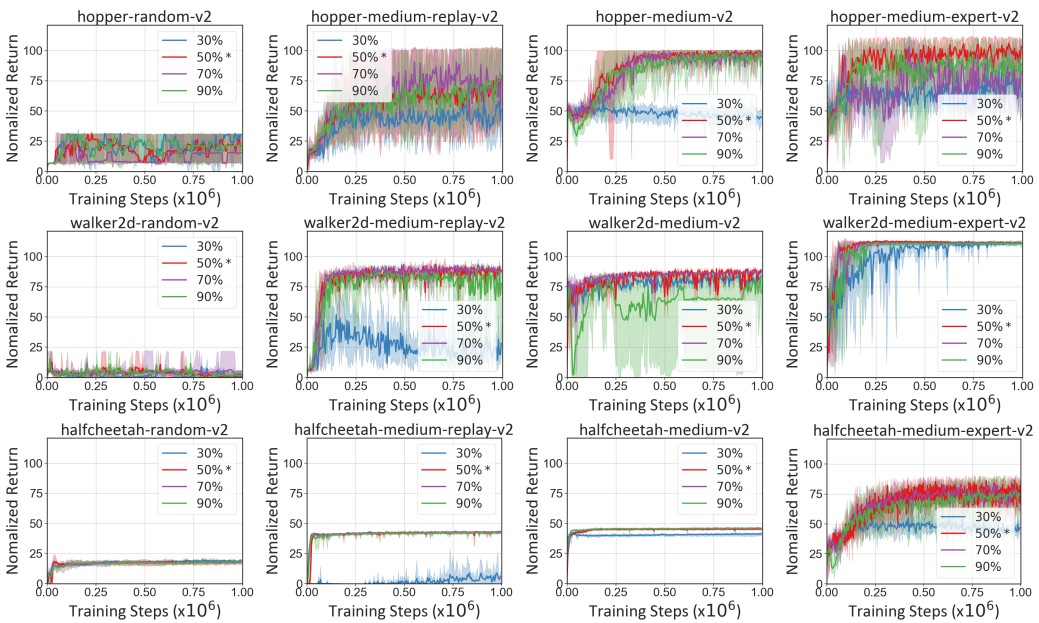

Figure 16: Ablation for $G$. Error bars indicate min and max over 5 seeds.

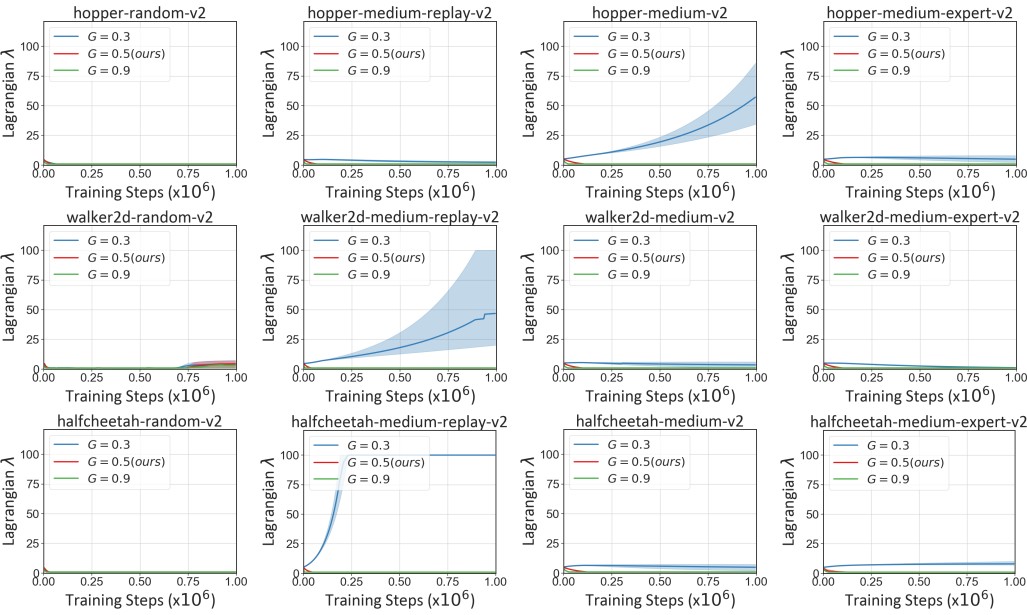

Figure 17: Ablation for $\lambda$. Error bars indicate min and max over 5 seeds.

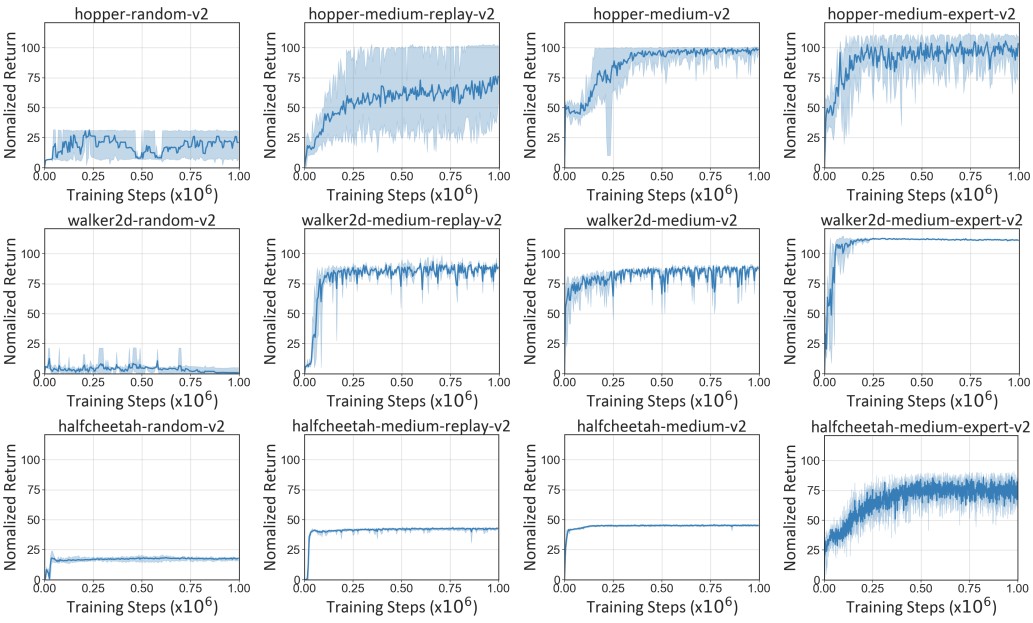

Figure 18: Learning curves for Mujoco Tasks. Error bars indicate min and max over 5 seeds.

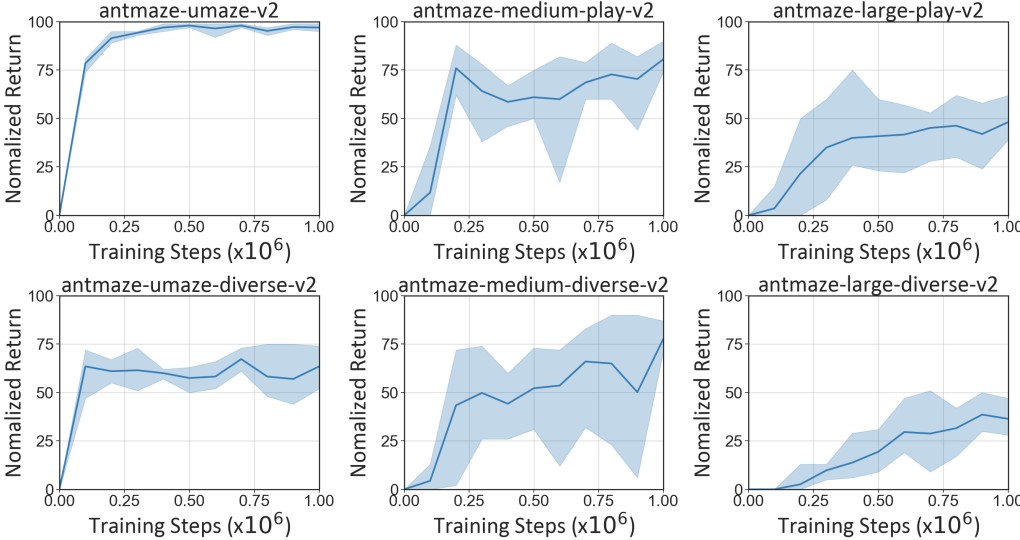

Figure 19: Learning curves for AntMaze Tasks. Error bars indicate min and max over 5 seeds.

