# OpenReview forum: "When Data Geometry Meets Deep Function: Generalizing Offline Reinforcement Learning"
_ICLR.cc/2023/Conference — ICLR 2023 poster_

### Official Review · Reviewer_hPVe · 2022-10-23

**Confidence:** 3
**Correctness:** 4
**Technical Novelty And Significance:** 3
**Empirical Novelty And Significance:** 3
**Recommendation:** 8

**Clarity, Quality, Novelty And Reproducibility:**

**Clarity** -- The clarity of the paper is reasonable, but could be improved somewhat. While it may be my relative lack of familiarity of the area, I found it difficult to follow some of the authors’ lines of reasoning. See weaknesses section above. However, other aspects of the paper, such as the equations and experiments are generally clearly well-defined and explained. For instance, Figure 1 clearly lays out the issue that the authors intend to tackle.

**Quality** -- The quality of the paper appears to be quite high. The authors provide a compelling empirical analysis on a variety of offline RL datasets/environments, comparing with the relevant model-free baselines, coupled with a theoretical analysis of their approach. The authors use figures to explain their motivation, aspects of their approach, and experiments. This all seems to meet the level of a top-level conference publication.

**Novelty** -- The paper is somewhat novel. While the method itself ultimately amounts to adding a regularizer to the policy optimization objective, getting to that point involves a non-trivial amount of insight, which likely took considerable work. Beyond the method, the authors also provide a fair amount of theoretical analysis, as well as a set of empirical analyses around the AntMaze environments looking at interpolation in a more focused way.

**Reproducibility** -- I suspect that the paper is reproducible. The method itself is fairly simple, training an additional distance estimator on the offline dataset and using this as an additional policy regularizer. The authors present results on a standard offline RL benchmark, where they generally perform comparably or favorably, using multiple seeds/evaluations. This suggests that the positive outcome of the results in not merely from random chance.

**Strength And Weaknesses:**

**Strengths**

- **Strong empirical evaluation**. The authors evaluate their method, DOGE, on a standard offline RL benchmark, D4RL, comparing with the relevant baseline approaches. This is in both the standard locomotion set of tasks, as well as the more difficult AntMaze environments. In all cases, the authors’ approach generally compares favorably or performs similarly. The authors also include an analysis on the AntMaze environments, removing transitions from some regions of the state-space. This allows them to test interpolation abilities of previous methods. I view this empirical evaluation as the strongest point of the paper.

- **Relatively simple method**. While the authors have included a considerable degree of theoretical analysis, the method that they arrive at is ultimately fairly simple, involving a learned, state-conditional action distance estimator trained from offline trajectories. This then enters policy optimization as an additional regularization term. I see this as no more complex than many previous methods, e.g., which learn behavior priors. This simplicity may lead to more widespread adoption of the method in the RL community, making DOGE a useful baseline for future works.

- **Empirical and theoretical analysis**. The authors combine empirical and theoretical analyses. While much of the theoretical analyses are outside of my area of expertise, I appreciate that these can be important. Again, this may broaden the appeal of the paper to the wider RL community.


**Weaknesses**

- **Unclear in places**. At various points throughout the paper, I found it difficult to connect the authors’ theoretical connections and motivations to the particular method proposed. For instance, the authors’ initial motivation is framed in terms of better interpolation of the state-action space of the dataset. Yet, the authors translate this into something akin to estimating the cross entropy of the behavior policy w.r.t. a uniform policy, conditioned on the state. It’s unclear how this exactly relates to the state-action distance initially motivated by the authors. I was also left uncertain about whether the authors are arguing that the state-space, action-space, or both should be interpolated. The initial motivation leads me to believe that it’s both, whereas the proposed method seems to tackle the action-space and the empirical analyses only tackle the state-space. Along similar lines, the state-conditional action-distance penalty is introduced as a regularizer, but it’s unclear how this translates into estimating and propagating state uncertainty. I would have thought that this regularizer would by integrated over space/time by entering the definition of the Q-value (similar to KL-regularizers, e.g., in SAC), but this seems to not be the case. I also found the diagrams difficult to interpret (although I did appreciate that they were included).

- **Unclear how, precisely, this tackles interpolation better than previous methods**. The authors motivate their approach from the perspective of improving the ability to interpolate regions of the dataset, using the perspective of deep Q-networks in the interpolation and extrapolation regimes. In interpolated regions, this error is bounded, whereas in extrapolation regions, it is not (although it will still be correct nearby the data manifold). This then leads the authors to propose their state-conditional action distance estimator. However, previous methods, like BEAR, use ensembles to estimate something like Q-value uncertainty, which is then penalized. Following the authors’ arguments, there should be relatively less uncertainty in interpolated regions as compared with extrapolated regions, allowing these methods to also handle interpolation. Where is my misunderstanding here? Why is this method uniquely suited to handling interpolation, whereas other methods, as the authors show empirically, are not? Ultimately, with a finite dataset and continuous action space, all methods have to interpolate for every single point — does this method just have a wider window of interpolation?

- **Applicability/generality unclear**. While reading the paper, I felt as though there were a number of unstated assumptions that the authors were making. The authors initially motivate their approach by considering interpolation vs. extrapolation in (continuous-action) deep Q-networks, though it seems the same technique could not be applied to discrete action space. The tasks considered by the authors consist of relatively smooth dynamics and reward functions, thus, interpolation in an environment like AntMaze is valid, especially if it allows you to connect two regions of the state-space. However, this strikes me as ultimately a heuristic, which may or may not be respected by the actual environment. Making this heuristic assumption is useful when it works out, but it’s not a general principle that should be relied upon. For example, if we consider a reward function in the shape of an inverted Mexican hat (e.g., imagine an environment with a heat source like a campfire that keeps the agent warm, but would burn the agent if it touches it directly), but you only observe data around the high-reward outer ring, then interpolation will lead you to estimate that the center is also high-value, which is fatally wrong.

**Summary Of The Paper:**

The authors suggest that current offline RL methods are overly conservative in interpolating regions of the dataset, whereas this should primarily be applied to extrapolation. They suggest that this should be controlled by the distance, which the bound/estimate using a state-conditional action distance estimator. Satisfying this distance constraint then becomes a regularizer for policy optimization. Theoretical analyses show that this provides a tighter bound on performance as compared with a previous method, BEAR. Results are presented on D4RL, comparing with relevant model-free baselines. There are also experiments on an extrapolation version of the task, removing some parts of the state/action space.

**Summary Of The Review:**

The paper presents a compelling set of empirical results, alongside theoretical analyses. While I’m fairly convinced that the method performs well against strong baselines in standard offline RL benchmarks, I found it difficult to follow some aspects of the paper, particularly with regard to motivating the approach, e.g., assumptions about the underlying environment, and translating it into a practical training technique.

**UPDATE:**

I appreciate the authors' detailed responses to me and the other reviewers. While there may be some unresolved concerns, particularly around the clarity of the paper, most of my main concerns have been addressed. I now feel that the paper warrants acceptance, and have increased my score from 6 --> 8.

---

> ### Author Response · Authors · 2022-11-13
> **References**
>
> [1] Jin, Ying, Zhuoran Yang, and Zhaoran Wang. "Is pessimism provably efficient for offline rl?." International Conference on Machine Learning. PMLR, 2021.
>
> [2] Bai, Chenjia, et al. "Pessimistic Bootstrapping for Uncertainty-Driven Offline Reinforcement Learning." International Conference on Learning Representations. 2021.
>
> [3] Deng, Zhihong, et al. "SCORE: Spurious COrrelation REduction for Offline Reinforcement Learning." arXiv preprint arXiv:2110.12468 (2021).
>
> [4] An, Gaon, et al. "Uncertainty-based offline reinforcement learning with diversified q-ensemble." Advances in neural information processing systems 34 (2021): 7436-7447.
>
> [5] Yang, Rui, et al. "RORL: Robust Offline Reinforcement Learning via Conservative Smoothing." arXiv preprint arXiv:2206.02829 (2022).
>
> [6] Anonymous. Lightweight uncertainty for offline reinforcement learning via bayesian posterior. In Submitted to The Eleventh International Conference on Learning Representations, 2023

---

> ### Author Response · Authors · 2022-11-13
> **Response to Reviewer  hPVe W3**
>
> >**W3. Applicability/generality unclear**
>
> >**...there were a number of unstated assumptions that the authors were making...For example, if we consider a reward function in the shape of an inverted Mexican hat (e.g., imagine an environment with a heat source like a campfire that keeps the agent warm, but would burn the agent if it touches it directly), but you only observe data around the high-reward outer ring, then interpolation will lead you to estimate that the center is also high-value, which is fatally wrong.**
>
>
> - This is an interesting point. If we understand correctly, the inverted Mexican hat example provided by the reviewer corresponds to how can our method generalize to OOD areas with drastically changed patterns. We must admit that our method is based on the implicit assumption that the data pattern we observe in the offline dataset can somewhat carry over to unobserved OOD areas. The inverted Mexican example actually corresponds to an extreme case that some of the OOD regions have completely different patterns than what we have observed in the given dataset. Unless some additional information is given, such as an explicit reward function indicating some areas have low rewards or some safety constraints, otherwise, it poses a problem that is almost impossible to solve for any learning-based algorithms due to the lack of effective information. We need to at least provide the agent with some related information to allow it to make a logical judgment on a completely unknown and different pattern. Still, we believe that it will be an interesting research direction about combining DNN generalization ability and safety constraint together to achieve safe OOD generalization in offline RL. We will explore it in our future work and thank you for this insightful comment.

---

> ### Author Response · Authors · 2022-11-13
> **Response to Reviewer hPVe W2**
>
> >**W2. Unclear how, precisely, this tackles interpolation better than previous methods**
>
> >**However, previous methods, like BEAR...Following the authors’ arguments, there should be relatively less uncertainty in interpolated regions as compared with extrapolated regions, allowing these methods to also handle interpolation. Where is my misunderstanding here? Why is this method uniquely suited to handling interpolation, whereas other methods, as the authors show empirically, are not?...**
> - For BEAR-like methods, they are trying to constrain the policy stay **in data support** (the region with $d_\mu(s,a)>0$ only) instead of **within or near the convex hull formed by the dataset** (the region with $d_\mu(s,a)>0$ and some regions with $d_\mu(s,a)=0$ but lie inside or near the convex hull). We agree with the reviewer that BEAR is also doing some sort of interpolation, but only limited to those in data support. In contrast, the convex hull covers not only data support data but also the OOD data that lies inside the convex hull, as data support is only a subset of the convex hull induced by the dataset. Therefore, DOGE allows more exploitation of generalizable OOD areas, which is fundamentally different from BEAR-like offline RL methods that strictly constrain policy learning only on in-distribution areas.
> - For some recent uncertainty-based offline RL methods, three main drawbacks exist:
>     1. **Computational expensive uncertainty estimation.** To measure a relatively accurate uncertainty value and achieve reasonable performance, one generally needs dozens of ensemble Q networks, which can be very costly and inefficient [2][4].
>     2. **Sensitive to hyperparameters tuning.** To achieve good performance, RORL [5] chooses different hyperparameters for almost every task. EDAC [4] varies the ensemble size across different tasks. PBRL [2] and SCORE [3] propose a carefully tuned annealing bc schedule to stabilize the training process. Such heavy hyperparameter tuning can be very problematic in practical offline RL applications since evaluation through environment interaction is generally not possible.
>     3. **Theory and implementation gap.** The theory parts of most uncertainty-based RL methods are built on Linear MDP and assume the uncertainty quantifier is a $\xi$-uncertainty quantifier [1-3][5]. In practical implementation, the uncertainty in these methods is typically approximated by the disagreement of ensemble networks, multiplied by a hyperparameter $\beta$. However, until now, there is no concrete theoretical support on whether bootstrapped ensembles can indeed produce a $\xi$-uncertainty quantifier, and one needs to adjust $\beta$ carefully to satisfy the $\xi$-uncertainty requirement.
> - Due to the above reasons, it is pretty hard to achieve reasonable empirical performance on difficult tasks like Antmaze. We run EDAC [4] and PBRL [2] on all Antmaze tasks based on official implementations. The results are summarized below as well as in Appendix F.3. None of these uncertainty-based methods can solve such hard tasks (which is also reported in a recent paper [6]), despite their good performance on simpler D4RL MuJoCo tasks. In contrast, our method quantifies the generalization ability of Q function from the perspective of dataset geometry and is trained using a simple regression loss (Eq (4)), which enjoys better training stability and simplicity.
>
> |Dataset|EDAC[4]|PBRL[2]|DOGE(ours)|
> |---|---|---|---|
> |Antmaze-u-v2|0|0|**97.0±1.8**|
> |Antmaze-u-d-v2|0|0|**64.5±9.3**|
> |Antmaze-m-p-v2|0|0|**80.6±6.5**|
> |Antmaze-m-d-v2|0|0|**77.6±6.1**|
> |Antmaze-l-p-v2|0|0|**48.2±8.1**|
> |Antmaze-l-d-v2|0|0|**36.4±9.1**|

---

> ### Author Response · Authors · 2022-11-13
> **Response to Reviewer hPVe W1**
>
> We thank the reviewer for the positive feedback of our work! Your comprehensive review and feedback are very helpful to improve the quality and clarity of our paper.
>
> >**W1. Unclear in places**
>
> >**It’s unclear how this exactly relates to the state-action distance initially motivated by the authors.**
> - We apologize for the unclear overall logical flow in our original submission. We have added Appendix A to our revised paper to provide a sketch of the overall logical flow of our paper. Please also refer to **General Response 1 and 2** for detailed discussions on the connection between our theoretical analysis and the final algorithm DOGE.
> - In short, in Theorem 1, we find that the sample-to-dataset distance plays a key role in reflecting the approximation error of deep Q functions. This motivates us to design a learnable distance function that is capable of capturing the overall dataset geometry. Our solution is the state-conditioned distance function $g(s,a)$ learned through solving the optimization problem in Eq.(6), which happens to have two desirable properties. Property 1 states that the learned optimal $g^*$ outputs the upper bound distance to the state-conditioned centroid of the training dataset. And in Property 2, we show that the negative gradient of $g^*$ at any extrapolated data points inside the convex hull of training data. So, our state-conditioned distance function can serve as an ideal tool to achieve convex-hull-based policy constraint, which can compel the learned policy to perform interpolation (relatively small and strictly upper bounded sample-to-dataset distance) rather than extrapolation (can have large sample-to-dataset distance). As a consequence, we can fully leverage the generalization power of deep Q functions and exploit in low approximation error regions (corresponds to small sample-to-dataset distance) without being too conservative or suffering from severe approximation error accumulation.
>
> >**I was also left uncertain about whether the authors are arguing that the state-space, action-space, or both should be interpolated....The state-conditional action-distance penalty is introduced as a regularizer, but it’s unclear how this translates into estimating and propagating state uncertainty.**
>
> - Motivated by Figure 2 and Theorem 1, we can see that the state-action sample-to-dataset distance plays an important role in measuring the controllability of Q values. So it is desirable to learn about the interpolated state-action data. However, as the **General Response 3** explains, OOD action is the main reason causing Q-value overestimation (during offline RL training, all states come from the dataset and thus there is no OOD issue for states), and measuring the state distance faces considerable learning difficulty. So, we focus on measuring action distance conditioned on dataset states in our practical algorithm DOGE, which is already sufficient for policy regularization.
>
>
> >**I also found the diagrams difficult to interpret (although I did appreciate that they were included).**
> - In the above responses, we explained why it is useful to characterize the state-conditioned geometry of offline datasets in practice. However, given an arbitrary state-action sample $(s,a)\in S\times A$, naively computing its distance to the closest data point in a large offline dataset can be prohibitively expensive and impractical. Therefore, it is ideal to have a learnable distance function to cheaply measure such sample-to-dataset distance while also being capable of capturing the overall dataset geometry.
> - Then, how can we achieve this in a learnable fashion? For a single state-action pair in the dataset, i.e., $(s,a)\sim\mathcal{D}$, we can uniformly sample some random actions $\hat{a}$ over the entire action space, and their distances to action $a$ form a V-shape curve with the bottom at the dataset action $a$ as the blue dash lines shown in Figure 3(a). If we combine all such V-shape curves for every state-action pair in the dataset $\mathcal{D}$, we have our final state-conditioned distance function $g(s,a)$ (black line in Figure 3(a)). This is exactly what the optimization problem Eq.(4) is doing, which finds a linear combination of all V-shape distance curves and outputs the upper bound of the state-conditioned sample-to-centroid distance of an arbitrary action $a$. Given a threshold $G$, we can define the feasible region solved by DOGE (Eq.(6)) as those state-conditioned distance values smaller than $G$ (grey shadow areas in Figure 3(a)).

---

> ### Author Response · Authors · 2022-11-22
> **Follow up**
>
> Dear reviewer hPVe,
>
> Thank you for the detailed comments and suggestions to improve our paper! Regarding your concerns, we have added the discussions and experiments in our latest revisions. We would really appreciate it if you could tell us whether your concerns are resolved. Your further comments and discussions will greatly help to improve the quality of our paper and we will be more than happy to engage in further discussions and address them.
>
> Thanks!

---

> ### Author Response · Authors · 2022-12-06
> **Thanks for the detailed review and for increasing the score!**
>
> Dear reviewer hPVe,
>
> Thank you so much for your detailed review and for raising your score from 6 to 8! We really appreciate it! Your comments helped a lot to improve the quality and clarity of our paper. We will keep the clarity issue of our paper in mind and revise our paper in the final version to be more clear.
>
> Thanks!
>
> Best regards,
>
> Authors of Paper 3842

---

### Official Review · Reviewer_kL5r · 2022-10-25

**Confidence:** 3
**Correctness:** 3
**Technical Novelty And Significance:** 3
**Empirical Novelty And Significance:** 2
**Recommendation:** 6

**Clarity, Quality, Novelty And Reproducibility:**

This paper would benefit from being proofread. I realize that English is not a first language for most people, and I am of course not using this to base my judgement of the work, but many sentences are ambiguous and make understanding the details and the arguments within this work difficult.

>  Inspired by this, we propose a new method, DOGE (Distance-sensitive Offline RL with better GEneralization)

Maybe that makes me old school but not every method needs a catchy name, much less one named after a 2010 meme. That just makes us look silly?

The paper is otherwise relatively straightforward. I feel like the mathematics of the paper are a bit more obscuring that they are enlightening, at least I don't obviously see important lessons that we can take from them, nor that are immediately reflected empirically.

I could probably reproduce this paper, perhaps not exactly, but certainly in spirit given the level of detail.

**Strength And Weaknesses:**

Strengths:
- The proposed method stems from a series of interesting intuitions
- The proposed method seems to be able to capture something about the MDP which increases its performance in an offline RL context

Weaknesses:
- The text is hard to read at times
- The theory within the paper relies heavily on quantities which may not scale intuitively
- There are some strange empirical evaluation choices, in particular:
    - If I understand correctly, the core of the proposed method is to contrain a (continuous) policy to output actions that are not "too far" from the training data's actions. Yet, the main way this is tested is to remove part of the _state space_ from the training data? This is not testing the right thing, and is not even testing what the theory predicts that this method should be better at.
- It's nice to have bounds, but what's even nicer is empiricially verifying that those bounds are meaningful and/or make meaningful predictions about trained models--this is missing from this work.


Additional comments:

> we observe that the value function approximated by DNN can interpolate well but struggles to extrapolate

There are papers showing extrapolation capabilities in deep RL, see e.g. the work of Packer et al. [1]

[1] Assessing Generalization in Deep Reinforcement Learning, Charles Packer, Katelyn Gao, Jernej Kos, Philipp Krähenbühl, Vladlen Koltun, Dawn Song, 2018


> Theorem 1

I'm not sure I fully get why the authors think this theorem is important.

1. The NTK assumptions provide kernel-like bounds, and what we get is litterally that the further we get from the dataset the poorer the approximation, this is already known. There is some extra work to make the result a bound over the projection, but that is also fairly straightforward.
2. This result a priori has nothing to do with the convex hull of the training data. (3) and the equation before (2) are the same equation. The main argument seems to be that because the convex hull of a finite set of points is naturally bounded, then (2) is bounded whereas (3) is not. This argument seems moot in practice, since inputs to neural networks (or any ML system really) are almost always bounded (images lie in RGB[0,1], language is a sequence of tokens, etc).
3. The result relies on an Euclidean norm, which quickly loses significance past a number of dimensions. In fact, it is likely that past a certain number of dimensions, the very distinction between interpolation and extrapolation lose meaning (see [2]). Take 100k images from Atari at random, then take a new image by running an agent or again at random. I'd wager that either (a) the new image is within the convex hull, or (b) its distance to the convex hull is upper bounded by Eq (2)'s $B$, because $B$ is close to 1.

I'm also not sure what $x$ stands for here. The error on a $Q$ function evidently depends on whether not just the state but also the action is in the dataset. Is $x$ just the concatenation of the two? What does it mean to concatenate an Atari image and a categorical label and then take norms? What about joint angles and torsion strengths?

[2] Learning in High Dimension Always Amounts to Extrapolation, Randall Balestriero, Jerome Pesenti, Yann LeCun



> State-conditioned distance function

> synthetic noise actions sampled from the uniform distribution

This is a strange choice, why the sampling? We know that MSE regresses to the mean, and we know the mean of $\mathbb{E}_{x'\sim U}[|x-x'|]$. The result (regressing to the centroid) should be the same.

> It is clear that the learned distance function can accurately predict the sample-to-dataset centroid distance

In the action domain, yes, perhaps. But what about the state domain?

> To evaluate the generalization ability of DOGE, we remove small areas of data from the critical pathways to the destination

As above, this does not evaluate the right thing.

I am left wondering after reading this paper if a simpler baseline might be just as effective, one that doesn't require $g$. In the limit, but when each state only appears once in the training data with one associated action*, $g$ is simply the distance between $\mu(s)$ and $a$. What happens when we replace $g(s,a)$ by $\|a-\mu(a)\|$ in (6) and (7)?

*In a continuous state-action space, isn't every state really only "visited once" anyways?



**Summary Of The Paper:**

This paper proposes to regularize continuous control policies trained offline by forcing them away from actions that are too far from the training data's actions. This is achieved by a learned distance function that predicts distance to the training data's actions. The authors show that such a setup has some interesting properties. Empirically this appears to yield improvements on the D4RL offline benchmark suite.

**Summary Of The Review:**

I am a bit conflicted about the paper, the intuitions behind the ideas are nice. The paper spends a lot of time trying to convince us of the usefulness of a variety of bounds (often closely based on prior bounds), used to justify the approach, but I am skeptical that these bounds are reflected in the results. This is worsened by the fact that some predictions made by the theory are not properly tested in practice (see above).

Update: the authors have addressed my concerns, and I find the additions to the paper and the appendix are useful to framing this work. I lean in favor of acceptance.

---

> ### Author Response · Authors · 2022-11-13
> **Response to Reviewer kL5r (Part 4/4)**
>
> >**For the state-conditioned distance function...We know that MSE regresses to the mean, and we know the mean of Ex′∼U[|x−x′|]. The result (regressing to the centroid) should be the same.**
> - They are not the same. See Figure 3(a) as an example, the optimal state-conditioned distance function averages the V-shape distance curves with the bottom centered at each dataset action.
> Hence for actions close to the state-conditioned centroids, they will get lower $g^*$ values, while further away samples get larger values. This is also reflected in Property 1, that $g^*$ assigns more weights (larger $C(s,a)$ in the first equation of Eq.(5)) in data-dense regions. Moreover, note that under the special design of $g(s,a)$, actions falling within the state-conditioned convex hull of the dataset will generally get relatively small $g^*$ values, even if they are OOD actions, see Figure 3(b) for an empirical visualization.
> - In addition, our distance function enjoys another desirable property (Property 2) that the negative gradient of $g^*(s,\hat{a})$ at any extrapolated action $\hat{a}$ points inside the convex hull of the dataset. This means our distance function can serve as an ideal tool to enforce the convex-hull-based policy constraint.
>
> >**For the state-conditioned distance function...In the action domain, yes, perhaps. But what about the state domain?**
> - Please see **General Response 3**. Specifically, in offline RL, we care more about the state-conditioned action distance function because OOD action causes overestimation and training instability. Measuring the state distance has a limited contribution to policy learning and increases training difficulty.
>
> >**In a continuous state-action space, isn't every state really only "visited once" anyways?**
> - Yes.
>
> >**I am left wondering after reading this paper if a simpler baseline might be just as effective, one that doesn't require g. In the limit, but when each state only appears once in the training data with one associated action,  g is simply the distance between μ(s) and a. What happens when we replace g(s,a) by |a−μ(a)| in (6) and (7)?**
> - The simpler baseline the reviewer mentioned is TD3+BC, which has been thoroughly compared in our experiments, see Table 1, Figure 4, Table 3, and Figure 8 (Figure 7 in the original version) in our paper. Our method outperforms TD3+BC by a large margin and demonstrates much better generalization performance.
> - Our method does not degenerate to TD3+BC in continuous MDPs. The following are detailed reasons:
>     - $g$ will not simply become a sample-to-sample distance function because $g$ is approximated by deep neural networks, which does not simply memorize each single training sample but fits the overall pattern of the underlying dataset. Figure 3(b) shows that $g$ can capture the global geometric pattern when each state-action pair appears only once, which exactly addresses the reviewer's concern.
>     - Simply replacing $g$ with $|a-\mu(s)|$ as in TD3+BC can only provide local information for each sample, which may be noisy. However, $g$ is trained on the entire dataset, which can provide the global geometric information for policy learning, just as discussed in Section 3.1.
> - To better illustrate the superiority of DOGE over TD3+BC, we add more comparisons with TD3+BC on a new set of mixed-quality datasets. In Halfcheetah-random dataset, we add different proportions (1% to 20%) of near-optimal Halfcheetah-medium-expert dataset to form new mixed datasets. The detailed results and discussion are included in Appendix F.2 in our revised paper. We also summarize the results in the following Table. We can see that DOGE enjoys substantial performance gains when the random dataset involves near-optimal data, while TD3+BC is heavily influenced by the local information from the larger proportion of low-quality random data. This can also be reflected from the learning curves of TD3+BC and DOGE in Figure 10 of Appendix F.2 in our revised paper, that TD3+BC suffers from severe oscillation and training instability, while DOGE enjoys a stable training process due to the use of policy constraints that captures the overall dataset geometry.
>
> ||Halfcheetah-r (Ratio=0\%)|Ratio=1\%|Ratio=3\%|Ratio=5\%|Ratio=10\%|Ratio=20\%|
> |---|---|---|---|---|---|---|
> |TD3+BC|11.0|23.2|24.2|47.2|48.5|48.1|
> |DOGE|17.8|38.9|49.4|50.0|51.0|54.4|
>
> >**I could probably reproduce this paper, perhaps not exactly, but certainly in spirit given the level of detail.**
> - We have uploaded our codes in the supplementary material. The reviewer can reproduce our results in this paper based on the instructions.
>
>
> >**For the issue about our algorithm name...**
> - We are sincerely sorry for the controversial algorithm name and we will take it seriously in our future works. Thank the reviewer for the suggestion!

---

> ### Author Response · Authors · 2022-11-13
> **Response to Reviewer kL5r (Part 3/4)**
>
> >**For Theorem 1... I'm not sure I fully get why the authors think this theorem is important... what we get is litterally that the further we get from the dataset the poorer the approximation, this is already known.**
> -  Although the conclusion in Theorem 1 is already observed in some supervised learning studies, it is largely ignored in modern offline RL studies. Most existing offline RL methods tend to treat all OOD actions equally without considering the generalization ability of the Q network on these samples, making their algorithms over-conservative. Theorem 1 in this paper offers a theoretical explanation for our empirical findings and serves as a starting point to motivate the design of our proposed algorithm DOGE.
>
>
> >**For Theorem 1...This result a priori has nothing to do with the convex hull of the training data...**
> - We agree with the reviewer that neural networks are usually bounded since inputs to neural networks are almost always bounded. However, Theorem 1 conveys an important insight that the distance to dataset matters and the interpolation data enjoy tighter bounds compared with extrapolated data, especially when the dataset only narrowly covers a large state-action space. Such cases are common in many practical high-dimensional offline RL tasks with partially covered datasets, which are extremely challenging to solve for existing offline RL methods. The finding in Theorem 1 serves as a motivating starting point for us to consider the sample-to-dataset distance and data geometry when designing our offline RL algorithm DOGE.
>
> >**For Theorem 1...The result relies on an Euclidean norm, which quickly loses significance past a number of dimensions...**
>
> - Measuring the proper distance in the high dimensional setting is definitely an important topic. It is possible to use some representation learning techniques to learn a low-dimensional embedding and then apply the same analysis. We thank the reviewer for pointing this out, which is surely an interesting direction for us to explore in the future.
> - For our practical algorithm DOGE, the Euclidean norm in our paper can actually be extended to other valid distance metrics or combined with representation learning without changing the overall framework of our algorithm. We implement the Euclidean norm mainly for simplicity and it has already exhibited good performance in offline RL benchmark tasks.
>
> >**For the state-conditioned distance function...Synthetic noise actions sampled from the uniform distribution. This is a strange choice, why the sampling?**
> - We first briefly describe why we need to learn such a distance function. Based on Theorem 1, we can see that the sample-to-dataset distance plays an important role in measuring the controllability of Q values. However, given an arbitrary state-action sample $(s,a)\in S\times A$, naively computing its distance to the closest data point in a large offline dataset can be prohibitively expensive and impractical. Therefore, it is ideal to have a learnable distance function to cheaply measure such sample-to-dataset distance while also being capable of capturing the overall dataset geometry.
> - Then, how can we achieve this in a learnable fashion? For a single state-action pair in the dataset, i.e., $(s,a)\sim\mathcal{D}$, we can uniformly sample some random actions $\hat{a}$ over the entire action space, and their distances to action $a$ form a V-shape curve with the bottom at the dataset action $a$ as the blue dash lines shown in Figure 3(a). If we combine all such V-shape curves for every state-action pair in the dataset $\mathcal{D}$, we have our final state-conditioned distance function $g(s,a)$. This is exactly what the optimization problem Eq.(4) is doing, which finds a linear combination of all V-shape distance curves and outputs the upper bound of the state-conditioned sample-to-centroid distance of an arbitrary action $a$. Sampling random action is the cheapest way to approximate such V-shape curves in a sample-based manner.
> - The reason that we sample from a uniform distribution is that sampling from uniform distribution essentially puts equal weight on the action space, which leads to the correct distance measures under Euclidean space as well as a linear combination form that preserves distance convexity. Sampling from other distributions may introduce non-linearity and non-convexity into the distance function, which corresponds to some other distance measures under a distorted space.

---

> ### Author Response · Authors · 2022-11-13
> **Response to Reviewer kL5r (Part 2/4)**
>
> >**It's nice to have bounds, but what's even nicer is empirically verifying  that those bounds are meaningful and/or make meaningful predictions about trained models--this is missing from this work.**
> - We are sorry for missing some direct validation of the theoretical bounds in our paper. For Theorem 1, the empirical results in Figure 2 is a clear evidence of its conclusions. For Theorem 2, we do not construct the empirical experiments because accurately calculating the bellman-consistent coefficient (Eq.(8)) is pretty hard, which requires evaluating the bellman error at distribution $v$ and solving an optimization problem. For Theorem 3, although some values such as $\alpha(\Pi_{\mathcal{B}})$ and $L(\Pi_{\mathcal{B}})$ are not easily obtainable, the insights from these quantities are directly transferable to the design logic of our proposed algorithm DOGE (please refer to General Response 2 and Appendix A for detailed discussion). We have empirically demonstrated the improved performance of DOGE on a series of experiments, which is consistent with the findings from our theories.
>
> >**There are papers showing extrapolation capabilities in deep RL, see e.g. the work of Packer et al.**
> - We thank the reviewer for the reference, and we have cited it in our revised paper. However, note that the concept of extrapolation in Packer et al. is different from our paper. In Packer et al., the extrapolation data are defined as those fall outside the training distribution (e.g. data from a different environment), whereas in our paper, the extrapolation data are defined more from the pure geometry perspective, which is the state-action pairs fall outside the convex hull formed by the offline dataset. Hence the concepts of extrapolation between Packer et al.'s paper and our paper are not the same. Second, Packer et al.'s study focuses on the online setting, where the agent can perform arbitrary exploration in the training environment. However, our analysis is conducted in the offline setting, where it is not possible to perform trial-and-error to gather information from OOD state-action pairs. Lastly, the evaluation of interpolation and extrapolation performance in Packer et al. is based on whether to use the same or different environment during testing, which is completely different from our problem setting. Consequently, due to these differences, we do not think the extrapolation conclusion provided by Packer et al. is directly transferable nor related to our work.
> - What we want to show in our paper is that existing offline RL methods largely overlook information from the geometry of offline datasets as well as the generalization benefits from deep neural networks, thus resulting in too much conservatism. By contrast, DOGE is designed from the perspective of generalization performance of the deep Q function and tries to relax such over-conservatism.

---

> ### Author Response · Authors · 2022-11-13
> **Response to Reviewer kL5r (Part 1/4)**
>
> We thank the reviewer for the thoughtful review and comments, which is very helpful to improve the clarity of our paper. Regarding the concerns from the reviewer, we provide the detailed responses as follows:
>
> >**The text is hard to read at times**
> - We are sincerely sorry for this.
>
> >**The theory within the paper relies heavily on quantities which may not scale intuitively**
> - We're sorry for the confusion in our theoretical analyses. Please see the General Responses 1 and 2 as well as Appendix A in our revised paper for an elaborated discussion on the overall logical flow of our paper.
>
> >**The core of the proposed method is to contrain a (continuous) policy to output actions that are not "too far" from the training data's actions.**
> - The core is not simply to constrain a policy to stay near training data's actions, but rather stay inside or near the state-conditioned convex hull induced by the dataset. This corresponds to an expanded feasible region that covers more generalizable OOD actions. This brings more chances to find the optimal policy as well as enable the learned policy to generalize better on OOD data.
>
> >**Yet, the main way this is tested is to remove part of the state space from the training data? This is not testing the right thing, and is not even testing what the theory predicts that this method should be better at.**
> - In our Antmaze with missing data experiments, we are not only removing part of the state space of the training data but rather, we remove the state-action tuples altogether in the joint state-action space. This actually corresponds to a harder case for generalization, since both state and action information is missing. It is often hard (sometimes impossible) to construct a test that preserves the full state space but only removes part of the action space. As in many tasks, the states and actions are bounded together in an MDP, solely removing a set of actions will make some states never reachable.
> - We construct the Antmaze experiments to test whether exploitation of OOD actions can help the policy generalize in OOD regions and remedy the missing transitions. Figure 4, Table 3, and Figure 8 (Figure 7 in the original paper) show that the previous over-conservative offline RL methods all fail miserably but our approach can generalize well, which exactly validates our idea.

---

> > ### Comment · Reviewer_kL5r · 2022-11-14
> > **Follow up**
> >
> > Thank you for your thorough response. I think I understand some aspects of the paper better now.
> >
> > I'm still unconvinced by the pertinence of Theorem 1, at least the way it is presented, as a justification for the method. In particular, I understand the argument based on distance (which is ultimately what $g$ is), but I do not understand how the argument relates to convex hulls, extrapolation, nor interpolation.
> >
> > Here, I think, quantitative validations of the latter (convex hulls, extrapolation, nor interpolation) would change my mind (although given my understanding of generalization in deep learning, I suspect this will not yield anything interesting and distance will be the main explanatory factor). To be clear, Figure 2 is a qualitative result, and even it does not really convince me of the pertinence of notions beyond distance.
> >
> > I'm still inclined to accept this paper, but I do think it would benefit from clearer quantitative evidence that explains the proposed method's success.
> >
> > Some direct responses:
> >
> > >  As in many tasks, the states and actions are bounded together in an MDP, solely removing a set of actions will make some states never reachable.
> >
> > I'm not sure I understand this passage; in a finite dataset in batchRL, it's typically the case that not every state or (s,a) pair is there in the first place. If the function approximator generalizes well, then we've solved the problem. What is the relationship with reachability here?
> >
> > >  [Packer et al.], ..., whereas in our paper, the extrapolation data are defined more from the pure geometry perspective, which is the state-action pairs fall outside the convex hull formed by the offline dataset.
> >
> > I agree that the setup is different, just wanted to point to other extrapolation-capable DeepRL results. If you consider the environment parameters as part of the state (and thus there is only just one giant MDP being solved), then it is indeed OOD extrapolation within the same MDP when e.g. the gravity constant changes.
> >
> > > Theorem 1 conveys an important insight that the distance to dataset matters and the interpolation data enjoy tighter bounds compared with extrapolated data, especially when the dataset only narrowly covers a large state-action space. Such cases are common in many practical high-dimensional offline RL tasks with partially covered datasets, which are extremely challenging to solve for existing offline RL methods.
> >
> > The claim that such cases (of interpolation vs extrapolation) are common does not appear true to me, at least not trivially true. I think it would make sense to have this quantitatively measured. For example, when removing a chunk of the state space in AntMaze, can you relate the size of the chunk (or something similar) to a quantity in Theorem 1?

---

> > > ### Author Response · Authors · 2022-11-16
> > > **Response to the follow up of Reviewer kL5r**
> > >
> > > Thank you for the time to engage in the discussion! Your further suggestions will be of great help in improving the quality and clarity of our paper.
> > >
> > > For the concern of the reviewer on the quantitative verification of Theorem 1 (especially the relevance of interpolation and extrapolation), we have added more quantitative experiments on MuJoCo-medium tasks in Appendix F.8 in the updated revision. We empirically evaluate the value of $\\|Q_\theta (x)-Q_\theta ({\rm Proj}_\mathcal{D}(x))\\|$ and analyze its relationship to the sample-to-dataset distance. The additional results in Appendix F.8 are consistent with Theorem 1. The upper bound on the value difference of the deep Q function is well controlled by the distance to dataset, and interpolation samples enjoy a tighter upper bound than most extrapolation samples (those that deviate far from the dataset).
> > >
> > > Belows are some direct responses to other comments:
> > > >**I'm not sure I understand this passage; in a finite dataset in batchRL, it's typically the case that not every state or (s,a) pair is there in the first place. If the function approximator generalizes well, then we've solved the problem. What is the relationship with reachability here?**
> > > - What we want to express in our previous response is that, it is generally not possible to construct an evaluation task, to only remove some part of the action space but remain the original state space well-defined in an MDP. For example, consider a 2D-navigation task, if all the upward direction actions are removed, then all states lying above the initial location become unreachable according to the MDP defined on the modified action space. Hence we choose to remove the joint state-action tuples together to evaluate the model generalization performance.
> > >
> > > >**The claim that such cases (of interpolation vs extrapolation) are common does not appear true to me, at least not trivially true. I think it would make sense to have this quantitatively measured.**
> > > - We thank the reviewer for the suggestion. We have added the quantitative experiments in Appendix F.8 of our updated paper. For the Antmaze example mentioned by the reviewer, the states in Antmaze tasks not only contain the 2D-position of the ant, but also its joint information (the ants also need to learn to walk). Therefore, it is difficult to determine whether the removed parts can be interpolated or extrapolated. Therefore, we give more alternative and quantitative experiments on the MuJoCo tasks, and please check Appendix F.8 for detailed results.

---

> > > ### Author Response · Authors · 2022-11-22
> > > **Follow up**
> > >
> > > Dear reviewer kL5r,
> > >
> > > >Here, I think, quantitative validations of the latter (convex hulls, extrapolation, nor interpolation) would change my mind.
> > >
> > > Thank you again for the time to engage in the discussion! Your further suggestions will be of great help in improving the quality and clarity of our paper.  We would really appreciate it if you could tell us whether our additional quantitative experiments on Theorem 1 solve your concerns.  We would be more than happy to engage in further discussions!
> > >
> > > Thanks!

---

> ### Author Response · Authors · 2022-12-06
> **Thanks for the detailed review and discussions**
>
> Dear reviewer kL5r,
>
> We would like to thank you again for your time engaged in the review and discussion phase! Your constructive comments do a great help to improve the quality of our paper. We're glad our responses have addressed your concerns and your favor of acceptance of our paper.
>
> Thanks!
>
> Best regards,
>
> Authors of Paper 3842

---

### Official Review · Reviewer_PeHF · 2022-10-25

**Confidence:** 3
**Correctness:** 3
**Technical Novelty And Significance:** 4
**Empirical Novelty And Significance:** 3
**Recommendation:** 6

**Clarity, Quality, Novelty And Reproducibility:**

My major clarity concerns are on Theorem 2 and Theorem 3. Theorem 2 is stated under the assumption of NTK, what is the exact meaning of that? Second, the $\epsilon_\mu$ terms bothers me. From the definition, it is the bellmen error term measured by distribution $\mu$. If the policy $\pi$ is good enough, then the bellman-consistent coefficient blows up. How should one interpret this? It might be better to provide a concrete rate of convergence in Theorem 3 and compare to existing literature. Both Theorem 2 and Theorem 3 does not directly reflect how to choose threshold $G$ in equation (6).

"Minimax-Optimal Off-Policy Evaluation with Linear Function Approximation" might also be a related work, where the distribution shift is measured by a class-restricted $\chi$-square divergence.

**Strength And Weaknesses:**

================= Strength =================

The paper focuses on a very important problem in RL: when offline data do not have strong data coverage, how to best exploit the logged data set. The proposed method is novel to me.

The paper is clearly written with many graphical illustrations. Most of the sections are easy to follow.

Empirical results are convincing, although I am not totally familiar with the experimental environments.

================= Weakness =================

The claim "Theoretical analysis demonstrates the superiority of our approach to existing methods that are solely based on data distribution or support constraints" may not be well supported, as in Section 3.3, the idea is to transform the policy constraint to Bellman-consistent coefficient. Moreover, Theorem 3 is a bit hard to interpret as no concrete rate is given. It would be helpful to elaborate the claim.

Theoretical results in Theorem 2 have a relatively vague connection to DOGE. At a first glance, the constraint threshold $G$ in equation (6) does not appear in Theorem 2 nor Theorem 3, making it unclear how to evaluate the advantage of DOGE. At the same time, Theorem 2 is still hard to interpret because of the complicated math expressions without intuitive explanations.

**Summary Of The Paper:**

Motivated by the over conservativity of existing offline RL algorithms, the paper proposes a Distance-sensitive Offline RL with better GEneralization (DOGE) method. In fact, DOGE describes a constraint set of possible policies, relying on a state-conditioned distance function. Such a distance function accounts for the data geometry. Theoretically, a connection between DOGE and bellman-consistent coefficient is established. Empirically, DOGE is demonstrated to have comparable or better performance versus the state-of-the-art methods.

**Summary Of The Review:**

I am interested to see any feedback from the author regarding the theory part of the paper. Overall, I think the paper in its current form is slightly below the acceptance bar, but I am willing to raise the rating based on further discussions.

---

> ### Author Response · Authors · 2022-11-13
> **Response to Reviewer PeHF (Part 2/2)**
>
> >**It might be better to provide a concrete rate of convergence in Theorem 3 and compare to existing literature.**
>
> - For existing practical offline RL algorithms, the most relevant theoretical result to our paper is BEAR (Kumar et al., 2019), which is built on support constraints. In Theorem 3, we have already compared our method with BEAR, and show that DOGE enjoys smaller error propagation value $L(\Pi_\mathcal{B})$ and suboptimality constant $\alpha(\Pi_\mathcal{B})$ due to allowing exploitation on generalizable OOD areas. Please refer to the discussion paragraph after Theorem 3 for details.
> - We're sorry for missing the convergence rate for Theorem 3. When the policy and Q function are approximated by deep neural networks, proving the convergence rate is one major challenge in deep RL and requires many non-trivial assumptions [1-2]. The difficulty is further exacerbated under offline RL setting, as the dataset properties (e.g., state-action space coverage) also impact the performance bound of offline RL policies, thus very few practical offline RL algorithms can derive concrete convergence rates. We will leave it as future work in our study.
>
>
> >**Both Theorem 2 and Theorem 3 does not directly reflect how to choose threshold G in equation (6).**
> - Theorem 3 does bring intuitive guidance for the choice of G. Theorem 3 suggests that a trade-off between policy improvement and conservatism should be balanced. So, the threshold $G$ in DOGE should not be too small or too large. As Theorem 3 states, the performance bound is related to $L(\Pi_{\mathcal{B}})$ and $\alpha(\Pi_{\mathcal{B}})$, where $L(\Pi_{\mathcal{B}})$ will be large when the learned policy deviates far away from dataset centroid, while $\alpha(\Pi_{\mathcal{B}})$ becomes large when the policy set is too small to cover the optimal policy. Therefore, a trade-off between policy improvement and conservatism should be considered when choosing policy constraint strength, while $G$ in DOGE just plays this role. Meanwhile, Table 7, Figure 14 and Figure 15 in Appendix F.5 (revised version of the paper) show that a small G (G=30%) will cause the feasible region of the constraint problem to be too small to contain the good solution, while an overly loosed G (G=90%, 100%) increases the risk of Q-value overestimation and thus suffers from performance drop and high variance. In our main results, we set $G=50\%$ non-parametrically based on the state-conditioned distance values of the training dataset, which means the distance function output on policy-generated actions should be controlled at least to the average level of distance function outputs on dataset samples. This is neither too conservative nor too aggressive for most tasks, which agrees with the insight from Theorem 3 and also offers reasonably good performance.

---

> > ### Author Response · Authors · 2022-11-13
> > **References**
> >
> > [1] Fan, Jianqing, et al. "A theoretical analysis of deep Q-learning." Learning for Dynamics and Control. PMLR, 2020.
> >
> > [2] Cai, Qi, et al. "Neural temporal-difference learning converges to global optima." Advances in Neural Information Processing Systems 32 (2019).

---

> ### Author Response · Authors · 2022-11-13
> **Response to Reviewer PeHF (Part 1/2)**
>
> We appreciate the reviewer for the constructive comments on our paper, which is very helpful to improve the clarity of our paper. Regarding the concerns from the reviewer, we provide the detailed responses as follows:
>
> >**Theoretical results in Theorem 2 have a relatively vague connection to DOGE... Theorem 2 is stated under the assumption of NTK, what is the exact meaning of that?**
> - Please see **General Response 1** and **2** for a detailed explanation of the connection between DOGE and the theoretical results.
> - For the intuitive explanation and meaning of Theorem 2, the key parts of the RHS of Eq (10) in Theorem 2 are $\mathcal{B}_2$ and $\mathcal{B}_3$. Both $\mathcal{B}_2$ and $\mathcal{B}_3$ are closely related to the sample-to-centroid distances $d_1$ and $d_2$, which can be minimized by forcing the state-conditioned distance function $g(s,a)$ to be small values because the optimal $g(s,a)$ exactly outputs the upper bound of sample-to-centroid distances according to Property 1 in Section 3.1. As DOGE is constructed by constraining $g(s,a)$ to be small during policy optimization, therefore, Theorem 2 guarantees that DOGE enjoys a tight upper bound of the Bellman-consistent coefficient, and so the policy set induced by DOGE is essentially a Bellman-consistent constrained policy set defined in Definition 2. This further leads to Theorem 3, which implies a tighter performance bound of policy learned by DOGE as compared to prior offline RL methods with data support constraints such as BEAR.
> - Our paper is motivated from exploiting the generalization behavior of Q functions approximated by deep neural networks. NTK is one of the most popular analytical tools for the generalization analysis of deep neural networks, thus is also adopted in our paper. The detailed description of the NTK assumption is elaborated in Appendix B.2, Assumption 1.
>
> >**The ϵμ terms bothers me. From the definition, it is the bellmen error term measured by distribution μ. If the policy π is good enough, then the bellman-consistent coefficient blows up. How should one interpret this?**
>
> - The $\epsilon_\mu$ term is the lower bound of the squared Bellman error for $\pi$ under distribution $\mu$. In the offline RL setting, we want to learn an optimal policy from data rather than strictly imitating the behavior policy $\mu$. If the policy $\pi$ is good enough, it is reasonable to expect some degree of deviation from $\mu$ in data, unless $\mu$ itself is already the optimal expert policy. The discrepancy of the learned $\pi$ and behavior policy $\mu$ will cause the $\epsilon_\mu$ not possible to take very small values. On the other hand, having an offline dataset induced by an optimal expert policy is rare in practical offline RL settings, and if we already have an expert policy-induced dataset, we could directly use BC to learn the optimal policy rather than resort to the more costly offline RL. Lastly, even if the policy $\pi$ is learned to be very similar to $\mu$, it is still reasonable to assume $\epsilon_\mu$ can be lower bounded by a small value because of the existence of Q network approximation error, such error is almost unavoidable in practical deep RL algorithms. Therefore, in all different cases, the $\frac{1}{\epsilon_\mu}$ term in the Bellman-consistent coefficient will not be too large, especially when we use deep neural networks as function approximators.

---

> ### Author Response · Authors · 2022-11-22
> **Follow up**
>
> Dear reviewer PeHF,
>
> >I am interested to see any feedback from the author regarding the theory part of the paper. Overall, I think the paper in its current form is slightly below the acceptance bar, but I am willing to raise the rating based on further discussions.
>
> Thank you for the detailed comments and suggestions to improve our paper! Regarding your concerns, we have added the discussions about the theoretical logic flow of our paper and more experiments in the updated paper. We would really appreciate it if you could tell us whether your concerns are resolved. Your further comments and discussions will greatly help to improve the quality of our paper and we will be more than happy to engage in further discussions and address them.
>
> Thanks！

---

> > ### Comment · Reviewer_PeHF · 2022-12-05
> > **Thank you for the detailed response**
> >
> > Many of my concerns are addressed in the revision and response. The $\epsilon_\mu$ term now makes sense to me.
> >
> > I would like to follow up on the NTK assumption, and the theories derived under it. The NTK assumption is stated in appendix --- infinitely wide two-layer neural networks trained with gradient flow (infinitesimal learning rate). In this case, the dynamics of neural networks can be well approximated by a linear model, the so-called neural tangent kernel. How does this setting deviate from Q-learning with linear function approximation? In linear function approximation, pessimistic offline policy learning algorithm has demonstrated its power without strong coverage assumption. What is the advantage of the method in the paper compared to pessimism? As there is no concrete rate provided in the paper, it is difficult to tell the advantage directly.
> >
> > From my understanding, the paper proposed a distance-based method to address the data coverage problem. By introducing a constraint, the policy space is restricted to exclude extreme ones. This idea is very interesting and empirical results illustrate the strength of the method. However, the theory part somewhat undermines its value due to the NTK (linear) assumption. In addition, as pointed out in the response, how to choose threshold $G$ is implicit in the paper. While in pessimism, an explicit bonus function can be computed to guide the policy learning, as a way to restrict the policy searching space.

---

> > > ### Author Response · Authors · 2022-12-06
> > > **Thanks for your further constructive comments!**
> > >
> > > Dear reviewer PeHF,
> > >
> > > We would like to thank the reviewer for the further constructive comments and your time engaged in the discussion phase! Regarding your concerns on the advantages of NTK over the linear function approximation of the pessimism paper [3], we provide the detailed responses as follows.
> > >
> > >
> > > - Theoretically:
> > >
> > >     - **Although the NTK assumptions (infinite width and infinitesimal learning rate) may seem strong, simple deep neural networks can also replicate their behavior reasonably well.** In fact, Section 6.2 of the original NTK paper [4] ablates the effects of network widths with a large learning rate (learning rate 1.0) in a regression task and finds that NTK gives good indications even for relatively small widths networks (50 units per layer). Many other papers, such as [5][6][7], build on NTK assumptions and also observe great alignments between theory and experiments using simple networks. Importantly, the quantitative experiments in Appendix F.8 of our paper are also consistent with Theorem 1, which further suggest the soundness of our NTK assumptions.
> > >
> > >     - **On the other hand, the pessimistic value iteration [3] relies on another set of strong assumptions about linear MDP and imposes tricky requirements on the choice of $\xi$-uncertainty quantifier under general MDP**. In the pessimism offline policy learning paper [3], a closed form of the $\xi$-uncertainty quantifier (the key ingredient to introduce pessimism and to reduce spurious correlation) can only be obtained under linear MDP, which requires not only the linear forms of Q-functions but also linear forms of reward functions and the transition dynamics (Section 4.2 in [3]). These are rather strong assumptions and unrealistic in many tasks. Under general MDP, however, the form of $\xi$-uncertainty quantifier remains unclear (Section 4.1 in [3]). No guidance is available to estimate a valid $\xi$-uncertainty quantifier in many tasks and hence bringing challenges to the parameter tuning.
> > >
> > >
> > > - Practically:
> > >     - As stated in the response of W2 to reviewer hPVe, practical offline RL algorithms that follow the pessimistic value iteration framework [3], including EDAC [8], PBRL [9], SCORE [10] and RORL [11], suffer some main drawbacks, such as **computational expensive uncertainty estimation and sensitivity to hyperparameters tuning**, since the form of $\xi$-uncertainty quantifier under general MDP is unknown. Please see our response of W2 to reviewer hPVe for detailed practical comparisons of DOGE over pessimistic offline policy learning methods.
> > >
> > >
> > > - For the choice of G:
> > >     - **It should be noted that we set the threshold of G as 50\% for all comparative tasks in our paper without tuning** (12 mujoco + 6 antmaze + 4 generalized antmaze + 5 comparison tasks with TD3+BC). We suggest the practical implementation of our algorithm can simply adopt 50% by default, which generally leads to good performance. Although the exact threshold choice of G is not explicitly implied in our theories, the intuition is actually quite straightforward, that is, **the policy-generated actions should be controlled to the average level of distance function outputs on dataset samples**, hence will not incur too conservative nor too aggressive regularizations. Figure 5 in Section 4.3 and Table 7, Figure 14 in Appendix F.5 also validate the robustness of different threshold choices of G (using values of 50\%~70\% consistently achieve good performance).
> > >
> > >
> > > Hope our responses address your additional concerns! Thank you again for your time and valuable comments!
> > >
> > > Best regards,
> > >
> > > Authors of Paper 3842

---

> > > > ### Author Response · Authors · 2022-12-06
> > > > **Additional References**
> > > >
> > > > [3] Jin, Ying, Zhuoran Yang, and Zhaoran Wang. "Is pessimism provably efficient for offline rl?." International Conference on Machine Learning. PMLR, 2021.
> > > >
> > > > [4] Jacot, Arthur, Franck Gabriel, and Clément Hongler. "Neural tangent kernel: Convergence and generalization in neural networks." Advances in neural information processing systems 31 (2018).
> > > >
> > > > [5] Arora, Sanjeev, et al. "On exact computation with an infinitely wide neural net." Advances in Neural Information Processing Systems 32 (2019).
> > > >
> > > > [6] Xu, Keyulu, et al. "How Neural Networks Extrapolate: From Feedforward to Graph Neural Networks." International Conference on Learning Representations. 2020.
> > > >
> > > > [7] Bietti, Alberto, and Julien Mairal. "On the inductive bias of neural tangent kernels." Advances in Neural Information Processing Systems 32 (2019).
> > > >
> > > > [8] An, Gaon, et al. "Uncertainty-based offline reinforcement learning with diversified q-ensemble." Advances in neural information processing systems 34 (2021): 7436-7447.
> > > >
> > > > [9] Bai, Chenjia, et al. "Pessimistic Bootstrapping for Uncertainty-Driven Offline Reinforcement Learning." International Conference on Learning Representations. 2021.
> > > >
> > > > [10] Deng, Zhihong, et al. "SCORE: Spurious COrrelation REduction for Offline Reinforcement Learning." arXiv preprint arXiv:2110.12468 (2021).
> > > >
> > > > [11] Yang, Rui, et al. "RORL: Robust Offline Reinforcement Learning via Conservative Smoothing." NeurIPS 2022.

---

> > > > ### Comment · Reviewer_PeHF · 2022-12-06
> > > > **Increase to a 6**
> > > >
> > > > Thank you for the additional response!
> > > >
> > > > I am happy to increase my rating to a 6. I agree with the authors that the method proposed in paper can be generally applied to MDPs, far beyond linear MDPs or those with linear function approximation.

---

> > > > > ### Author Response · Authors · 2022-12-06
> > > > > **Thanks for the detailed review and for increasing the score!**
> > > > >
> > > > > Dear reviewer PeHF,
> > > > >
> > > > > Thank you so much for your detailed review and for raising your score from 5 to 6! We really appreciate it! We'll add these discussions about the advantages over linear function approximation in our final version. Your comments helped a lot to improve the quality and clarity of our paper!
> > > > >
> > > > > Thanks!
> > > > >
> > > > > Best regards,
> > > > >
> > > > > Authors of Paper 3842

---

### Official Review · Reviewer_z7dd · 2022-10-30

**Confidence:** 4
**Clarity, Quality, Novelty And Reproducibility:** The paper is easy to follow, and the …
**Correctness:** 2
**Technical Novelty And Significance:** 2
**Empirical Novelty And Significance:** 2
**Recommendation:** 3

**Strength And Weaknesses:**

Strength: The idea seems interesting, and new, and certainly improves over policy constraints in their experiments (TD3+BC vs their method).

Weaknesses:

- I think I don't understand why the approach should perform well in general -- doesn't it depend on what kind of a function we train for g(a, s)? What if g(a, s) were a table, would the method not be just TD3+BC? But the paper doesn't seem to discuss the connection between the complexity of g(s, a) and the efficacy of the method.

- Taking the above reasoning, shouldn't a value function conservatism method like CQL already take into account the geometry of the data? Why should the proposed method be better than that, if it is not just for tuning issues?

- The theory doesn't seem to discuss the role of g and the complexity of g. It builds on the Bellman-consistent pessimism paper, yet no comparisons to the approach from that paper and their theoretical results exists in the paper.

- Empirical results: besides benchmark results, more clear ablations and analysis are needed to understand why the method actually works. Can we do ablations on the network architectures for both TD3+BC and CQL, and the proposed approach to see how all of these behave with smoothing?

**Summary Of The Paper:**

This paper proposes a method for accounting for the geometry of the data in devising methods for enforcing pessimism in offline RL. To do so, the paper ends up training a state-conditioned distance function between actions, and enforces a policy constraint using the learned distance function. The hope is that this is useful when the distance function can generalize (though I have questions about this below).

**Summary Of The Review:**

Overall, I am not convinced that the method is robust and reliable unless the concerns of g(s, a) are discussed. The theoretical analysis doesn't explain why g(s, a) should be good in practice or why alternatives such as value conservatism are not doing the same job. More empirical evidence and improved theoretical analysis is needed.

-------

## After Rebuttal

Thanks for the responses! I revisited Theorem 2, but I still don't see why $d_1$ and $d_2$ alone control the the Bellman concentrability coefficient tightly. There are many more terms in there that depend on the Q-function class in Theorem 2 ($Q(s_0, a_0)$, $\epsilon_\mu$), and in general, it cannot be said that the sample to centroid distance is alone what it takes to control this distributional shift. This just makes me think further that the approach would better benefit from a "geometry constraint" on the Q-function class (as it would more tightly control the Bellman concentrability coefficient) as opposed to the continuous action space distance constraint.

Regarding comparison with TD3+BC conceptually -- Sorry, I am still not sure. In equation 4, it seems like $g(s, \hat{a}) = ||\hat{a} - a_\text{data}||_2$ at the optimum. This means that the learned function when Equation 4 is solved exactly is measuring the L-2 distance to the dataset action (assume for a second that there is only one action at a given state in the dataset). Then isn't it identical to TD3+BC? When multiple actions are observed for a given state in the dataset, one could argue that even TD3+BC would have a similar global effect, as you would train $\pi(s)$ to be close to $a_1$, $a_2$, ... for all actions observed in the data, which would achieve the same kind of a smoothing effect as in Figure 3 (a).

It seems like the author's intuition goes beyond a state-conditioned function to some smoothing over the dataset more generally. But this is not clearly explained, and it is not clear why other methods and TD3+BC, most directly, will also not enjoy some kind of global smoothing effect more generally with a small function class.

So I don't understand how is this true  _"In TD3+BC, the BC term simply minimizes the absolute sample-to-sample distance between actions from the policy and the dataset, which essentially avoids the policy from producing OOD actions. Whereas in DOGE, the state-conditioned distance function  characterizes a sample-to-dataset distance that captures the overall data geometry. "

Overall, I am inclined to keep my original score. Since my score is at a disagreement from other reviewers, I will reduce my confidence by 1.

---

> ### Author Response · Authors · 2022-11-13
> **Response to z7dd W3~W4**
>
> >**W 3.1. The theory doesn't seem to discuss the role of g and the complexity of g**
>
> - Please see **General Response 2** for the role of $g$ in the our theoretical analysis. Briefly, $g(s,a)$ outputs the upper bound of the sample-to-centroid distance, which is closely related to $d_1$, $d_2$ in Theorem 2 and $L(\Pi_\mathcal{B})$ in Theorem 3. Therefore, $g$ does play a key role in our theory. In our revised paper, we also provide the learning curves as well as ablations on the expressivity of the state-conditioned distance function $g(s,a)$ in Appendix F.4.
>
> >**W 3.2. It builds on the Bellman-consistent pessimism paper, yet no comparisons to the approach from that paper and their theoretical results exists in the paper.**
> - The original Bellman-consistent pessimism paper (Xie et al., 2021a) only provides a conceptual algorithm and not possible to be practically implemented. Their algorithm and bounds rely on non-calculable values such as function class complexity, which is hard to evaluate for deep neural networks. By contrast, our performance bound builds on more achievable and intuitive values such as the sample-to-centroid distance in Theorem 2. This can also lead to a simple practical algorithm DOGE, based on the insight that we can minimize such distance value to achieve tightened performance bound.
>
> >**W4. Can we do ablations on the network architectures for both TD3+BC and CQL, and the proposed approach to see how all of these behave with smoothing?**
>
> - As we show in Table 2 in Appendix E, the Q and policy network architectures in DOGE are implemented with simple MLPs, which have the same network architecture as TD3+BC and CQL for a fair comparison. In DOGE, we learn an extra state-conditioned distance $g(s, a)$ network, which is also a simple MLP. We provide the learning curves of $g$ in Figure 11 (Appendix F.4), which show that the learned distance function is easy to train as it is learned through simple supervised learning. Moreover, we also show that DOGE is robust to the training quality of $g$ (see ablation on the different network configuration of $g$ in Table 6, Appendix F.4) as well as the random sample $N$ used for distance training (see Figure 5(c) in the main text and Figure 13 in Appendix F.5 for detailed ablation).
> - We are not sure about what "smoothing" exactly refers to mentioned by the reviewer in this comment. We are happy to address your concern if the reviewer could explain more about it.

---

> ### Author Response · Authors · 2022-11-13
> **Response to z7dd W1~W2**
>
> We thank the reviewer for the detailed comments and suggestions to improve our paper. Regarding the concerns from the reviewer, we describe the detailed responses as follows:
>
> > **W1.  I don't understand why the approach should perform well in general -- doesn't it depend on what kind of a function we train for g(a, s)?...**
> - It should be noted that our work is based on the continuous MDP setting. We have clarified this in Section 2.1. The table example in the discrete action space mentioned by the reviewer does not really fall within our scope. In our continuous MDP setting, the state-conditioned distance function $g(s, a)$ is a simple MLP and is trained with a supervised learning loss as in Eq. (4). It outputs the upper bound of the distance to the state-conditioned centroid $a_o(s)$ of the dataset, and can compel the learned policy to stay inside the convex hull of the dataset when used as a policy constraint, as demonstrated in Property 1, 2, and Figure 3 in Section 3.1. Based on our theoretical analysis (Theorem 1) and the empirical evidence presented in Section 2.2, we observe that deep Q functions can generalize better on interpolated data within the convex hull of the dataset, hence using such a distance function as a policy constraint can serve as a perfect tool to enable better offline policy learning on generalizable in-convex-hull OOD actions without suffering severe extrapolation errors.
> - It should also be noted that the proposed state-conditioned distance function $g(s,a)$ works in a very different way compared with the BC term in TD3+BC. In TD3+BC, the BC term simply minimizes the absolute sample-to-sample distance between actions from the policy and the dataset, which essentially avoids the policy from producing OOD actions. Whereas in DOGE, the state-conditioned distance function $g(s,a)$ characterizes a sample-to-dataset distance that captures the overall data geometry. An OOD action can still have relatively small $g(s,a)$ as long as they lie inside or near the boundary of the convex hull of the dataset. Enable policy learning on reliable OOD actions is the main reason why DOGE gains performance and can solve more complex tasks like AntMaze, in which TD3+BC typically fails. Theorem 3 and our experiment also provide theoretical and empirical evidence of the improved generalization ability of DOGE.
> - However, studying the distance function in discrete space is definitely an interesting topic. For instance, in a discrete action space, we may need to use a proper distance measure for discrete variables, such as Hamming distance or Manhattan distance on one-hot encoded discrete variables. We will explore it in the future and thank you for pointing it out.
>
>
> > **W2. ...Why should the proposed method be better than CQL, if it is not just for tuning issues?**
>
> - As discussed in the Introduction and Related Works in our paper, **CQL penalizes all OOD actions equally and does not consider the global dataset geometry information.** Going deeper, CQL tries to solve a $\chi^2$-divergence regularized RL problem (see Eq (13) in Appendix C in the original CQL paper), and the mode-seek nature of $\chi^2$-divergence drives CQL to assign low values for all OOD actions brutally, which impedes policy generalization. In contrast, DOGE constrains the policy based on a much milder distance constraint (Eq. (6)), which provides the global dataset geometry information and enables policy learning on more generalizable in-distribution and within or near convex-hull actions, as Figure 3, Property 1 and 2 shows.
> - Therefore, DOGE can be perceived as a convex hull based policy constraint method, allowing more exploitation at generalizable OOD actions and hence getting higher chances to find the optimal policy as compared to CQL. This is also reflected in our experiments that DOGE achieves SOTA performance on both MuJoCo and AntMaze tasks.

---

> ### Author Response · Authors · 2022-11-22
> **Follow up**
>
> Dear reviewer z7dd,
>
> Thank you for the detailed comments and suggestions to improve our paper! Regarding your concerns, we have added the discussions and ablations on $g$ network complexity. We would really appreciate it if you could tell us whether your concerns are resolved. Your further comments and discussions will greatly help to improve the quality of our paper and we will be more than happy to engage in further discussions and address them.
>
> Thanks！

---

> ### Author Response · Authors · 2022-12-08
> **A Kind Reminder in Phase 2**
>
> Dear reviewer z7dd,
>
> We're terribly sorry to bother you with this reminder. However, it's a kind reminder that about 4 days are left for the rebuttal.
>
> As the discussion period is coming to a close, we wanted to check back to see whether you have any remaining questions. **We would be happy to clarify further, and grateful for any other feedback you may provide.** We understand you may be very busy due to your work and the overlaps with NeurIPS, so we really appreciate your time engaged in the review and rebuttal phase.
>
> Thank you very much and look forward to your replies!
>
> Best regards,
>
> Authors of Paper 3842

---

### Author Response · Authors · 2022-11-13
**General Responses**

Regarding some common concerns from the reviewers, we provide the general responses as follows:

>**1: The logical flow of the theoretical analyses in our paper**
- We realized that the logical flow of our theoretical analysis is a bit long and contains too many details, which caused some confusion to reviewers. So in our revised manuscript, we added a dedicated section in Appendix A to provide a sketch of the overall logical flow in our theoretical analysis, as well as its close relationship with the proposed algorithm DOGE. We hope this can help the readers to gain a better understanding of the theoretical insights in our algorithm design.

>**2. The connection between the proposed algorithm and Theorem 2, 3.**
- Our theoretical analysis is actually closely connected to the proposed algorithm DOGE, which directly motivates our algorithm design. As we have shown in the discussion of Theorem 2 and 3 in the main text as well as the proofs in the Appendix D.1 and D.2 (Appendix C.1 and C.2 in the initial submission), the performance bound in Theorem 3 is closely related to the state-conditioned centroid of the dataset. Specifically, the proof of Theorem 2 is dependent on the centroid of dataset $x_0$ as well as the sample-to-centroid distance $d_1$ and $d_2$ (Eq.(32)-(38) in Appendix D.1). Based on Theorem 2, a small distance value will lead to a well-bounded Bellman-consistent coefficient $\mathcal{B}(v,\mu,\mathcal{F},\pi)\leq l(k)$ by finite constants $l(k)$. And $l(k)$ is further used in $L(\Pi_{\mathcal{B}})$ to construct the final performance bound of DOGE (Theorem 3, see proof in Appendix D.2). Hence control the sample-to-centroid distance plays a key role in our theoretical analysis to enable a tightened performance gap to the optimal policy. Based on the above theoretical analysis, we construct DOGE, which provides a practical method to achieve the distance regularization objective by minimizing the state-conditioned distance function $g(s,a)$, which is exactly the upper bound of the state-conditioned sample-to-centroid distance of the dataset (Property 1 in Section 3.1).

>**3. The reason to use the state-conditioned action distance function rather than the joint state-action distance function**
- Although in principle we can learn a joint state-action distance in a similar way as in Eq.(4) by sampling over joint state-action space and regressing on the distance values, it can cause severe learning difficulties. As in many RL problems, the state dimension is much larger than the action dimension, sampling over state space will need an impractically large amount of random samples and lead to a challenging high-dimensional learning problem, which makes the joint state-action distance function hard to train.
- Second, in typical offline RL setting, it is actually not necessary to use a joint state-action distance function. This is because the states $s$ and $s'$ we use for offline RL training are all from the dataset, and the distributional shift and Q-value overestimation are mainly caused by counterfactual querying on OOD actions $a'$ produced by the policy $\pi(a'|s')$ during policy evaluation. As we will never train on states outside of the datasets, hence learning the distance between a policy-generated action and dataset actions conditioned on a given state is already sufficient for policy regularization, similar to well-known policy constraints offline RL methods like BCQ [1], BEAR [2] and TD3+BC [3] but with a different divergence measure.



[1] Scott Fujimoto, David Meger, and Doina Precup. Off-policy deep reinforcement learning without exploration. ICML 2019.

[2] Aviral Kumar, Justin Fu, Matthew Soh, George Tucker, and Sergey Levine. Stabilizing off-policy q-learning via bootstrapping error reduction. NeurIPS 2019.

[3] Scott Fujimoto and Shixiang Shane Gu. A minimalist approach to offline reinforcement learning. NeurIPS 2021.

---

### Author Response · Authors · 2022-11-13
**Revision Summary**

We would like to thank the reviewers for their detailed review comments and constructive suggestions. We respond to each individual reviewer below. We’ve also updated the paper with several modifications to address the suggestions and concerns of the reviewers. The summary of changes in the updated version of the paper is as follows:

1. We've added a discussion and a sketch of the overall logical flow in our theoretical analyses. Please refer to Appendix A in our revised paper. We hope this can help the readers to better understand our paper.
2. We've added more detailed illustrative and empirical comparisons with TD3+BC in Appendix F.2.
3. We've added the comparisons with uncertainty-based methods in Appendix F.3.
4. We've added the learning curves as well as ablations on the expressivity of the proposed state-conditioned distance function in Appendix F.4.
5. We've added more quantitative experiments on Theorem 1 in Appendix F.8.

---

### Decision · Program_Chairs · 2023-01-20

**Decision:**

Accept: poster

**Justification For Why Not Higher Score:**

The connection between theoretical results and final algorithm is not very clearly motivated and communicated, and the improved RL results are not clearly shown as being connected to the motivation of the approach.

**Justification For Why Not Lower Score:**

There is interesting insight and strong empirical results in the paper. Even if it leaves us with some hanging questions, the paper as is is quite interesting, may be practically useful, and the questions may eventually be answered by other offline RL researchers who adopt these techniques.

**Metareview: Summary, Strengths And Weaknesses:**

This paper observes that deep Q function approximators trained on an offline dataset interpolate well within the convex hull of training data, even on samples that did not exist in that data. This prompts a more lenient view of "conservatism" during offline learning, which refrains from penalizing samples that the deep Q function approximator can in fact perform well on, even if they might in a strict sense be "out-of-distribution" of the offline data.

Strengths:
- Interesting and intuitive idea, framed in terms of the shape of the offline dataset and how it may permit interpolation to OOD samples.
- Works well on multiple environments.
- Interesting empirical observations about the generalization capabilities of deep Q function approximators.

Weaknesses:
- No clear empirical demonstration of DOGE having achieved a more lenient form of pessimism, or benefiting from it, even though this is the motivation of the approach. Instead, we only see final RL results, which, while impressive, can often be rather sensitive to hyperparameter tuning.
- The connection between the theoretical analysis and the actual algorithm is not communicated very well, and the writing, while improved during the response phase, is still somewhat abstruse.
- This is not a major weakness, but the method is rather close to a slew of offline RL approaches: BEAR, CQL, and TD3-BC, and differs mainly in the insight about interpolatability of deep Q networks. This could be converted to a strength by evaluating direct "plug-in" improvements for many commonly used algorithms today, but this is not currently in the paper.


**Note From Pc:**

if the above contains the word "oral" or "spotlight" please see: "oral" presentation means -> notable-top-5% and "spotlight" means -> notable-top-25%. As stated in our emails, we are disassociating presentation type from AC recommendations